# A Riemannian Framework for Learning Reduced-order Lagrangian Dynamics

**Katharina Friedl**[1]   **Noémie Jaquier**[1,2]   **Jens Lundell**[1]   **Tamim Asfour**[2]   **Danica Kragic**[1]

[1] Division of Robotics, Perception, and Learning   [2] Institute for Anthropomatics and Robotics
KTH Royal Institute of Technology          Karlsruhe Institute of Technology (KIT)
{kfriedl,jaquier,jelundel,dani}@kth.se, asfour@kit.edu

## Abstract

By incorporating physical consistency as inductive bias, deep neural networks display increased generalization capabilities and data efficiency in learning nonlinear dynamic models. However, the complexity of these models generally increases with the system dimensionality, requiring larger datasets, more complex deep networks, and significant computational effort. We propose a novel geometric network architecture to learn physically-consistent reduced-order dynamic parameters that accurately describe the original high-dimensional system behavior. This is achieved by building on recent advances in model-order reduction and by adopting a Riemannian perspective to jointly learn a non-linear structure-preserving latent space and the associated low-dimensional dynamics. Our approach enables accurate long-term predictions of the high-dimensional dynamics of rigid and deformable systems with increased data efficiency by inferring interpretable and physically-plausible reduced Lagrangian models.

## 1 Introduction

Deep learning models recently emerged as powerful tools for learning the continuous-time dynamics of physical systems. In contrast to classical system identification methods which rely on analytical derivations, black-box approaches learn to predict the behavior of nonlinear systems using large amounts of trajectory data. However, collecting such data is prohibitively expensive, and the learned models often predict trajectories that violate the laws of physics, e.g., by not conserving energy.

Gray-box approaches tackle such limitations by incorporating physics priors as inductive biases into deep learning architectures. These methods have been broadly investigated for different systems, such as Lagrangian and Hamiltonian mechanics, and function approximators, including neural networks (Lutter et al., 2019; Greydanus et al., 2019; Cranmer et al., 2020; Lutter & Peters, 2023) and Gaussian processes (Tanaka et al., 2022; Evangelisti & Hirche, 2022). Several physically-consistent models were extended to capture non-conservative forces and contact mechanics (Hochlehnert et al., 2021; Zhong et al., 2021), and to ensure longer-term stability of the identified dynamics by including differential solvers (Chen et al., 2018) into the training loop (Finzi et al., 2020; Zhong & Leonard, 2020). Model performances were also improved by representing the dynamics using coordinates adapted to the physical system at hand (Finzi et al., 2020; Celledoni et al., 2023; Duong & Atanasov, 2021). Altogether, gray-box approaches are physically consistent, versatile, more data-efficient, and display increased generalization capabilities compared to black-box methods (Lutter & Peters, 2023; Greydanus et al., 2019). Still, they have, so far, only learned the dynamics of low-dimensional systems, i.e., typically 2-5 dimensions. Learning the dynamics of high-dimensional systems, such as fluid flows, continuum mechanics, and robots, arguably remains an open problem. This is due to their increasing complexity, requiring more complex network architectures and more training data.

Predicting trajectories of high-dimensional systems is also notoriously difficult due to the computational cost of solving high-dimensional and highly nonlinear differential equations. In this context, model order reduction (MOR) techniques find a computationally efficient yet descriptive low-dimensional surrogate system — a reduced-order model (ROM) — of a given high-dimensional dynamical system or full-order model (FOM) with known dynamics (Schilders et al., 2008). MOR is typically achieved by projecting the FOM onto a lower-dimensional space via linear (Thieffry et al., 2019; Farhat et al., 2015; Carlberg et al., 2015) or nonlinear (Sharma et al., 2023; Barnett & Farhat, 2022) mappings. In particular, Autoencoders (AEs) were shown to be well suited to

extract descriptive nonlinear reduced representations from data (Lee & Carlberg, 2020; Buchfink et al., 2023; Otto et al., 2023). Recent approaches investigated the preservation of physics-induced geometric structures, i.e. Lagrangian or Hamiltonian structures, of FOMs within projection-based ROMs (Carlberg et al., 2015; Lee & Carlberg, 2020; Otto et al., 2023; Hesthaven et al., 2022). This structure awareness enables the derivation of shared theoretical properties such as energy conservation and stability preservation (Buchfink et al., 2023), as well as the design of low-dimensional control strategies on the ROM (Lepri et al., 2024). Buchfink et al. (2024) recently unified various structure-preserving MOR techniques by adopting a differential-geometric perspective on the problem. In this framework, ROMs are defined on embedded submanifolds of the high-dimensional Riemannian or symplectic manifold associated with a given Lagrangian or Hamiltonian system.

Despite their advantages, MOR methods are intrusive in that they assume known high-dimensional dynamic parameters which are generally difficult to obtain. Sharma & Kramer (2024) presented a novel non-intrusive approach that identifies low-dimensional dynamic parameters in a structure-preserving linear subspace obtained from high-dimensional state observations. However, this approach is limited to systems that are linearly reducible and displays limited expressivity by constraining the range of the ROM parameters. *In this paper*, we propose a more general and expressive approach in the form of a physics-inspired geometric deep network that learns the continuous-time dynamics of high-dimensional systems. Our *first contribution* is to adopt a differential geometry perspective and jointly learn *(1)* a structure-preserving non-linear reduced representation of the high-dimensional generalized coordinates and *(2)* the associated low-dimensional dynamic parameters. Specifically, our approach leverages a constrained AE (Otto et al., 2023) to learn a latent embedded submanifold in which physically-consistent low-dimensional dynamics are learned with a Lagrangian neural network (LNN) (Cranmer et al., 2020; Lutter & Peters, 2023). Our approach differs from (Greydanus et al., 2019; Zhong & Leonard, 2020; Botev et al., 2021) in that it does not consider high-dimensional observations (images) of low-dimensional physical systems but systems with high-dimensional state spaces. As *second* and *third contributions*, we reformulate both the LNN and the AE to account for the intrinsic geometry of their parameters. To this end, we introduce positive-definite layers in the LNN to parameterize the (reduced) mass-inertia matrix and use Riemannian optimization to infer the LNN parameters and biorthogonal AE weights.

Our approach infers physically-plausible models as both the ROM and the FOM conserve the reduced energy along reduced unforced Lagrangian trajectories and their embeddings. Moreover, it infers interpretable reduced-order dynamic parameters. We validate our approach by learning the dynamics of three simulated high-dimensional rigid and deformable systems: a pendulum, a rope, and a thin cloth. Our results demonstrate that it efficiently learns reduced-order dynamics leading to accurate long-term predictions of high-dimensional systems.

## 2 BACKGROUND

This section introduces the mathematical preliminaries and network architectures for learning reduced Lagrangian dynamics. We adopt a differential-geometric perspective to simultaneously investigate the use of nonlinear MOR projections and the structure preservation of Lagrangian systems.

### 2.1 LAGRANGIAN DYNAMICS ON THE CONFIGURATION MANIFOLD

We consider an $n$-degrees-of-freedom (DoF) mechanical system whose configuration space is identified with an $n$-dimensional smooth manifold $\mathcal{Q}$ with a simple global chart. Velocities $\dot{q}$ at each configuration $q \in \mathcal{Q}$ lie in the tangent space $\mathcal{T}_q \mathcal{Q}$, an $n$-dimensional vector space composed of all vectors tangent to $\mathcal{Q}$ at $q$. The disjoint union of all tangent spaces $\mathcal{T}_q \mathcal{Q}$ forms the tangent bundle $\mathcal{T}\mathcal{Q}$, a smooth $2n$-dimensional manifold. The configuration manifold can be equipped with a Riemannian metric, i.e., a smoothly-varying inner product acting on $\mathcal{T}\mathcal{Q}$. We consider the kinetic-energy metric, which, given a choice of local coordinates, equals the system's mass-inertia matrix $M(q)$.

A Lagrangian system is a tuple $(\mathcal{Q}, \mathcal{L})$ of a Riemannian configuration manifold $\mathcal{Q}$ and a smooth time-independent Lagrangian function $\mathcal{L} : T\mathcal{Q} \to \mathbb{R}$. The Lagrangian function is given by the difference between the system's kinetic $T(q, \dot{q})$ and potential $V(q)$ energies as $\mathcal{L}(q, \dot{q}) = T(q, \dot{q}) - V(q) = \frac{1}{2}\dot{q}^\intercal M(q)\dot{q} - V(q)$. Following the principle of least action, the equations of motion are given by the Euler-Lagrange equations $\frac{d}{dt}\left(\frac{\partial \mathcal{L}}{\partial \dot{q}}\right) - \frac{\partial \mathcal{L}}{\partial q} = \tau$ with generalized forces $\tau$. These equations are

$$M(q)\ddot{q} + c(q, \dot{q}) + g(q) = \tau, \quad \text{with} \quad c(q, \dot{q}) = \left(\frac{d}{dt}M(q)\right)\dot{q} - \frac{1}{2}\left(\frac{\partial}{\partial q}\left(\dot{q}^\intercal M(q)\dot{q}\right)\right)^\intercal, \quad (1)$$

and $g(q) = \frac{\partial V(q)}{\partial q}$, where $c(q, \dot{q})$ represents the influence of Coriolis forces. Given a time interval $\mathcal{I} = [t_0, t_f]$, trajectories $\gamma : \mathcal{I} \to \mathcal{T}\mathcal{Q} : t \mapsto (q(t), \dot{q}(t))^\mathsf{T}$ of the Lagrangian system are obtained by solving the initial value problem (IVP)

$$\begin{cases} \frac{d}{dt}\gamma\big|_t = X\big|_{\gamma(t)} = \begin{pmatrix} \dot{q}(t) \\ M^{-1}(q)(\tau - c(q,\dot{q}) - g(q)) \end{pmatrix} & \in \mathcal{T}_{\gamma(t)}\mathcal{T}\mathcal{Q}, \\ \gamma(t_0) = \begin{pmatrix} q_0 \\ \dot{q}_0 \end{pmatrix} & \in \mathcal{T}\mathcal{Q}, \end{cases} \tag{2}$$

with $X\big|_{\gamma(t)}$ the vector field defined by the Euler-Lagrange equations. Note that, for Lagrangian systems, trajectories $\gamma$ are lifted curves on the tangent bundle $\mathcal{T}\mathcal{Q}$.

## 2.2 MODEL ORDER REDUCTION

MOR methods consider a high-dimensional dynamical system, i.e., a FOM, with known dynamic parameters, which generates smooth trajectories $\gamma : \mathcal{I} \to \mathcal{M}$ characterized by an IVP of the form

$$\begin{cases} \frac{d}{dt}\gamma\big|_t &= X\big|_{\gamma(t)} \in \mathcal{T}_{\gamma(t)}\mathcal{M}, \quad t \in \mathcal{I}, \\ \gamma(t_0) &= \gamma_0 \in \mathcal{M}, \end{cases} \tag{3}$$

where $X\big|_{\gamma(t)}$ is a smooth vector field defining the evolution of the system, so that $\gamma(t) \in \mathcal{M}$. Note that we have $\mathcal{M} = \mathcal{T}\mathcal{Q}$ in (2). The goal of MOR is to accurately and efficiently approximate the set of solutions $S = \{\gamma(t) \in \mathcal{M} \mid t \in \mathcal{I}\} \subseteq \mathcal{M}$ of the IVP (3). To this end, MOR methods learn a reduced manifold $\check{\mathcal{M}}$ with $\dim(\check{\mathcal{M}}) = d \ll \dim(\mathcal{M}) = n$ and a ROM $\frac{d}{dt}\check{\gamma}\big|_t = \check{X}\big|_{\check{\gamma}(t)} \in \mathcal{T}_{\check{\gamma}(t)}\check{\mathcal{M}}$.

Following Buchfink et al. (2024), we identify $\check{\mathcal{M}}$ with an embedded submanifold $\varphi(\check{\mathcal{M}}) \subseteq \mathcal{M}$ via a smooth embedding $\varphi : \check{\mathcal{M}} \to \mathcal{M}$. The reduced initial value $\check{\gamma}_0 = \rho(\gamma_0)$ and vector field $\check{X}\big|_{\check{\gamma}(t)} = d\rho\big|_{\gamma(t)} X\big|_{\gamma(t)}$ are defined via the point and tangent reduction maps $\rho : \mathcal{M} \to \check{\mathcal{M}}$ and $d\rho\big|_x : \mathcal{T}_x\mathcal{M} \to \mathcal{T}_{\rho(x)}\check{\mathcal{M}}$ associated with $\varphi$, where $x \in \mathcal{M}$. The reduction maps must satisfy the projection properties

$$\rho \circ \varphi = \mathrm{id}_{\check{\mathcal{M}}} \quad \text{and} \quad d\rho\big|_{\varphi(\check{x})} \circ d\varphi\big|_{\check{x}} = \mathrm{id}_{\mathcal{T}_{\check{x}}\check{\mathcal{M}}}, \quad \forall \check{x} \in \check{\mathcal{M}}. \tag{4}$$

Trajectories of the original system are then obtained via the approximation $\gamma(t) \approx \varphi(\check{\gamma}(t))$. In this paper, we consider the case where the high-dimensional dynamics are unknown. We leverage structure-preserving MOR methods to learn a non-linear reduced representation $\check{\mathcal{Q}}$ of the high-dimensional Lagrangian configuration space $\mathcal{Q}$ along with a Lagrangian ROM $(\check{\mathcal{Q}}, \check{\mathcal{L}})$.

A key aspect of MOR is the construction of the embedding $\varphi$ and corresponding point reduction $\rho$. AEs are well suited for representing these nonlinear mappings due to their expressivity and scalability (Lee & Carlberg, 2020; Otto et al., 2023). An AE consists of an encoder network $\rho : \mathbb{R}^n \to \mathbb{R}^d$ that maps a high-dimensional vector $x \in \mathbb{R}^n$ to a latent representation $z \in \mathbb{R}^d$ with $d \ll n$, and a decoder network $\varphi : \mathbb{R}^d \to \mathbb{R}^n$ that reconstructs an approximation of the original data. Given a dataset $\{x_i\}_{i=1}^N$, the AE parameters $\Psi$ are trained by minimizing the reconstruction error

$$\ell_{\mathrm{AE}}(\Psi) = \frac{1}{N} \sum_{i=1}^N \|\varphi \circ \rho(x_i) - x_i\|^2. \tag{5}$$

Notice that the projection properties (4) hold approximately when the loss (5) is small. However, in this paper, we leverage a constrained AE (Otto et al., 2023) that guarantees (4), which is essential when learning reduced representations of high-dimensional Lagrangian systems.

## 2.3 LAGRANGIAN NEURAL NETWORKS

In contrast to MOR, LNN consider low-dimensional systems with unknown dynamics that we aim to learn. The key idea of LNNs is to include physics-based inductive bias into deep networks, ensuring the learned dynamics models conserve energy and lead to physically-plausible trajectories. In this context, DeLaN (Lutter et al., 2019; Lutter & Peters, 2023) proposed to model the kinetic energy (via the mass-inertia matrix) and the potential energy of a Lagrangian function $\mathcal{L}$ as two networks $M(q; \theta_\mathrm{T})$ and $V(q; \theta_\mathrm{V})$ with parameters $\theta = \{\theta_\mathrm{T}, \theta_\mathrm{V}\}$. Trajectories of the learned dynamical

system are then obtained via the equations of motion (1) by solving the IVP (2). Given a set of $N$ observations $\{\boldsymbol{q}_i, \dot{\boldsymbol{q}}_i, \ddot{\boldsymbol{q}}_i, \boldsymbol{\tau}_i\}_{i=1}^{N}$, the networks are trained to minimize the loss

$$\ell_{\text{LNN}}(\boldsymbol{\theta}) = \frac{1}{N} \sum_{i=1}^{N} \|\boldsymbol{f}(\boldsymbol{q}_i, \dot{\boldsymbol{q}}_i, \boldsymbol{\tau}_i; \boldsymbol{\theta}) - \ddot{\boldsymbol{q}}_i\|^2 + \lambda \|\boldsymbol{\theta}\|_2^2, \tag{6}$$

where $\boldsymbol{f}(\boldsymbol{q}, \dot{\boldsymbol{q}}, \boldsymbol{\tau}; \boldsymbol{\theta}) = \boldsymbol{M}^{-1}(\boldsymbol{q}; \boldsymbol{\theta}_{\text{T}})(\boldsymbol{\tau} - \boldsymbol{c}(\boldsymbol{q}, \dot{\boldsymbol{q}}) - \boldsymbol{g}(\boldsymbol{q}; \boldsymbol{\theta}_{\text{V}}))$ and $\lambda$ is a regularization constant. Following this approach, the total energy $\mathcal{E} = T(\boldsymbol{q}, \dot{\boldsymbol{q}}) + V(\boldsymbol{q}) = \frac{1}{2}\dot{\boldsymbol{q}}^{\mathsf{T}}\boldsymbol{M}(\boldsymbol{q})\dot{\boldsymbol{q}} + V(\boldsymbol{q})$, of unforced systems is conserved. Next, we propose an enhanced geometric version of DeLaN, which we then use in Section 4 to learn the dynamics of a Lagrangian ROM.

# 3 GEOMETRIC LAGRANGIAN NEURAL NETWORKS

The mass-inertia matrix $\boldsymbol{M}(\boldsymbol{q})$ acts as the Riemannian metric on the configuration manifold $\mathcal{Q}$ for Lagrangian systems with quadratic kinetic energy structure. As such, $\boldsymbol{M}(\boldsymbol{q})$ belongs to the manifold of symmetric positive-definite (SPD) matrices $\mathcal{S}_{++}^n$ (Bhatia, 2007; Pennec et al., 2006). Existing LNNs enforce the symmetric positive-definiteness by predicting the Cholesky decomposition $\boldsymbol{L}(\boldsymbol{q})$ with a Euclidean network and describing the mass-inertia matrix as $\boldsymbol{M} = \boldsymbol{L}\boldsymbol{L}^{\mathsf{T}}$. However, this parametrization overlooks the Riemannian geometry of both Cholesky and SPD spaces (see also App. B). Here, we propose to learn the mass-inertia matrix via SPD networks, whose building blocks are formulated based on the Riemannian geometry of the SPD manifold.

## 3.1 LEARNING POSITIVE-DEFINITE MASS-INERTIA MATRICES ON THE SPD MANIFOLD

Our goal is to learn a mapping $\boldsymbol{q} \mapsto \boldsymbol{M}(\boldsymbol{q}) : \mathcal{Q} \to \mathcal{S}_{++}^n$, where $\mathcal{S}_{++}^n = \{\boldsymbol{\Sigma} \in \text{Sym}^n \mid \boldsymbol{\Sigma} \succ \boldsymbol{0}\}$ is the Riemannian manifold of SPD matrices and $\text{Sym}^n$ is the space of symmetric matrices (see App. A.1 for a background on the SPD manifold and associated Riemannian operations). To this end, we build on recent works that generalize classical neural networks to operate directly on the SPD manifold by leveraging its gyrovectorspace structure (López et al., 2021; Nguyen, 2022; Nguyen et al., 2024). However, these networks consider that both inputs and outputs belong to $\mathcal{S}_{++}^n$.

We propose a novel SPD network $\boldsymbol{M}(\boldsymbol{q}; \boldsymbol{\theta}_{\text{T}}) = (g_{\mathcal{S}_{++}^n} \circ g_{\text{Exp}} \circ g_{\mathbb{R}})(\boldsymbol{q})$ with three consecutive components, as illustrated in Fig. 1-*left*. First, the input configuration $\boldsymbol{q} \in \mathcal{Q}$ is fed to a standard Euclidean multilayer perceptron (MLP) $g_{\mathbb{R}} : \mathbb{R}^n \to \mathbb{R}^{n(n+1)/2}$, whose output elements are identified with those of a symmetric matrix $\boldsymbol{U} \in \text{Sym}^n$. Second, $\boldsymbol{U}$ is interpreted as an element of the tangent space $\mathcal{T}_{\boldsymbol{P}}\mathcal{S}_{++}^n$, and mapped onto the SPD manifold via an exponential map layer $\boldsymbol{X} = \text{Exp}_{\boldsymbol{P}}(\boldsymbol{U})$. Note that this type of layers is commonly used in hyperbolic neural networks (Shimizu et al., 2021). Third, $\boldsymbol{X}$ is fed to a set of $L_{\text{spd}}$ layers $g_{\mathcal{S}_{++}^n}^{(l)} : \mathcal{S}_{++}^n \to \mathcal{S}_{++}^n$, which outputs the mass-inertia matrix $\boldsymbol{M} \in \mathcal{S}_{++}^n$. We consider and compare two different fully-connected (FC) SPD layers — which are analogous to the FC layers in MLPs —, namely the gyrocalculus-based (GyroAI) layers of (López et al., 2021; Nguyen, 2022), and the gyrospace hyperplane-based (GyroSpd$_{++}$) layers of (Nguyen et al., 2024). Both can be integrated with the ReEig layers (Huang & Gool, 2017) which are the SPD counterparts to ReLU-activations (see App. C for details on the SPD layers). In total, the parameters $\boldsymbol{\theta}_{\text{T}}$ of our SPD network consist of the MLP parameters $\boldsymbol{\theta}_{\text{T,}\mathbb{R}}$, and the SPD layer parameters $\boldsymbol{\theta}_{\text{T,}\mathcal{S}_{++}^n} \in \mathcal{S}_{++}^n \times \ldots \times \mathcal{S}_{++}^n$, including the basepoint $\boldsymbol{P}$ of the exponential map.

## 3.2 MODEL TRAINING AND PARAMETER OPTIMIZATION

The proposed geometric LNN models the kinetic energy via the mass-inertia matrix parametrized as a SPD network $\boldsymbol{M}(\boldsymbol{q}; \boldsymbol{\theta}_{\text{T}})$ and the potential energy via a standard Euclidean MLP $V(\boldsymbol{q}; \boldsymbol{\theta}_{\text{V}})$. The network parameters $\boldsymbol{\theta} = \{\boldsymbol{\theta}_{\text{T}}, \boldsymbol{\theta}_{\text{V}}\}$ are obtained as for classical LNNs by minimizing (6). However, as several parameters of the SPD network belong to $\mathcal{S}_{++}^n$, their geometry must be taken into account during training. To do so, we leverage Riemannian optimization (Absil et al., 2007; Boumal, 2023) to solve a problem of the form $\min_{\boldsymbol{x} \in \mathcal{M}} \ell(\boldsymbol{x})$ on a Riemannian manifold $\mathcal{M}$, where $\ell$ is the loss to minimize and $\boldsymbol{x} \in \mathcal{M}$ is the optimization variable. Conceptually, each iteration step in a first-order (stochastic) Riemannian optimization method consists of the three successive operations

$$\boldsymbol{\eta}_t \leftarrow h\big(\text{grad}\,\ell(\boldsymbol{x}_t), \boldsymbol{\tau}_{t-1}\big), \quad \boldsymbol{x}_{t+1} \leftarrow \text{Exp}_{\boldsymbol{x}_t}(-\alpha_t \boldsymbol{\eta}_t), \quad \boldsymbol{\tau}_t \leftarrow \text{PT}_{\boldsymbol{x}_t \to \boldsymbol{x}_{t+1}}(\boldsymbol{\eta}_t). \tag{7}$$

First, the update $\boldsymbol{\eta}_t \in \mathcal{T}_{\boldsymbol{x}_t}\mathcal{M}$ is computed as a function $h$ of the Riemannian gradient $\text{grad}\,\ell$ at the estimate $\boldsymbol{x}_t$ and of $\boldsymbol{\tau}_{t-1}$, a parallel-transported previous update $\boldsymbol{\eta}_{t-1} \in \mathcal{T}_{\boldsymbol{x}_{t-1}}\mathcal{M}$ to the new $\mathcal{T}_{\boldsymbol{x}_t}\mathcal{M}$

Figure 1: Flowchart of the forward dynamics of the proposed reduced-order LNN. The reduction mappings and embeddings of the Lagrangian ROM are depicted in blue and parametrized via a constrained AE with biorthogonal layers (*right*). The ROM dynamics are learned via a latent geometric LNN depicted in orange. The mass-inertia matrix is parametrized via a SPD network (*left*).

(see App. A for the relevant Riemannian operations). Then, an update of the estimate $x_t$ is obtained via projecting the update $\eta_t$, that is scaled by a learning rate learning rate $\alpha_t$ onto the manifold with the exponential map. Finally, the current update is parallel-transported to the tangent space of the updated estimate to prepare for the next iteration. Note that the function $h$ is determined by the optimization method. We use the Riemannian Adam (Becigneul & Ganea, 2019) implemented in Geoopt (Kochurov et al., 2020) to optimize the geometric LNN parameters. Notice that $\mathcal{M}$ is defined as a product of Euclidean and SPD manifolds to optimize the parameters $\boldsymbol{\theta}$.

## 4 LEARNING REDUCED-ORDER LAGRANGIAN DYNAMICS

Like other gray-box models, our geometric LNN faces limitations in scaling to high-dimensional systems due to the increasing complexity of their dynamics. In many cases, the solutions of high-dimensional equations of motion can be well approximated by a substantially lower-dimensional surrogate dynamic model. Building on this assumption, we propose to reduce the problem dimensionality by learning a latent dynamical system that preserves the original Lagrangian structure. The latent space and the reduced dynamics parameters are inferred jointly via structure-preserving MOR and a latent geometric LNN, as explained next. The complete forward dynamics of the proposed reduced-order LNN (RO-LNN) are illustrated in Fig. 1. Notice that we consider physical systems with high number of DoFs with observations $\{\boldsymbol{q}_i, \dot{\boldsymbol{q}}_i, \ddot{\boldsymbol{q}}_i, \boldsymbol{\tau}_i\}_{i=1}^T$, and not high-dimensional representations of inherently low-dimensional state spaces as would be the case with images.

### 4.1 LAGRANGIAN REDUCED-ORDER MODEL

Preserving the properties of the original FOM is crucial to ensure that the learned ROM displays similar behaviors. Therefore, we leverage the geometric framework introduced in Buchfink et al. (2024) and learn a reduced Lagrangian $(\check{\mathcal{Q}}, \check{\mathcal{L}})$ via structure-preserving MOR. As Lagrangian trajectories are lifted curves $\boldsymbol{\gamma}(t) = (\boldsymbol{q}(t), \dot{\boldsymbol{q}}(t))^\mathsf{T}$, the manifold to be reduced is the tangent bundle $\mathcal{T}\mathcal{Q}$. We define the lifted embedding $\varphi : \mathcal{T}\check{\mathcal{Q}} \to \mathcal{T}\mathcal{Q}$ for a smooth embedding $\varphi_{\mathcal{Q}} : \check{\mathcal{Q}} \to \mathcal{Q}$ as the pair

$$\varphi(\check{\boldsymbol{q}}, \dot{\check{\boldsymbol{q}}}) = (\varphi_{\mathcal{Q}}(\check{\boldsymbol{q}}), \ d\varphi_{\mathcal{Q}}|_{\check{\boldsymbol{q}}}\dot{\check{\boldsymbol{q}}}), \tag{8}$$

with $d\varphi_{\mathcal{Q}}|_{\check{\boldsymbol{q}}} : \mathcal{T}_{\check{\boldsymbol{q}}}\check{\mathcal{Q}} \to \mathcal{T}_{\varphi_{\mathcal{Q}}(\check{\boldsymbol{q}})}\mathcal{Q}$ denoting the pushforward, or differential, of $\varphi_{\mathcal{Q}}$ at $\check{\boldsymbol{q}}$. Analogously, we define a reduction map $\rho(\boldsymbol{q}, \dot{\boldsymbol{q}})$ as the pair $(\rho_{\mathcal{Q}}(\boldsymbol{q}), d\rho_{\mathcal{Q}}|_{\boldsymbol{q}}\dot{\boldsymbol{q}})$ of point and tangent reductions $\rho_{\mathcal{Q}} : \mathcal{Q} \to \check{\mathcal{Q}}$ and $d\rho_{\mathcal{Q}}|_{\boldsymbol{q}} : \mathcal{T}_{\boldsymbol{q}}\mathcal{Q} \to \mathcal{T}_{\rho_{\mathcal{Q}}(\boldsymbol{q})}\check{\mathcal{Q}}$.

Given $\varphi$ and $\rho$, the observed states $\{\boldsymbol{q}_i, \dot{\boldsymbol{q}}_i\}_{i=1}^T$ are first mapped to the ROM via the point reduction mapping to obtain reduced initial values. Then, the reduced Lagrangian function is constructed via the pullback of the lifted embedding as

$$\check{\mathcal{L}} = \varphi^*\mathcal{L} = \mathcal{L} \circ \varphi. \tag{9}$$

The Euler-Lagrange equations of the reduced Lagrangian yield

$$\check{\boldsymbol{M}}(\check{\boldsymbol{q}})\ddot{\check{\boldsymbol{q}}} + \check{\boldsymbol{c}}(\check{\boldsymbol{q}}, \dot{\check{\boldsymbol{q}}}) + \check{\boldsymbol{g}}(\check{\boldsymbol{q}}) = \check{\boldsymbol{\tau}}. \tag{10}$$

In the intrusive case (Buchfink et al., 2024), the reduced parameters are given as a function of the known high-dimensional dynamics as $\check{\boldsymbol{M}}(\check{\boldsymbol{q}}) = d\varphi_{\mathcal{Q}}|_{\check{\boldsymbol{q}}}^\mathsf{T} \, \boldsymbol{M}(\boldsymbol{q}) \, d\varphi_{\mathcal{Q}}|_{\check{\boldsymbol{q}}}, \check{\boldsymbol{g}}(\check{\boldsymbol{q}}) = d\varphi_{\mathcal{Q}}|_{\check{\boldsymbol{q}}}^\mathsf{T} \, \boldsymbol{g}(\boldsymbol{q})$,

$\check{\boldsymbol{\tau}} = d\varphi_{\mathcal{Q}}|_{\check{\boldsymbol{q}}}^{\mathsf{T}} \boldsymbol{\tau}$, and $\check{\boldsymbol{c}}(\check{\boldsymbol{q}}, \dot{\check{\boldsymbol{q}}})$ is computed as function of $\check{\boldsymbol{M}}(\check{\boldsymbol{q}})$ as in (1). Instead, we assume unknown dynamics and propose learn the reduced parameter with a geometric LNN. Specifically, we parametrize the reduced mass-inertia matrix and the reduced potential energy of $\check{\mathcal{L}}$ as two networks $\check{\boldsymbol{M}}(\check{\boldsymbol{q}}; \boldsymbol{\theta}_{\check{\mathrm{T}}})$ and $\check{V}(\check{\boldsymbol{q}}; \boldsymbol{\theta}_{\check{\mathrm{V}}})$ with parameters $\boldsymbol{\theta} = \{\boldsymbol{\theta}_{\check{\mathrm{T}}}, \boldsymbol{\theta}_{\check{\mathrm{V}}}\}$. The ROM (10) can then be used to efficiently compute reduced trajectories $\check{\boldsymbol{\gamma}}(t)$ as solutions $\boldsymbol{\gamma}(t) \approx \varphi(\check{\boldsymbol{\gamma}}(t))$ of the FOM (1). It is worth emphasizing that the preservation of the Lagrangian structure in the ROM guarantees the conservation of the reduced total energy $\check{\mathcal{E}}$ along the solutions $\check{\boldsymbol{\gamma}}(t)$ of (10) and the corresponding image curves $\varphi(\check{\boldsymbol{\gamma}}(t))$ as $\check{\mathcal{E}} = \mathcal{E} \circ \varphi$. This property allows us to learn physically-consistent reduced dynamics. Next, we discuss how to learn $\varphi_{\mathcal{Q}}$ and $\rho_{\mathcal{Q}}$.

## 4.2 LEARNING THE EMBEDDING AND POINT REDUCTION

For increased expressivity, we parametrize the point reduction $\rho_{\mathcal{Q}}$ and embedding $\varphi_{\mathcal{Q}}$ of our reduced Lagrangian as the encoder and decoder of an AE. To ensure that the reduction map $\rho(\boldsymbol{q}, \dot{\boldsymbol{q}})$ satisfies the projection properties (4), we leverage the constrained AE architecture introduced by Otto et al. (2023). The encoder and decoder networks are given as a composition of feedforward layers $\rho_{\mathcal{Q}} = \rho_{\mathcal{Q}}^{(1)} \circ \ldots \circ \rho_{\mathcal{Q}}^{(L)}$ and $\varphi_{\mathcal{Q}} = \varphi_{\mathcal{Q}}^{(L)} \circ \ldots \circ \varphi_{\mathcal{Q}}^{(1)}$ with $\rho_{\mathcal{Q}}^{(l)} : \mathbb{R}^{n_l} \to \mathbb{R}^{n_{l-1}}$, $\varphi_{\mathcal{Q}}^{(l)} : \mathbb{R}^{n_{l-1}} \to \mathbb{R}^{n_l}$, and $d = n_0 \leq \ldots \leq n_L = n$, where $d$ denotes the latent space dimension. Notice that the pushforward $d\varphi_{\mathcal{Q}}|_{\check{\boldsymbol{q}}}$ and tangent reduction $d\rho_{\mathcal{Q}}|_{\boldsymbol{q}}$ are the differentials of the encoder and decoder networks.

The key to guarantee the projection properties (4) lies in the specific construction of layer pairs as

$$\rho_{\mathcal{Q}}^{(l)}(\boldsymbol{q}^{(l)}) = \sigma_- \left( \boldsymbol{\Psi}_l^{\mathsf{T}}(\boldsymbol{q}^{(l)} - \boldsymbol{b}_l) \right) \qquad \text{and} \qquad \varphi_{\mathcal{Q}}^{(l)}(\check{\boldsymbol{q}}^{(l-1)}) = \boldsymbol{\Phi}_l \sigma_+(\check{\boldsymbol{q}}^{(l-1)}) + \boldsymbol{b}_l, \qquad (11)$$

where $(\boldsymbol{\Phi}_l, \boldsymbol{\Psi}_l)$ and $(\sigma_+, \sigma_-)$ are pairs of weight matrices and smooth activation functions, respectively, and $\boldsymbol{b}_l$ are bias vectors. By constraining the pairs to satisfy $\boldsymbol{\Psi}_l^{\mathsf{T}} \boldsymbol{\Phi}_l = \boldsymbol{I}_d$ and $\sigma_- \circ \sigma_+ = \mathrm{id}$, each layer satisfies $\rho_{\mathcal{Q}}^{(l)} \circ \varphi_{\mathcal{Q}}^{(l)} = \mathrm{id}_{\mathbb{R}^{n_{l-1}}}$ and the constrained AE fullfills (4). Otto et al. (2023) adhere to the first constraint by defining pairs of biorthogonal matrices $(\boldsymbol{\Phi}_l, \boldsymbol{\Psi}_l)$ via an overparametrization resulting in additional loss terms. Instead, we adopt a geometric approach that accounts for the Riemannian geometry of biorthogonal matrices. Specifically, we minimize the AE reconstruction error (5) via Riemannian optimization (see Equation (7)) by considering each pair $(\boldsymbol{\Phi}_l, \boldsymbol{\Psi}_l)$ as elements of the biorthogonal manifold $\mathcal{B}_{n_l, n_{l-1}} = \{(\boldsymbol{\Phi}, \boldsymbol{\Psi}) \in \mathbb{R}^{n_l \times n_{l-1}} \times \mathbb{R}^{n_l \times n_{l-1}} : \boldsymbol{\Psi}^{\mathsf{T}} \boldsymbol{\Phi} = \boldsymbol{I}_{n_{l-1}}\}$ (see App. A.2 for a background on the biorthogonal manifolds and associated operations). As it will be shown later in our experiments, optimizing the biorthogonal weights on the biorthogonal manifold is crucial when jointly optimizing the latent space and the associated reduced-order dynamic parameters. The second constraint is met by utilizing the smooth, invertible activation functions defined in (Otto et al., 2023, Equation 12). Additional details on the AE architecture, including activation functions, and layer derivatives, are provided in App. D.

## 4.3 MODEL TRAINING

Finally, we propose to jointly learn the parameters $\boldsymbol{\Xi} = \{\boldsymbol{\Phi}_l, \boldsymbol{\Psi}_l, \boldsymbol{b}_l\}_{l=1}^L$ of the AE and $\boldsymbol{\theta} = \{\boldsymbol{\theta}_{\check{\mathrm{T}}, \mathbb{R}}, \boldsymbol{\theta}_{\check{\mathrm{T}}, \mathcal{S}_{++}^d}, \boldsymbol{\theta}_{\check{\mathrm{V}}}\}$ of the latent geometric LNN. We consider two losses, both of which minimize the AE reconstruction error, the latent LNN loss, and a joint error on the reconstructed predictions.

**Training on acceleration.** Given a training set $\{\boldsymbol{q}_i, \dot{\boldsymbol{q}}_i, \ddot{\boldsymbol{q}}_i, \boldsymbol{\tau}_i\}_{i=1}^N$, the acceleration loss is

$$\ell_{\mathrm{ROM, acc}} = \frac{1}{N} \sum_{i=1}^N \underbrace{\|\tilde{\boldsymbol{q}}_i - \boldsymbol{q}_i\|^2 + \|\dot{\tilde{\boldsymbol{q}}}_i - \dot{\boldsymbol{q}}_i\|^2 + \|\ddot{\tilde{\boldsymbol{q}}}_i - \ddot{\boldsymbol{q}}_i\|^2}_{\ell_{\mathrm{AE}}} + \underbrace{\|\ddot{\tilde{\boldsymbol{q}}}_{\mathrm{p},i} - \ddot{\check{\boldsymbol{q}}}_i\|^2}_{\ell_{\mathrm{LNN},d}} + \underbrace{\|\ddot{\tilde{\boldsymbol{q}}}_{\mathrm{p},i} - \ddot{\boldsymbol{q}}_i\|^2}_{\ell_{\mathrm{LNN},n}} + \lambda\|\boldsymbol{\theta}\|_2^2. \qquad (12)$$

$\tilde{\boldsymbol{q}}_i, \dot{\tilde{\boldsymbol{q}}}_i, \ddot{\tilde{\boldsymbol{q}}}_i$ are the reconstructed position, velocity, and acceleration obtained by successively applying the lifted point reduction and embedding, and its derivatives, and $\ddot{\check{\boldsymbol{q}}}_i$ is the reduced acceleration. The latent acceleration predictions $\ddot{\check{\boldsymbol{q}}}_{p,i}$ are obtained from the reduced equations of motion (10) as

$$\ddot{\check{\boldsymbol{q}}}_{\mathrm{p}} = \boldsymbol{f}(\check{\boldsymbol{q}}, \dot{\check{\boldsymbol{q}}}, \check{\boldsymbol{\tau}}; \boldsymbol{\theta}) = \check{\boldsymbol{M}}^{-1}(\check{\boldsymbol{q}}; \boldsymbol{\theta}_{\check{\mathrm{T}}})(\check{\boldsymbol{\tau}} - \check{\boldsymbol{c}}(\check{\boldsymbol{q}}, \dot{\check{\boldsymbol{q}}}) - \check{\boldsymbol{g}}(\check{\boldsymbol{q}}; \boldsymbol{\theta}_{\check{\mathrm{V}}})), \qquad (13)$$

The acceleration of the FOM is then computed as $\ddot{\tilde{\boldsymbol{q}}}_{\mathrm{p}} = d\varphi_{\mathcal{Q}}|_{\check{\boldsymbol{q}}} \ddot{\check{\boldsymbol{q}}}_{\mathrm{p}} + d^2\varphi_{\mathcal{Q}}|_{(\check{\boldsymbol{q}}, \dot{\check{\boldsymbol{q}}})} \dot{\check{\boldsymbol{q}}}$, as shown in Fig. 7.

**Training with multi-step integration.** The loss (12) requires computing the second derivatives of the embedding and point reduction, i.e., the Hessian matrices of the AE, which result in significant

computational efforts increasing with the dimension of the FOM. Moreover, it considers only single steps, while the learned dynamics are expected to predict multiple steps. Therefore, we also consider a multi-step loss that which numerically integrates the latent acceleration predictions (13) before decoding. This is achieved via $H$ Euler forward steps with constant integration time $\Delta t$. Given sets of observations $\{\boldsymbol{q}_i(\mathcal{I}_i), \dot{\boldsymbol{q}}_i(\mathcal{I}_i), \boldsymbol{\tau}_i(\mathcal{I}_i)\}_{i=1}^N$ over intervals $\mathcal{I}_i = [t_i, t_i + H\Delta t]$, the multi-step loss is

$$\ell_{\text{ROM,ODE}} = \frac{1}{HN} \sum_{i=1}^N \sum_{j=1}^H \underbrace{\|\tilde{\boldsymbol{q}}_i(t_{i,j}) - \boldsymbol{q}_i(t_{i,j})\|^2 + \|\dot{\tilde{\boldsymbol{q}}}_i(t_{i,j}) - \dot{\boldsymbol{q}}_i(t_{i,j})\|^2}_{\ell_{\text{AE}}}$$
$$+ \underbrace{\|\dot{\tilde{\boldsymbol{q}}}_{\text{p},i}(t_{i,j}) - \dot{\boldsymbol{q}}_i(t_{i,j})\|^2}_{\ell_{\text{LNN},d}} + \underbrace{\|\ddot{\tilde{\boldsymbol{q}}}_{\text{p},i}(t_{i,j}) - \ddot{\boldsymbol{q}}_i(t_{i,j})\|^2}_{\ell_{\text{LNN},n}} + \gamma \|\boldsymbol{\theta}\|_2^2,$$

(14)

with latent velocity predictions $\dot{\check{\boldsymbol{q}}}_{\text{p},i}(t_{i,j}) = \int_{t_i}^{t_{i,j}} \boldsymbol{f}\left(\check{\boldsymbol{q}}_i(t), \dot{\check{\boldsymbol{q}}}_i(t), \check{\boldsymbol{\tau}}_i(t); \boldsymbol{\theta}\right)$, velocity reconstructions $\dot{\tilde{\boldsymbol{q}}}_{\text{p},i}(t_{i,j}) = d\varphi_{\mathcal{Q}}\left(\dot{\check{\boldsymbol{q}}}_i(t_{i,j})\right)$, and $t_{i,j} = t_i + j\Delta t$. Fig. 1 illustrates the resulting forward model.

## 5 EXPERIMENTS

We first evaluate the geometric LNN on a simulated 2-DoF planar pendulum. Second, we evaluate our geometric RO-LNN to learn the dynamics of three simulated high-dimensional systems: a 16-DoF pendulum, a 192-DoF rope, and a 600-DoF thin cloth. Our experiments demonstrate the ability of our approach to learn reduced-order dynamics, resulting in accurate long-term predictions. Moreover, they highlight the importance of geometry as additional inductive bias in both LNN and AE. Details about simulation environments, datasets, network architectures, and training are provided in App. F for each experiment. Additional results are provided in App. G. A video and source code are available at `https://sites.google.com/view/reduced-lagrangians`.

### 5.1 LEARNING LAGRANGIAN DYNAMICS WITH GEOMETRIC LNNS

We start by evaluating the long-term prediction accuracy of the geometric LNN introduced in Section 3 on a 2-DoF planar pendulum. Train and test trajectories are obtained by solving its equations of motion with randomly sampled initial positions and zero velocities. We compare the performance of our geometric LNN against DeLaN, which parametrizes the mass-inertia matrix via a Cholesky network. We consider two variants, where the kinetic and potential energy networks are independent (Lutter & Peters, 2023) or share parameters ($\boldsymbol{\theta}_{\text{T}} \cap \boldsymbol{\theta}_{\text{V}}$) (Lutter et al., 2019). Moreover, we evaluate the performance of different architectures for our SPD network $\boldsymbol{M}(\boldsymbol{q}; \boldsymbol{\theta}_{\text{T}})$. First, we compare two exponential map layers, where the basepoint $\boldsymbol{P} \in \mathcal{S}_{++}^n$ is either set as $\mathbf{I}$ or learned as a parameter. Second, we evaluate the influence of the SPD layers by comparing performances without any SPD layers, and with GyroAI+ReEig, GyroSpd$_{++}$, or GyroSpd$_{++}$+ReEig layers.

Fig. 2-*left* shows the acceleration prediction errors on testing data for selected architectures trained on acceleration data. We observe that our geometric LNNs consistently outperform both DeLaNs, especially in low-data regimes. Interestingly, the geometric LNNs with SPD layers do not noticeably outperform those employing solely Euclidean and exponential-map layers (see App. G.1.3 for an extensive analysis on SPD layers and App. G.1.5 for training times). Fig. 2-*middle, right* depict long-term trajectory predictions obtained via Euler integration of the predicted state derivatives. In addition to the aforementioned models, we consider several geometric LNNs trained with multi-step integration. We observe that the geometric LNNs lead to significantly better long-term predictions than the DeLaNs, with the SPD layers following the previously-observed performance trend (see also App. G.1.1). Moreover, the geometric LNN trained over 8 integration steps outperforms those trained on accelerations, and results in the most accurate predictions (see also App. G.1.4). Overall, our results validate the effectiveness of considering additional inductive bias given by *(1)* the intrinsic geometry of the mass-inertia matrix, and *(2)* multi-step integration when training LNNs. These results are further validated on a second dataset in App. G.1.2.

### 5.2 LEARNING REDUCED-ORDER LAGRANGIAN DYNAMICS

Next, we learn reduced-order dynamics of several simulated high-dimensional physical systems. Due to the limited performance improvements and increased computational complexity of the SPD layers, we only evaluate SPD networks $\boldsymbol{M}(\boldsymbol{q}; \boldsymbol{\theta}_{\text{T}}) = (g_{\text{Exp}} \circ g_{\mathbb{R}})(\boldsymbol{q})$ with $L_{\text{T}, \mathcal{S}_{++}^n} = 0$.

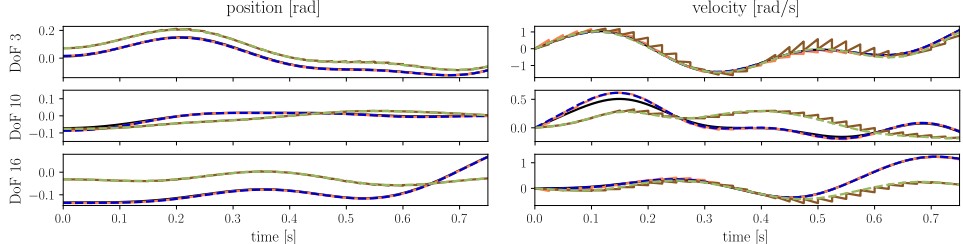

Figure 2: 2-DoF pendulum: *Left*: Median acceleration prediction error for different LNNs and training set sizes $\mathcal{D}_{\text{train}}$ over 10 test trajectories. Shaded regions represent first and third quartiles. *Middle, right:* Relative error of numerically-integrated position and velocity predictions with respect to the ground-truth trajectory over a prediction horizon $H_{\text{test}} = 2000$.

Figure 3: 16-DoF pendulum: Comparisons of the position and velocity predictions from the RO-LNNs trained on acceleration (—) and via multi-step integration (—) with the ground truth (—). The corresponding AE reconstructions (····) and (····) are depicted for completeness. The model is updated with a new initial condition $(\boldsymbol{q}_0, \dot{\boldsymbol{q}}_0)^{\mathsf{T}}$ every $0.025\,\text{s}$ ($H_{\text{test}} = 25$).

### 5.2.1 COUPLED PENDULUM (16 DoFs)

As a first ROM example, we consider a 16-DoF pendulum. We ensure that the dynamics are reducible by constraining the motion of the last 12-DoF as nonlinear combinations of the first 4-DoF. As such, the high-dimensional pendulum is reducible to 4 latent dimensions.

**Learning High-Dimensional Dynamics.** We model the high-dimensional dynamics with our RO-LNN. The parameters of the structure-preserving AE and latent geometric LNN are jointly trained via Riemannian optimization. We set the latent space dimension to $d = 4$ and the exponential map basepoint to $\mathbf{I}$ (see App. F.2 for implementation details). We consider models trained on acceleration data via (12), and with multi-step integration using (14) with $H_{\text{train}} = 8$. Fig. 3 shows the AE reconstruction and RO-LNN predictions ($H_{\text{test}} = 25$) for selected DoFs of a test trajectory. Average testing errors for different $H_{\text{test}}$ are reported in App. G.2.1. We observe that the RO-LNN trained with multi-step integration consistently outperforms the RO-LNN trained on accelerations, validating the importance of considering successive states during training. Moreover, we compare our RO-LNN with full-order LNNs that directly learn high-dimensional dynamic parameters. As shown in Table 1, even the best-performing FOM produces prediction errors that are orders of magnitude higher than the RO-LNN errors. Such erroneous acceleration predictions did not allow us to obtain stable velocity and position predictions. We also compare our models with the Lagrangian operator inference (L-OpInf) (Sharma & Kramer, 2024), and a variant of L-OpInf that uses the same linear projection method with a latent LNN for parameter identification (Sharma et al., 2024). For fair comparisons, we replace the latent network used in (Sharma et al., 2024) by a geometric LNN trained with multi-step integration (see App. F.2.2). L-OpInf shows significantly higher errors than the RO-LNNs. While L-OpInf with LNN reaches lower errors, they still remain higher than for the similarly-trained RO-LNN (ODE), thus showcasing the limitations of linear subspaces.

**Joint training.** It is important to emphasize that the quality of the AE-reconstructed states is crucial for learning accurate dynamics. If the ROM cannot effectively capture the solution space of the

Table 1: Comparison of mean and standard deviation of prediction errors over 10 test trajectories.

| | LNN | L-OpInf | L-OpInf (LNN, ODE) | RO-LNN (acc) | RO-LNN (ODE) |
|---|---|---|---|---|---|
| $\|\ddot{\tilde{\boldsymbol{q}}}_{\text{p}} - \ddot{\boldsymbol{q}}\|/\|\ddot{\boldsymbol{q}}\|$ | $(1.97 \pm 1.49) \times 10^2$ | — | — | $(5.87 \pm 4.74) \times 10^{-1}$ | $\mathbf{(3.53 \pm 2.17) \times 10^{-1}}$ |
| $\|\dot{\tilde{\boldsymbol{q}}}_{\text{p}} - \dot{\boldsymbol{q}}\|/\|\dot{\boldsymbol{q}}\|$ | — | — | — | $(1.33 \pm 0.79) \times 10^0$ | $\mathbf{(3.24 \pm 3.43) \times 10^{-1}}$ |
| $\|\tilde{\boldsymbol{q}}_{\text{p}} - \boldsymbol{q}\|/\|\boldsymbol{q}\|$ | — | $(1.31 \pm 0.85) \times 10^1$ | $(2.45 \pm 0.96) \times 10^{-1}$ | $(3.07 \pm 1.86) \times 10^{-1}$ | $\mathbf{(9.39 \pm 3.16) \times 10^{-3}}$ |

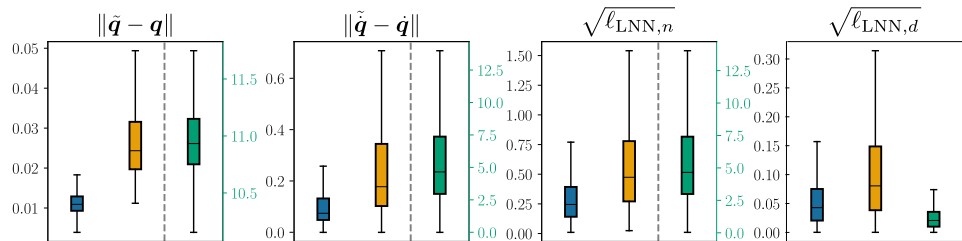

Figure 4: Median and quartiles of the errors of jointly-trained (on-biorthogonal-manifold (■) vs overparametrized (■)) and sequentially-trained (■) models. Note that the latter have their own $y$-axis in the left three plots, which is at least an order of magnitude higher.

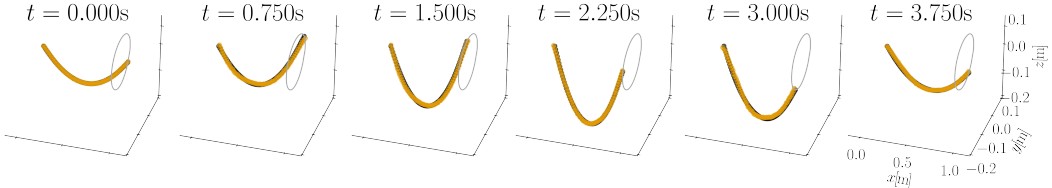

Figure 5: Predicted rope position (—) and ground truth (—) at selected timesteps for a prediction horizon $H_\text{test} = 25$. The grey circle depicts the circular trajectory of the end of the rope.

FOM, the learned dynamics may systematically deviate from the ground truth. Here, we analyze the influence of the AE on the overall performance of our RO-LNN. Specifically, we compare our model with the joint training process with *(1)* a ROM which sequentially trains the independent constrained AE and latent LNN, and *(2)* a jointly-trained ROM that employs the overparametrization of (Otto et al., 2023), and thus overlooks the geometry of the biorthogonal AE weights. All models are trained with $d = 4$ via multi-step integration with $H_\text{train} = 8$ and $\Delta t = 10^{-3}$s.

Fig. 4 shows AE reconstruction errors (position and velocity) along with the high-dimensional and latent prediction errors averaged over 10 testing trajectories. We observe that the jointly-trained models outperform the sequentially-trained one. Although the latter learns accurate dynamics in the latent space, the reconstructed high-dimensional dynamics result in significantly higher prediction errors (see the right $y$-scale). Moreover, the overparametrized model leads to higher errors in all loss components compared to the RO-LNN trained on the biorthogonal manifold. The overparametrized AE is also noticeably harder to train jointly with the latent LNN: In our experiments, 4 out of 6 trainings led to nan-values. Altogether, our results suggest that considering the intrinsic geometry of the biorthogonal manifold enhances the performance and stability of the parameter optimization (see App. G.2.2 for further comparisons).

### 5.2.2 ROPE (192 DOFs)

Next, we consider high-dimensional deformable systems and learn the dynamics of a 192-dimensional rope. Trajectories are generated by tracing a circle with one end while keeping the other fixed. We train our RO-LNN with $d = 10$ via multi-step integration (see App. F.3 for details on the data and implementation). Fig 5 shows the predicted rope configurations for a horizon of $0.025\,$s ($H_\text{test} = 25$). Our model accurately predicts the high-dimensional dynamics of the rope (see also the predictions for selected DoFs in App. G.3). Notice that our attempts to train a full-order LNN were hindered by the high dimensionality of the system. Similarly, our attempts to train L-OpInf did not result in any stable solution, thus substantiating the limitations of linear methods.

**Latent space dimension.** Selecting an appropriate latent dimension is an important consideration for MOR. Here, we study the influence of the latent dimension on the performance of our RO-LNN. Table 2 reports average reconstruction and prediction errors of RO-LNNs with latent dimensions $d = \{4, 6, 10, 14\}$ on 10 testing trajectories. We observe that errors initially decrease as the latent dimension increases, suggesting that higher-dimensional latent spaces better capture the original high-dimensional dynamics. However, the errors increase beyond a certain latent dimension, indicating that the latent LNN becomes harder to train. In other words, the choice of latent dimension trades off between latent space expressivity and the limitations of LNNs in higher dimensions.

Table 2: Rope reconstruction errors and prediction loss components from (14) for RO-LNNs.

| $d$ | $\|\bar{q} - q\|^2$ | $\|\dot{\bar{q}} - \dot{q}\|^2$ | $\ell_{\text{LNN},n}$ | $\ell_{\text{LNN},d}$ |
|---|---|---|---|---|
| 4 | $2.95 \times 10^{-2} \pm 3.24 \times 10^{-2}$ | $4.63 \times 10^{-2} \pm 7.69 \times 10^{-2}$ | $5.06^{-2} \pm 8.23 \times 10^{-2}$ | $3.00 \times 10^{-3} \pm 1.34 \times 10^{-2}$ |
| 6 | $8.19 \times 10^{-3} \pm 9.11 \times 10^{-3}$ | $1.52 \times 10^{-2} \pm 2.92 \times 10^{-2}$ | $2.40 \times 10^{-2} \pm 4.57 \times 10^{-2}$ | $\mathbf{1.81 \times 10^{-3} \pm 5.69 \times 10^{-3}}$ |
| 10 | $\mathbf{4.75 \times 10^{-3} \pm 5.73 \times 10^{-3}}$ | $\mathbf{6.53 \times 10^{-3} \pm 1.62 \times 10^{-2}}$ | $\mathbf{1.43 \times 10^{-2} \pm 3.45 \times 10^{-2}}$ | $2.87 \times 10^{-3} \pm 8.92 \times 10^{-3}$ |
| 14 | $6.16 \times 10^{-2} \pm 6.52 \times 10^{-2}$ | $5.66 \times 10^{-2} \pm 8.48 \times 10^{-2}$ | $9.22 \times 10^{-1} \pm 1.19 \times 10^{-1}$ | $5.06 \times 10^{-2} \pm 1.08 \times 10^{-1}$ |

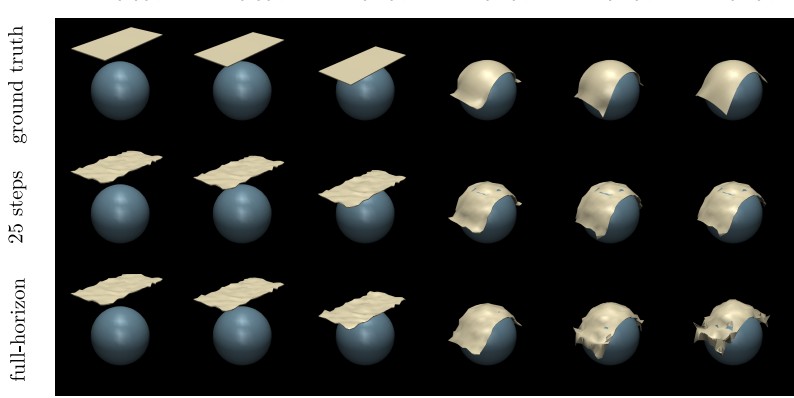

Figure 6: Predicted cloth configuration at selected times for 25-steps and 2500-steps horizons.

### 5.2.3 CLOTH (600 DoFs)

As a final example, we learn the dynamics of an even higher-dimensional deformable system: A simulated 600-DoF thin cloth falling onto a sphere. We generate different trajectories by randomly varying the sphere radius, which lead to different actuations applied to the deformable structure. We train our RO-LNN with $d = 10$ via multi-step integration on a set of $|\mathcal{D}_{\text{train}}| = 8000$ samples (see App. F.4 for details on data and implementation). Fig. 6-*middle* depicts the predicted cloth configurations for a horizon of $0.0025 \, \text{s}$ ($H_{\text{test}} = 25$), showing that our model accurately predicts the cloth high-dimensional dynamics (see also the predictions for selected DoFs and errors in App. G.4).

**Full-horizon predictions.** We evaluate the prediction quality of our model over a longer time horizon. Namely, given an initial state $(q_0, \dot{q}_0)^\mathsf{T}$, we use our model to predict the motion of the cloth for $H_{\text{test}} = 2500$ timesteps. Fig. 6-*bottom* shows the cloth configurations predicted over the full horizon (see App. G.4 for predictions of selected DoFs and energy). Our model accurately predicts the cloth high-dimensional dynamics over the full horizon, suggesting that it successfully learned the constraints and actuation effects of the cloth falling on a sphere.

## 6 DISCUSSION

This paper proposed a novel geometric network architecture for learning the dynamics of high-dimensional Lagrangian systems. By leveraging geometry and physics as inductive bias, our model is physically-consistent and conserves the reduced energy, infers interpretable reduced dynamic parameters, and effectively learns reduced-order dynamics that accurately describe the behavior of high-dimensional rigid and deformable systems. It is worth emphasizing the ubiquitous role of geometry in our model. First, considering the geometry of Lagrangian systems allows us to define nonlinear embeddings that ensure the preservation of the energy along solution trajectories and the interpretability of the model. Second, considering the geometry of full-order LNNs, latent LNNs, and AE parameters during their (joint) optimization leads to increased performance, especially at low-data regimes, in addition to guaranteeing that they belong to the manifold of interest.

It is important to note that the prediction quality can only be as good as the AE reconstruction. In other words, if the ROM cannot effectively capture the FOM solution space, the learned dynamics will systematically deviate from the ground truth. In this sense, the selection of the latent dimension is crucial, but non-trivial due to the unknown degree of coupling in the FOM. In future work, we plan to extend our approach to applications beyond system identification. In particular, we will work on the design control strategy based on the learned reduced dynamics for robotic manipulation of deformable objects and dynamic control of soft robots.

ACKNOWLEDGMENTS

This work was supported by ERC AdV grant BIRD, Knut and Alice Wallenberg Foundation and Swedish Research Council.

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

# A RIEMANNIAN MANIFOLDS

## A.1 THE MANIFOLD OF SPD MATRICES

In this section, we provide a brief overview on the Riemannian manifold of symmetric positive-definite (SPD) matrices. The set of SPD matrices is defined as $\mathcal{S}_{++}^n = \{\boldsymbol{\Sigma} \in \mathrm{Sym}^n \mid \boldsymbol{\Sigma} \succ \mathbf{0}\}$ with $\mathrm{Sym}^n$ denotes the set of symmetric $\mathbb{R}^{n \times n}$ matrices. The set $\mathcal{S}_{++}^n$ forms a smooth manifold of dimension $\dim(\mathcal{S}_{++}^n) = \frac{n(n+1)}{2}$, which can be represented as the interior of a convex cone embedded in $\mathrm{Sym}^n$. The tangent space $\mathcal{T}_{\boldsymbol{\Sigma}}\mathcal{S}_{++}^n$ at any point $\boldsymbol{\Sigma} \in \mathcal{S}_{++}^n$ is identified with the symmetric space $\mathrm{Sym}^n$ with origin at $\boldsymbol{\Sigma}$.

The SPD manifold can be endowed with various Riemannian metrics. In this paper, we consider the affine-invariant metric (Pennec et al., 2006), which is widely used due to its excellent theoretical properties. The affine-invariant metric defines the inner product $g : \mathcal{T}_{\boldsymbol{\Sigma}}\mathcal{S}_{++}^n \times \mathcal{T}_{\boldsymbol{\Sigma}}\mathcal{S}_{++}^n \to \mathbb{R}$ given, for two matrices $\boldsymbol{T}_1, \boldsymbol{T}_2 \in \mathcal{T}_{\boldsymbol{\Sigma}}\mathcal{S}_{++}^n$, as

$$\langle \boldsymbol{T}_1, \boldsymbol{T}_2 \rangle_{\boldsymbol{\Sigma}} = \mathrm{tr}(\boldsymbol{\Sigma}^{-\frac{1}{2}} \boldsymbol{T}_1 \boldsymbol{\Sigma}^{-1} \boldsymbol{T}_2 \boldsymbol{\Sigma}^{-\frac{1}{2}}). \tag{15}$$

The affine-invariant metric defines the following geodesic distance

$$d_{\mathcal{M}}(\boldsymbol{\Lambda}, \boldsymbol{\Sigma}) = \|\log(\boldsymbol{\Sigma}^{-\frac{1}{2}} \boldsymbol{\Lambda} \boldsymbol{\Sigma}^{-\frac{1}{2}})\|_{\mathrm{F}}. \tag{16}$$

To map back and forth between the Euclidean tangent space and the manifold, we employ the corresponding exponential and logarithmic maps $\mathrm{Exp}_{\boldsymbol{\Sigma}} : \mathcal{T}_{\boldsymbol{\Sigma}}\mathcal{S}_{++}^n \to \mathcal{S}_{++}^n$ and $\mathrm{Log}_{\boldsymbol{\Sigma}} : \mathcal{S}_{++}^n \to \mathcal{T}_{\boldsymbol{\Sigma}}\mathcal{S}_{++}^n$. The exponential and logarithmic maps are then computed in closed form as

$$\boldsymbol{\Lambda} = \mathrm{Exp}_{\boldsymbol{\Sigma}}(\boldsymbol{S}) = \boldsymbol{\Sigma}^{\frac{1}{2}} \exp(\boldsymbol{\Sigma}^{-\frac{1}{2}} \boldsymbol{S} \boldsymbol{\Sigma}^{-\frac{1}{2}}) \boldsymbol{\Sigma}^{\frac{1}{2}}, \tag{17}$$

$$\boldsymbol{S} = \mathrm{Log}_{\boldsymbol{\Sigma}}(\boldsymbol{\Lambda}) = \boldsymbol{\Sigma}^{\frac{1}{2}} \log(\boldsymbol{\Sigma}^{-\frac{1}{2}} \boldsymbol{\Lambda} \boldsymbol{\Sigma}^{-\frac{1}{2}}) \boldsymbol{\Sigma}^{\frac{1}{2}}, \tag{18}$$

where $\exp(\cdot)$ and $\log(\cdot)$ denote the matrix exponential and logarithm functions. The parallel transport $\mathrm{PT}_{\boldsymbol{\Sigma} \to \boldsymbol{\Lambda}} : \mathcal{T}_{\boldsymbol{\Sigma}}\mathcal{S}_{++}^n \to \mathcal{T}_{\boldsymbol{\Lambda}}\mathcal{S}_{++}^n$ allowing us to move elements between different tangent spaces is defined as

$$\tilde{\boldsymbol{T}} = \mathrm{PT}_{\boldsymbol{\Sigma} \to \boldsymbol{\Lambda}}(\boldsymbol{T}) = \boldsymbol{A}_{\boldsymbol{\Sigma} \to \boldsymbol{\Lambda}} \, \boldsymbol{T} \, \boldsymbol{A}_{\boldsymbol{\Sigma} \to \boldsymbol{\Lambda}}^{\mathsf{T}}, \tag{19}$$

with $\boldsymbol{A}_{\boldsymbol{\Sigma} \to \boldsymbol{\Lambda}} = \boldsymbol{\Lambda}^{\frac{1}{2}} \boldsymbol{\Sigma}^{-\frac{1}{2}}$. The exponential map, logarithmic map, and parallel transport are key operations for optimizing parameters on the SPD manifold in Sections 3 and 4.

## A.2 THE BIORTHOGONAL MANIFOLD

Pairs of matrices matrices $\boldsymbol{\Phi}, \boldsymbol{\Psi} \in \mathbb{R}^{n \times d}$ with $n \geq d \geq 1$ that fullfil $\boldsymbol{\Psi}^{\mathsf{T}} \boldsymbol{\Phi} = \boldsymbol{I}$ form the smooth manifold $\mathcal{B}_{n,d} = \{(\boldsymbol{\Phi}, \boldsymbol{\Psi}) \in \mathbb{R}^{n \times d} \times \mathbb{R}^{n \times d} \mid \boldsymbol{\Psi}^{\mathsf{T}} \boldsymbol{\Phi} = \boldsymbol{I}_d\}$ (Otto et al., 2023). The biorthogonal manifold $\mathcal{B}_{n,d}$ is an embedded submanifold of the Euclidean product space $\mathbb{R}^{n \times d} \times \mathbb{R}^{n \times d}$ of dimension $\dim(\mathcal{B}_{n,d}) = 2nd - d^2$. The tangent space at a point $(\boldsymbol{\Phi}, \boldsymbol{\Psi}) \in \mathcal{B}_{n,d}$ is given by

$$\mathcal{T}_{(\boldsymbol{\Phi}, \boldsymbol{\Psi})} \mathcal{B}_{n,d} = \{(\boldsymbol{V}, \boldsymbol{W}) \in \mathbb{R}^{n \times d} \times \mathbb{R}^{n \times d} \; : \; \boldsymbol{W}^{\mathsf{T}} \boldsymbol{\Phi} + \boldsymbol{\Psi}^{\mathsf{T}} \boldsymbol{V} = \mathbf{0}\}. \tag{20}$$

Vector can be projected from the embedding space onto the tangent space via the projection operation $\mathrm{Proj}_{(\boldsymbol{\Phi}, \boldsymbol{\Psi})} : \mathbb{R}^{n \times d} \times \mathbb{R}^{n \times d} \to \mathcal{T}_{(\boldsymbol{\Phi}, \boldsymbol{\Psi})} \mathcal{B}_{n,d}$ is given by

$$\mathrm{Proj}_{(\boldsymbol{\Phi}, \boldsymbol{\Psi})}(\boldsymbol{X}, \boldsymbol{Y}) = (\boldsymbol{X} - \boldsymbol{\Psi} \boldsymbol{A}, \boldsymbol{Y} - \boldsymbol{\Phi} \boldsymbol{A}^{\mathsf{T}}), \tag{21}$$

where $\boldsymbol{A}$ is the solution of the Sylvester equation $\boldsymbol{A}(\boldsymbol{\Phi}^{\mathsf{T}} \boldsymbol{\Phi}) + (\boldsymbol{\Psi}^{\mathsf{T}} \boldsymbol{\Psi}) \boldsymbol{A} = \boldsymbol{Y}^{\mathsf{T}} \boldsymbol{\Phi} + \boldsymbol{\Psi}^{\mathsf{T}} \boldsymbol{X}$.

Riemannian operations on the biorthogonal manifold are generally difficult to obtain. Therefore, we leverage a retraction $\mathrm{R}_{(\boldsymbol{\Phi}, \boldsymbol{\Psi})} : \mathcal{T}_{(\boldsymbol{\Phi}, \boldsymbol{\Psi})} \mathcal{B}_{n,d} \to \mathcal{B}_{n,d}$ as first-order approximation for the exponential map. The retraction is formulated as

$$\mathrm{R}_{(\boldsymbol{\Phi}, \boldsymbol{\Psi})}(\boldsymbol{V}, \boldsymbol{W}) = \left((\boldsymbol{\Phi} + \boldsymbol{V}) \left[(\boldsymbol{\Psi} + \boldsymbol{W})^{\mathsf{T}} (\boldsymbol{\Phi} + \boldsymbol{V})\right]^{-1}, \boldsymbol{\Psi} + \boldsymbol{W}\right). \tag{22}$$

A first-order approximation of the parallel transport operation on the biorthogonal manifold $\mathrm{PT}_{(\boldsymbol{\Phi}_1, \boldsymbol{\Psi}_1) \to (\boldsymbol{\Phi}_2, \boldsymbol{\Psi}_2)} : \mathcal{T}_{(\boldsymbol{\Phi}_1, \boldsymbol{\Psi}_1)} \mathcal{B}_{n,d} \to \mathcal{T}_{(\boldsymbol{\Phi}_2, \boldsymbol{\Psi}_2)} \mathcal{B}_{n,d}$ is then defined via the successive application of the retraction and projection as $\mathrm{PT}_{(\boldsymbol{\Phi}_1, \boldsymbol{\Psi}_1) \to (\boldsymbol{\Phi}_2, \boldsymbol{\Psi}_2)} = \mathrm{Proj}_{(\boldsymbol{\Phi}_2, \boldsymbol{\Psi}_2)} \circ \mathrm{R}_{(\boldsymbol{\Phi}_1, \boldsymbol{\Psi}_1)}$.

## B    DRAWBACKS OF THE CHOLESKY DECOMPOSITION IN LNNS

The Cholesky decomposition $M = LL^\mathsf{T}$ with lower-triangular matrix $L \in \mathbb{R}^{n \times n}$ is commonly-employed to parametrize symmetric positive-definite matrices $M \in \mathcal{S}_{++}^n$. For instance, DeLaN Lutter et al. (2019); Lutter & Peters (2023) enforces the symmetric positive-definiteness of the mass-inertia matrix $M$ by inferring the lower-triangular matrix via a Cholesky network $L(q; \theta_T)$ and reconstructing the mass-inertia matrix as $M = LL^\mathsf{T}$. While inferring $L$ guarantees the symmetric positive-definiteness of $M$, this method defines the distances between two SPD matrices based on the Euclidean distance between their Cholesky decompositions, therefore neglecting the curvature of $\mathcal{S}_{++}^n$. This choice of distance suffers from the problematic swelling effect (Feragen & Fuster, 2017; Lin, 2019), where the volume of the SPD matrices grows significantly while interpolating between two SPD matrices of identical volumes. Solutions inferred via a Cholesky decomposition will suffer from this swelling effect and lead to inaccurate predictions of the dynamics. These undesired effect can be avoided by directly inferring the mass-inertia matrix $M$ in the SPD manifold $\mathcal{S}_{++}^n$ equipped with the affine-invariant metric (Pennec et al., 2006). To do so, we parametrize the mass-inertia matrix with a SPD network $M(q; \theta_T)$, as explained in Section 3.1, and by optimizing its parameters using Riemannian optimization methods, as explained in Section 3.2.

## C    LAYER TYPES IN SPD-NETWORKS

Here, we briefly introduce the different layers used in the SPD networks of Section 3.

**Euclidean Layers.** Our SPD network leverages classical fully-connected layers to model functions that return elements on the tangent space of a manifold. The output of the $l$-th Euclidean layer $x^{(l)}$ is given by

$$x^{(l)} = \sigma(A_l x^{(l-1)} + b_l),\tag{23}$$

with $A_l \in \mathbb{R}^{n_l \times n_{l-1}}$ and $b_l \in \mathbb{R}^{n^{(l)}}$ the weight matrix and bias of the layer $l$, and $\sigma$ a nonlinear activation function of choice.

**Exponential Map Layers.** In the case of SPD manifolds, the exponential map layer is used to map layer inputs $X^{(l-1)} \in \mathrm{Sym}^n$ from the tangent space to the manifold. The layer output is given by

$$X^{(l)} = \mathrm{Exp}_P(X^{(l-1)}),\tag{24}$$

with $P \in \mathcal{S}_{++}^n$ denoting the basepoint of the considered tangent space. One option is to define $P$ as equal to the identity matrix $\mathbf{I}$. In that case, the layer input is assumed to lie in the tangent space at the origin of the cone. As the tangent spaces only give a local approximation to the manifold's curvature, the basepoint can instead be learned as a network parameter $P \in \theta_{T, \mathcal{S}_{++}^n}$ to potentially increase expressivity.

**Gyrocalculus-based FC SPD Layers (GyroAI).** Together with gyrocalculus formulations for the SPD manifold, López et al. (2021) and Nguyen (2022) presented a version of fully connected layers for SPD data. With $\oplus$ as addition and $\otimes$ as scaling operations for SPD gyrovectors,

$$X^{(l)} = \left(A_l \otimes X^{(l-1)}\right) \oplus B_l,\tag{25}$$

defines the $l$-th gyrolayer output $X^{(l)} \in \mathcal{S}_{++}^n$ for the input $X^{(l-1)} \in \mathcal{S}_{++}^n$. Here, the gyrovector operations correspond to $A \otimes X = \exp(A * \log(X))$ with $*$ denoting pointwise multiplication and $X \oplus B = X^{\frac{1}{2}} B X^{\frac{1}{2}}$. Matrices $A_l, B_l \in \mathcal{S}_{++}^n$ are part of the network parameters $\theta_{T, \mathcal{S}_{++}^n}$.

**Gyrospace hyperplane-based FC SPD Layers (GyroSpd$_{++}$).** Inspired by their equivalent in hyperbolic spaces (Shimizu et al., 2021), another version of a gyrocalculus-based FC layer for SPD networks was proposed by Nguyen et al. (2024). Here, the classic combination of matrix product with a weight and addition of a bias is treated as hyperplane equation. Therefore, these fully-connected layers compute the distance to a hyperplane that is located in the origin of the manifold from a specific point $V$, that is obtained from the layer input $X^{(l-1)} \in \mathcal{S}_{++}^n$ via the gyrocalculus formulation of the hyperplane equation

$$V(X^{(l-1)}) = \langle \ominus B_l \oplus X^{(l-1)}, A_l \rangle,\tag{26}$$

that is parametrized by the network parameters $\boldsymbol{A}_l, \boldsymbol{B}_l \in \mathcal{S}_{++}^n$. The additive inverse $\ominus$ fulfills $\ominus \boldsymbol{X} = \boldsymbol{X}^{-1}$. The final layer output is given by

$$\boldsymbol{X}^{(l)} = \exp((\boldsymbol{X}^{(l-1)}) * \boldsymbol{I}_{\mathrm{v}}), \tag{27}$$

where $\boldsymbol{I}_{\mathrm{v}(i,j)} = \frac{1}{\sqrt{2}}$ for $i \neq j$ and $\boldsymbol{I}_{\mathrm{v}(i,j)} = 1$ for $i = j$.

**ReEig Nonlinearities.** The ReEig layers were first introduced by Huang & Gool (2017), and were extended to a more general context in Nguyen (2022). Even though $\mathcal{S}_{++}^n$ already classifies as non-linear space, this type of layer adds the option of adding further nonlinearities to the network. The layer output is defined as

$$\boldsymbol{X}^{(l)} = \sigma_{\mathrm{spd}}(\boldsymbol{X}^{(l-1)}) = \boldsymbol{U}\mathrm{diag}(\max(\epsilon\mathbf{I}, \boldsymbol{V}))\boldsymbol{U}^\mathsf{T}, \tag{28}$$

with eigenvalue decomposition of $\boldsymbol{X}^{(l-1)} = \boldsymbol{U}\mathrm{diag}(\boldsymbol{V})\boldsymbol{U}^\mathsf{T}$. The constant $\epsilon$ acts as rectification threshold to avoid smaller eigenvalues and hereby regularize the output. Under the affine-invariant distance, this nonlinearity layer is analogous to a ReLu nonlinearity in the Euclidean network layers.

# D  CONSTRAINED AE: IMPLEMENTATION DETAILS

The constrained AE is implemented following (Otto et al., 2023). To guarantee the projection properties, the nonlinear activation functions employed in the encoder and decoder network $\sigma_-$ and $\sigma_+$ must satisfy $\sigma_- \circ \sigma_+ = \mathrm{id}$. To do so, they are defined by

$$\sigma_\pm(x_i) = \frac{bx_i}{a} \mp \frac{\sqrt{2}}{a\sin(\alpha)} \pm \frac{1}{a}\sqrt{\left(\frac{2x_i}{\sin(\alpha)\cos(\alpha)} \mp \frac{\sqrt{2}}{\cos(\alpha)}\right) + 2a}, \tag{29}$$

with

$$\begin{cases} a &= \csc^2(\alpha) - \sec^2(\alpha), \\ b &= \csc^2(\alpha) + \sec^2(\alpha). \end{cases} \tag{30}$$

The activations then resemble smooth, rotation-symmetric versions of the common leaky ReLu activations. The parameter $0 < \alpha < \frac{\pi}{4}$ sets the slope of the activation functions. Throughout our experiments, we set $\alpha = \frac{\pi}{8}$.

For each layer $l$, the weight matrices are given by the pair $(\boldsymbol{\Phi}_l, \boldsymbol{\Psi}_l) \in \mathcal{B}_{n_l, n_{l-1}}$ (see App. A.2). In the proposed RO-LNN, we train these weights via Riemannain optimization on the biorthogonal manifold, analogous to the Riemannian optimization on the SPD manifold presented in Section 3.2. Notice that the optimization of the AE parameters $\boldsymbol{\Xi} = \{\boldsymbol{\Phi}_l, \boldsymbol{\Psi}_l, \boldsymbol{b}_l\}_{l=1}^L$ takes place on a product of Euclidean and biorthogonal manifolds.

When learning dynamic latent space for Lagrangian MOR, the AE should learn the lifted embedding (8) and corresponding reduction maps. Therefore, the outputs of the encoder and decoder networks are differentiated with respect to their inputs. In our implementation, we take layerwise analytical derivatives and obtain the full differentials via the chain rule. A second derivative of the encoder and decoder networks enables training on acceleration data. Full-dimensional acceleration data is encoded via the reduction map $\rho_\mathcal{Q}$ as $\ddot{\tilde{\boldsymbol{q}}} = d\rho_\mathcal{Q}|_{\boldsymbol{q}}\ddot{\boldsymbol{q}} + d^2\rho_\mathcal{Q}|_{(\boldsymbol{q},\dot{\boldsymbol{q}})}\dot{\boldsymbol{q}}$, and latent accelerations are reconstructed via the embedding $\varphi_\mathcal{Q}$ as $\ddot{\tilde{\boldsymbol{q}}} = d\varphi_\mathcal{Q}|_{\tilde{\boldsymbol{q}}}\ddot{\tilde{\boldsymbol{q}}} + d^2\varphi_\mathcal{Q}|_{(\tilde{\boldsymbol{q}},\dot{\tilde{\boldsymbol{q}}})}\dot{\tilde{\boldsymbol{q}}}$. This requires the expensive computation of second-order derivatives, but is nevertheless part of our experiments.

Given a dataset of $T$ observations $\{\boldsymbol{q}_i, \dot{\boldsymbol{q}}_i, \ddot{\boldsymbol{q}}_i\}_{i=1}^T$, the AE is trained on a loss

$$\ell_{\mathrm{cAE}} = \frac{1}{N}\sum_{i=1}^N \|\tilde{\boldsymbol{q}}_i - \boldsymbol{q}_i\|^2 + \|\dot{\tilde{\boldsymbol{q}}}_i - \dot{\boldsymbol{q}}_i\|^2 + \|\ddot{\tilde{\boldsymbol{q}}}_i - \ddot{\boldsymbol{q}}_i\|^2, \tag{31}$$

of position reconstructions $\tilde{\boldsymbol{q}}$, velocity reconstructions $\dot{\tilde{\boldsymbol{q}}}$, and acceleration reconstruction $\ddot{\tilde{\boldsymbol{q}}}$. If the velocity or acceleration encodings are not relevant, the respective terms can simply be removed.

We compare the performance of the constrained AE trained via Riemannian optimization with the training scheme introduced in (Otto et al., 2023), where the biorthogonality of weights is achieved

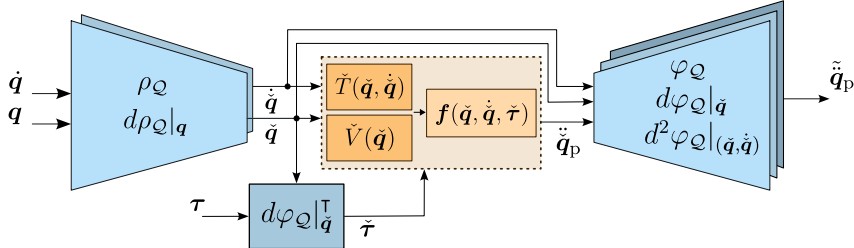

Figure 7: Flowchart of the forward dynamics of the proposed reduced-order LNN trained on acceleration via (12). The reduction mappings and embeddings of the Lagrangian ROM are depicted in blue and parametrized via a constrained AE with biorthogonal layers. The ROM dynamics are learned via a latent geometric LNN depicted in orange. The mass-inertia matrix is parametrized via the SPD network described in Section 3.

via the introduction of additional penalty losses. In this case, the overall loss only reaches a minimum value if the overparametrized weight matrices actually fulfill the biorthogonality conditions for each layer $l$. The overparametrized constrained AE loss is given by

$$\ell_{\text{cAE, op}} = \ell_{\text{cAE}} + \ell_{\text{cAE, reg}} \quad \text{with} \quad \ell_{\text{cAE, reg}} = \sum_{l=1}^{L} \|\boldsymbol{\Phi}_{\text{e}}^{(l)}\|_{\text{F}}^2 + \|\boldsymbol{\Phi}_{\text{d}}^{(l)}\|_{\text{F}}^2 + \|\boldsymbol{\Phi}_{\text{e}}^{(l)\mathsf{T}}\boldsymbol{\Phi}_{\text{d}}^{(l)} - \mathbf{I}\|_{\text{F}}^2 \|(\boldsymbol{\Phi}_{\text{e}}^{(l)\mathsf{T}}\boldsymbol{\Phi}_{\text{d}}^{(l)})^{-1}\|_{\text{F}}^2,$$

(32)

with the loss $\ell_{\text{cAE, reg}}$ regularizing the biorthogonality of the AE weights. Note that this overparametrization does not guarantee the biorthogonality condition, in contrast to the Riemannian approach. In practice, the additional regularization loss $\ell_{\text{cAE, reg}}$ required an additional learning rate, resulting in more possible combinations of hyperparameters.

## E    TRAINING ON ACCELERATION

Fig. 7 illustrates the complete forward dynamics of the proposed RO-LNN trained on acceleration by minimizing the loss (12).

## F    ADDITIONAL EXPERIMENTAL DETAILS

This section presents additional details on the experimental setup of Section 5. Position and velocity predictions are obtained via Euler forward integration.

### F.1    2-DoF PENDULUM OF SECTION 5.1.

#### F.1.1    SIMULATION AND DATA GENERATION

For this experiment, we use a 2-DoF pendulum implemented in MUJOCO (Todorov et al., 2012). The pendulum is connected via hinge joints. Both links $i$ are modeled as cylinders of radius $r_i = 0.025\,\text{m}$, length $l_i = 0.4\,\text{m}$, and mass $m_i = 0.1\,\text{kg}$. The simulated environment is entirely free of dissipative effects. We generate two different datasets $\mathcal{D}_{\text{pend}} = \{\{\boldsymbol{q}_{n,k}, \dot{\boldsymbol{q}}_{n,k}, \ddot{\boldsymbol{q}}_{n,k}, \boldsymbol{\tau}_{n,k}\}_{k=1}^{K}\}_{n=1}^{N}$ consisting of $N$ trajectories with $K$ samples per trajectory. The state evolution is simulated in MUJOCO with an RK4-solver with timestep $\Delta t = 10^{-3}\,\text{s}$ over a time interval $\mathcal{I} = [0, T]\text{s}$.

**Dataset 1: Unactuated pendulum.** The initial configurations are randomly sampled from the interval $q_i(t = 0) \in [0, 30]\,°$ for each DoF $i = \{1, 2\}$. The initial velocities are set to $\dot{\boldsymbol{q}}(t = 0) = \boldsymbol{0}$. Each trajectory is recorded for $T = 2\text{s}$. All models presented in Section 5.1 in the main text are trained on this dataset.

**Dataset 2: Sine-Tracking pendulum.** Each joint is controlled to follow a reference trajectory via an inverse dynamics torque control law. The reference trajectories are set to follow sinusoidal shapes as $q_{\text{ref},i} = A_i \sin(2\pi f_i t + \phi)$, with $\phi_i = \arcsin\left(\frac{q_{i,0}}{A_i}\right)$, $q_{i,0} \in [0, 30]\,°$, $A_i \in [1, 30]\,°$, $f_i \in \left[\frac{1}{15}, 1\right]\text{s}^{-1}$, and $i = \{1, 2\}$. $\dot{\boldsymbol{q}}_{\text{ref},i}$ and $\ddot{\boldsymbol{q}}_{\text{ref},i}$ are obtained as first and second derivatives of $q_{\text{ref},i}$ with respect to time. Initial positions and velocity are set to match the reference trajectories as

$q_i(t = 0) = q_{\text{ref},i}(t = 0)$ and $\dot{q}_i(t = 0) = \dot{q}_{\text{ref},i}(t = 0)$ Each trajectory is recorded for $T = 3.5\,\text{s}$. Results of models trained on this datasets are presented in App. G.1.

### F.1.2 ARCHITECTURES & TRAINING

**Geometric LNN architecture.** For all models, the potential energy network $V(\boldsymbol{q}; \boldsymbol{\theta}_V)$ consists of $L_V = 2$ hidden Euclidean SoftPlus layers of $64$ neurons. The kinetic energy of all models is parametrized via the mass-inertia matrix of the system. We consider several architectures for the mass-inertia matrix network $\boldsymbol{M}(\boldsymbol{q}; \boldsymbol{\theta}_T)$, which differ in their components $g_{\mathbb{R}}$, $g_{\text{Exp}}$, and $g_{\mathcal{S}_{++}^n}$ as follows:

- $g_{\mathbb{R}}$: The Euclidean component consists of $L_{T,\mathbb{R}}$ Euclidean layers on the tangent space of the SPD manifold with SoftPlus activations and $64$ neurons per layer.

- $g_{\text{Exp}}$: The basepoint $\boldsymbol{P} \in \mathcal{S}_{++}^n$ is either set as the origin $\boldsymbol{I}$, or learned as a network parameter $\boldsymbol{P_\theta} \in \mathcal{S}_{++}^n$.

- $g_{\mathcal{S}_{++}^n}$: The SPD component consists of $L_{T,\mathcal{S}_{++}^n}$ hidden SPD layers. We consider two different FC SPD layers, namely GyroAI (25), and GyroSpd$_{++}$ (26). Both FC layers can be augmented with a ReEig nonlinearity (28). The regularization constant $\epsilon$ of the nonlinearity is uniformly set as $\epsilon = 1 \times 10^{-4}$. Note that for the SPD layers, the common notion of number of neurons per layer or layer width is not directly applicable. As the layers act as a transformation $\mathcal{S}_{++}^n \to \mathcal{S}_{++}^n$ via network parameters $\boldsymbol{\theta}_{T,\mathcal{S}_{++}^n}$, they can be interpreted as carrying a single transformation per layer, or as transformations of manifold dimension $\dim(\mathcal{S}_{++}^n) = \frac{n(n+1)}{2}$.

**Baseline LNN architectures.** We compare the proposed geometric LNN against DeLaN, which parametrizes the mass-inertia matrix via a Cholesky network. We consider two variants here: *(1)* The kinetic and potential energy networks are independent as in (Lutter & Peters, 2023). Both networks consist of 2 Euclidean SoftPlus layers of $64$ neurons. Hereby, the potential energy network is fully equivalent to the ones employed in the geometric LNN. The kinetic energy network returns a Cholesky decomposition $\boldsymbol{L}(\boldsymbol{q})$ of the mass-inertia matrix $\boldsymbol{M}(\boldsymbol{q})$; *(2)* The kinetic and potential energy networks share parameters, i.e., $\boldsymbol{\theta}_T \cap \boldsymbol{\theta}_V$ as in (Lutter et al., 2019). The MLP consists of 2 hidden Euclidean SoftPlus layers of $64$ neurons, and separate output layers return the potential energy and the Cholesky decomposition.

**Training on Acceleration Data.** Each training dataset $\mathcal{D}_{\text{train}}$ consist of randomly-sampled data from 30 trajectories from one of the datasets presented in Section F.1.1. The models are trained by minimizing the LNN loss (6) for 3000 epochs with architecture-specific learning rates. If not further specified, both $g_{\mathbb{R}}$ and $g_{\mathcal{S}_{++}^n}$ consist of each 2 layers.

**Training on multi-step Integration.** For these experiments, we randomly sample recordings $i$ form 30 trajectories, which then form the training dataset $\mathcal{D}_{\text{train}} = \left\{ \{\boldsymbol{q}\}_{t_{0,i}}^{t_{0,i}+H\Delta t}, \{\dot{\boldsymbol{q}}\}_{t_{0,i}}^{t_{0,i}+H\Delta t}, \{\boldsymbol{\tau}\}_{t_{0,i}}^{t_{0,i}+H\Delta t} \right\}$. Note that this training dataset depends on the ODE solver's timestep $\Delta t$ and desired prediction horizon $H$. The models are trained by minimizing the loss

$$\ell_{\text{LNN}} = \frac{1}{HN} \sum_{i=1}^{N} \sum_{j=1}^{H} \int_{t_i}^{t_i+j\Delta t} \|\dot{\tilde{\boldsymbol{q}}}_{\text{p},i}(t_{i,j}) - \dot{\tilde{\boldsymbol{q}}}_i(t_{i,j})\|^2 + \|\tilde{\dot{\boldsymbol{q}}}_{\text{p},i}(t_{i,j}) - \dot{\boldsymbol{q}}_i(t_{i,j})\|^2 + \gamma \|\boldsymbol{\theta}\|_2^2 \quad (33)$$

for 1500 epochs with architecture-specific learning rates. Notice that the loss (33) is the equivalent of (14) for FOMs. In this category, the featured kinetic energy networks all consist of 2 hidden layers, and the exponential map layer is implemented with respect to $\boldsymbol{I}$.

**Long-term prediction Experiments.** In those experiments, we use both models trained on acceleration data and with multi-step integration. In both scenario, the training datasets contain $|\mathcal{D}_{\text{train}}| = 8000$ samples and are trained for 2000 epochs.

Table 3: Nonlinear combinations for DoFs 5 to 16 of the coupled pendulum.

| DoF | $f(q_1, q_2, q_3, q_4)$ |
|---|---|
| $q_5$ | $q_3 - \cos(q_2)$ |
| $q_6$ | $q_1 + 0.1\sin(q_2)$ |
| $q_7$ | $q_4 \cos(q_2)$ |
| $q_8$ | $q_1 + q_3^2$ |
| $q_9$ | $1.5 \sin(q_2)$ |
| $q_{10}$ | $-q_4 q_0$ |
| $q_{11}$ | $\sin(q_1)$ |
| $q_{12}$ | $0.4 q_3 q_4$ |
| $q_{13}$ | $-0.9 q_1 - q_2 + q_3 - 2q_4^2$ |
| $q_{14}$ | $-3 \sin(q_3)$ |
| $q_{15}$ | $-2q_3^2$ |
| $q_{16}$ | $-0.9 q_1^2$ |

## F.2  Coupled Pendulum (16 DoFs) of Section 5.2.1

### F.2.1  Simulation and Data Generation

In this experiment, we consider a 16-DoF coupled pendulum. The pendulum is connected via hinge joints. All links $i$ are modeled as capsules of radius $r_i = 0.05\,\mathrm{m}$, cylinder length $l_i = 0.5\,\mathrm{m}$, and mass $m_i = 1\,\mathrm{kg}$. The simulated environment is entirely free of dissipative effects. We generate a dataset $\mathcal{D}_{\mathrm{pend16}} = \{\{\boldsymbol{q}_{n,k}, \dot{\boldsymbol{q}}_{n,k}, \ddot{\boldsymbol{q}}_{n,k}, \boldsymbol{\tau}_{n,k}\}_{k=1}^{K}\}_{n=1}^{N}$ consisting of $N$ trajectories with $K$ samples per trajectory. The first 4-DoF are simulated in MuJoCo (Todorov et al., 2012) with an RK4-solver with timestep $\Delta t = 10^{-3}\,\mathrm{s}$ over a time interval $\mathcal{I} = [0,3]$s, while the last 12 DoFs are constrained to nonlinear combinations of the first 4, as described in Table 3. The initial configurations of the first 4 DoFs are randomly sampled from the interval $q_i(t = 0) \in [0, 30]\,^\circ$. The initial velocities are $\dot{\boldsymbol{q}}(t = 0) = \boldsymbol{0}$.

### F.2.2  Architectures & Training

Our RO-LNN is composed of a constrained AE and a latent geometric LNN. We consider a latent space of dimension $d = n_0 = 4$ to match the known underlying dimension of the high-dimensional pendulum dynamics. The constrained AE is implemented with 4 biorthogonal encoder and decoder layers $\rho_{\mathcal{Q}}^{(l)} : \mathbb{R}^{n_l} \to \mathbb{R}^{n_{l-1}}$ and $\varphi_{\mathcal{Q}}^{(l)} : \mathbb{R}^{n_{l-1}} \to \mathbb{R}^{n_l}$ of sizes $n_l = [8, 16, 16, 16]$. For the latent geometric LNN, both the potential and kinetic energy network consist of 2 hidden Euclidean layers of 64 neurons. The kinetic energy network employs an exponential map layer with basepoint $\mathrm{Exp}_{\mathbf{I}}$.

We consider two versions of the RO-LNN. The first model is trained on acceleration data by minimizing loss (12). The second model is trained with multi-step integration by minimizing (14). We consider $H_{\mathrm{train}} = 8$ latent integration steps and an integration time constant of $10^{-3}\,\mathrm{s}$. The losses are optimized using Riemannian Adam. For the first model, we use a learning rate of $5 \times 10^{-2}$ for the AE parameters $\boldsymbol{\Xi}$, $1 \times 10^{-5}$ for the LNN parameters $\boldsymbol{\theta}$, and a regularization $\gamma = 1 \times 10^{-6}$. For the second model, the learning rate for the parameters of the LNN is $2 \times 10^{-4}$ with $\gamma = 2 \times 10^{-5}$. Both models are trained until convergence, i.e., for 4000 and 3000 epochs, respectively.

We compare our RO-LNN with *(1)* a full-order geometric LNNs, *(2)* L-OpInf (Sharma & Kramer, 2024), and *(3)* L-Opinf with a latent LNN, as described in (Sharma et al., 2024). For the full-order geometric LNNs, we only report results of the best-performing FOM out of 10 models. The kinetic energy network consists of $L_{\mathrm{T},\mathbb{R}} = 2$ Euclidean hidden SoftPlus layers with 128 neurons, an exponential map layer with basepoint $\boldsymbol{P} = \mathbf{I}$, and no SPD layers, i.e., $L_{\mathrm{T},\mathcal{S}_{++}^n} = 0$. The potential energy network is likewise composed of 2 Euclidean SoftPlus layers of 128 neurons. The full-order LNN was trained for 3000 epochs. The learning rate for the LNN parameters $\boldsymbol{\theta}$ is set to $1 \times 10^{-5}$, and $\gamma = 1 \times 10^{-5}$.

For L-OpInf (Sharma & Kramer, 2024), we consider a projection onto a 4-dimensional linear subspace, as for our RO-LNN. We implemented L-OpInf in Python and solved the optimization problem leading to the reduced equations of motion with CVXPY. As described in (Sharma & Kramer, 2024),

we set the low-dimensional mass-inertia matrix as a constant identity matrix, i.e., $\check{M} = \mathbf{I}_d$. In our experiments, we optimize for $15000$ samples of ground-truth position data.

For L-OpInf with latent LNN (Sharma et al., 2024), we considered the same projection to a 4-dimensional linear subspace as for the L-OpInf baseline. For fair comparisons, we enhance the latent LNN compared to that used in (Sharma et al., 2024). In our experiment, the latent dynamic parameters are learned by a geometric LNN trained via the multi-step integration loss (14), for which we use the same hyperparameters as for the RO-LNN trained with the same loss. Note that, for both versions of L-OpInf, the model is fed with ground truth data every $H_{\text{test}} = 25$ steps to maintain consistency in comparison with the RO-LNN. We used the same Euler forward integration scheme as with our other models.

When investigating joint training in Section 5.2, the previously-described RO-LNN trained with multi-step integration is compared to *(1)* a version trained via the overparametrization of biorthogonal weights, and *(2)* a model that was trained separately, by sequentially training the AE and latent LNN. The overparametrized model uses the overparametrized loss (32), with $\ell_{\text{cAE, reg}}$ multiplied by a weighting constant $1 \times 10^{-5}$. For the sequentially-trained model, the first $3000$ epochs were trained using only the AE components $\ell_{\text{AE}}$ of the loss (14). For the consecutive $3000$ epochs, only the LNN components $\ell_{\text{LNN}}$ were active of (14). All models were trained on $|\mathcal{D}_{\text{train}}| = 24000$ samples, which were randomly sampled from $20$ trajectories. The testing dataset consists of $10$ trajectories.

### F.3  ROPE (192 DoFs) OF SECTION 5.2.2

#### F.3.1  SIMULATION AND DATA GENERATION

The rope is implemented in MUJOCO via an elastic cable. Over a total length of $1\,\text{m}$, the rope consists of $i = 64$ equally-spaced capsule-shaped masses of $m_i = 0.1\,\text{kg}$ and $r_i = 0.02\,\text{m}$. Twisting and bending stiffness are set to $5 \times 10^5\,\text{Pa}$, and the damping value of the joints is set to $0.1$. As the masses are connected via ball joints, the rope can move in all directions of the workspace. For training and testing, we generate the datasets $\mathcal{D}_{\text{rope}} = \{\{\boldsymbol{q}_{n,k}, \dot{\boldsymbol{q}}_{n,k}, \boldsymbol{\tau}_{n,k}\}_{k=1}^{K}\}_{n=1}^{N}$ consisting of $N$ trajectories with $K$ samples per trajectory. The generalized coordinates of the system are chosen to be the Cartesian positions $q_i = [x_i, y_i, z_i]$ of each mass's center of mass in the world frame. To generate trajectories, we consider a scenario where one end of the rope is fixed to the origin and the other end is controlled to move along a circular trajectory of radius $r$. This mimics a handheld manipulation task.

For each reference trajectory, a radius $r_i$ is sampled randomly from $r_i \in [0.05, 0.4]\,\text{m}$. To achieve non-planar motions, the circles are tilted by a random angle $\theta \in [-90, 90]^\circ$. The end of the rope is set to track such a reference at an angular velocity $120^\circ\,\text{s}^{-1}$ via a PD control. The state evolution is simulated with an RK4-solver with timestep $\Delta t = 10^{-3}\,\text{s}$ over a time interval $\mathcal{I} = [0, 3.8]\text{s}$.

#### F.3.2  ARCHITECTURES & TRAINING

For our experiments on the rope, we consider four RO-LNNs that only differ from one another in the dimensions of the latent space. Each network is composed of a constrained AE and a latent geometric LNN. The constrained AE is implemented with $4$ biorthogonal encoder and decoder layers $\rho_{\mathcal{Q}}^{(l)} : \mathbb{R}^{n_l} \to \mathbb{R}^{n_{l-1}}$ and $\varphi_{\mathcal{Q}}^{(l)} : \mathbb{R}^{n_{l-1}} \to \mathbb{R}^{n_l}$ of sizes $n_l = [64, 64, 128, 192]$. We consider latent spaces of dimension $d = \{4, 6, 10, 14\}$. In all scenarios, the kinetic energy network of the latent geometric LNN consists of $2$ hidden Euclidean layers of $64$ neurons with SoftPlus activations, and an exponential map layer $\text{Exp}_{\mathbf{I}}$. The potential energy network also consists of $2$ hidden Euclidean layers of $64$ neurons with SoftPlus activations.

All models are trained on $H_{\text{train}} = 8$ integration steps by minimizing the loss (14) with Riemannian Adam (Becigneul & Ganea, 2019). All models are trained with a learning rate of $8 \times 10^{-2}$ for the AE parameters $\boldsymbol{\Xi}$, $1 \times 10^{-5}$ for the LNN parameters $\boldsymbol{\theta}$, with a regularization $\gamma = 1 \times 10^{-6}$. All models are trained until convergence for $3000$ epochs on $24000$ samples of training data, sampled randomly from $20$ different trajectories. For the testing dataset, we record $10$ full trajectories.

### F.4 Cloth (600 DoFs) of Section 5.2.3

#### F.4.1 Simulation and Data Generation

The deformable thin cloth is modelled in MUJOCO as a flexible composite object with 200 masses of $m_i = 0.1\,\text{kg}$ equally spaced over a width of $0.1\,\text{m}$ and length of $0.2\,\text{m}$. For this model, the generalized coordinates are given by the Cartesian positions $q_i = [x_i, y_i, z_i]$ of each masses' center of mass in the world frame.

For training and testing, we generate datasets $\mathcal{D}_{\text{cloth}} = \{\{\boldsymbol{q}_{n,k}, \dot{\boldsymbol{q}}_{n,k}, \boldsymbol{\tau}_{n,k}\}_{k=1}^{K}\}_{n=1}^{N}$ consisting of $N = 20$ trajectories during training, and $N = 10$ trajectories during testing. The state evolution is simulated with an RK4-solver with timestep $\Delta t = 10^{-4}\,\text{s}$ over a time interval $\mathcal{I} = [0, 0.25]\text{s}$, resulting in $K = 2500$ samples. The trajectories contain recordings of the cloth falling on a sphere from a height of $0.13\,\text{m}$ in the center above the origin of the sphere. To vary scenarios, the radius of the sphere is randomly-sampled from $r \in [0.02, 0.12]\,\text{m}$.

#### F.4.2 Architectures & Training

The RO-LNN is composed of a constrained AE with layer sizes $n_l = [32, 64, 128, 600]$ for the $l = 1, ..., 4$ encoder and decoder layers $\rho_{\mathcal{Q}}^{(l)} : \mathbb{R}^{n_l} \to \mathbb{R}^{n_{l-1}}$ and $\varphi_{\mathcal{Q}}^{(l)} : \mathbb{R}^{n_{l-1}} \to \mathbb{R}^{n_l}$, and a latent space dimension $n_0 = 10$. The kinetic energy network of the latent geometric LNN consists of 2 hidden Euclidean layers of 64 neurons with SoftPlus activations, and an exponential map layer $\text{Exp}_{\mathbf{I}}$. The potential energy network also consists of 2 hidden Euclidean layers of 64 neurons with SoftPlus activations.

The featured model is trained on $H_{\text{train}} = 8$ integration steps via the loss (14) optimized using Riemannian Adam with a learning rate of $5 \times 10^{-2}$ for the AE parameters $\boldsymbol{\Xi}$, and $2 \times 10^{-4}$ for the LNN parameters $\boldsymbol{\theta}$, and a regularization $\gamma = 2 \times 10^{-5}$. THe model is trained for 3000 epochs.

## G Additional Experimental Results

### G.1 Learning Lagrangian Dynamics with Geometric LNNs

This section presents additional results on learning the dynamics of a 2-DoF pendulum with the geometric LNN presented in Section 3.

#### G.1.1 Additional Results on the Experiment of Section 5.1

**Trajectory prediction.** Fig. 8 complements Fig. 2-*middle, right* by depicting the long-term position and velocity predictions of the different geometric LNNs considered in Section 5.1. The velocity predictions follow the same trend as the positions: The geometric LNNs outperform the DeLaNs, the geometric LNNs trained via multi-step integration outperform those trained on acceleration data, and the geometric LNNs with SPD layers do not noticeably outperform those employing solely Euclidean and exponential map layers.

**Energy conservation.** Fig. 9 displays the total energy of the different LNNs of Figs. 2-8 along the predicted trajectory. As the pendulum is unactuated, the energy levels — that can be learned up to a constant — should be constant over the predicted trajectory. We observe that all LNNs approximately conserve the total predicted energy $\mathcal{E}$, with the DeLaNs showing the highest fluctuations. Note that, since the predicted trajectories are based on Euler forward integration, we do not expect the total energy of the system to be perfectly preserved.

#### G.1.2 Sine-Tracking 2-DoF Pendulum

We reproduce the experiment of Section 5.1 on a different dataset, namely on the sine-tracking 2-DoF pendulum dataset introduced in App. F.1.1.

Fig. 10 depicts the acceleration prediction errors for selected architectures. Similarly as for the unactuated dataset, the LNNs with Cholesky networks are outperformed by the geometric LNNs, and even more so in the low-data regime. In contrast to the unactuated pendulum dataset, the geometric

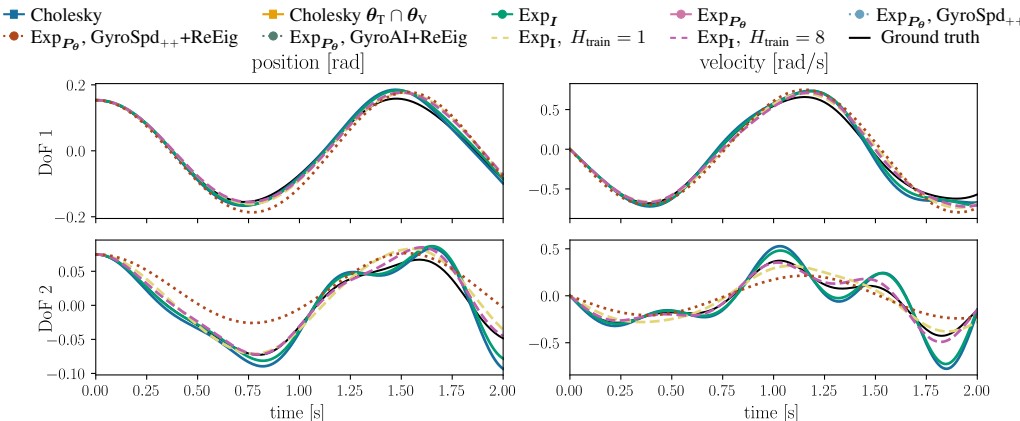

Figure 8: Numerically-integrated position and velocity predictions and ground truth trajectory (—) for LNNs trained on the unactuated 2-DoF pendulum dataset.

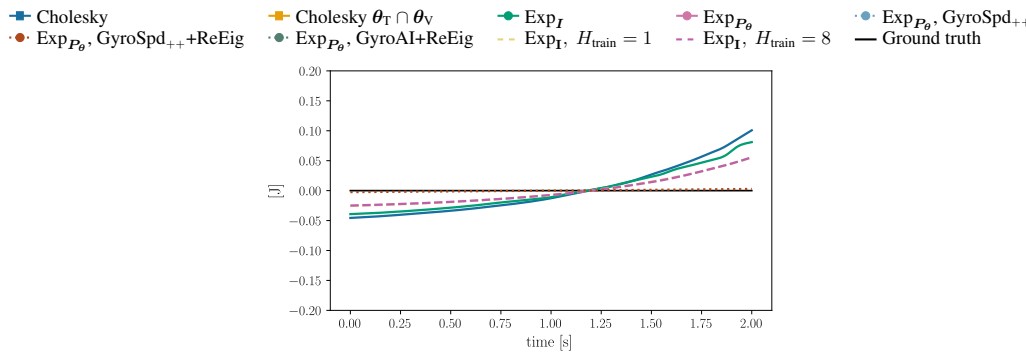

Figure 9: Total energy $\check{\mathcal{E}}$ (up to a constant) of different LNNs.

LNNs featuring SPD layers result in slightly lower acceleration errors. Fig. 11 depict long-term trajectory prediction obtained by integrating the state predictions given by the LNN. We observe that training the LNNs with multi-step integration is crucial to obtain accurate trajectory prediction for the sine-tracking dataset. Generally, the results on this second dataset are aligned with those presented in Section 5.1 and validate the effectiveness of considering the geometry of the mass-inertia matrix and of training the models via multi-step integration when learning Lagrangian dynamics.

### G.1.3 INFLUENCE OF SPD LAYERS

Next, we evaluate the influence of SPD layers within the kinetic energy network $M(q; \theta_{\mathrm{T}})$ of the geometric LNN. We consider the network architectures described in App. F.1.2 and train the models with acceleration data. We vary the type of SPD layers, as well as the number of hidden layers $L_{\mathrm{T},\mathbb{R}}$, $L_{\mathrm{T},\mathcal{S}_{++}^n}$ of the Euclidean and SPD components of the kinetic energy network. Each model is trained with two random seeds and tested over 10 unseen testing trajectories from the corresponding dataset. We report the lower prediction error across the two trained models. The average acceleration prediction errors for the unactuated and sine-tracking datasets are given in Tables 4 and 5, respectively.

Despite the increased performance of geometric LNNs over DeLaNs, the reported prediction errors do not allow us to observe any clear influence of the choice of SPD layers, exponential map base-points, or Euclidean and SPD layers depths across training dataset sizes. In particular, increasing the depth of SPD layers does not seem to increase the expressivity of the overall SPD network.

We hypothesize that the limited improvements obtained by adding the GyroSpd$_{++}$, GyroAI, and ReEig layers in our SPD network may be due to practical issues related to the training of these

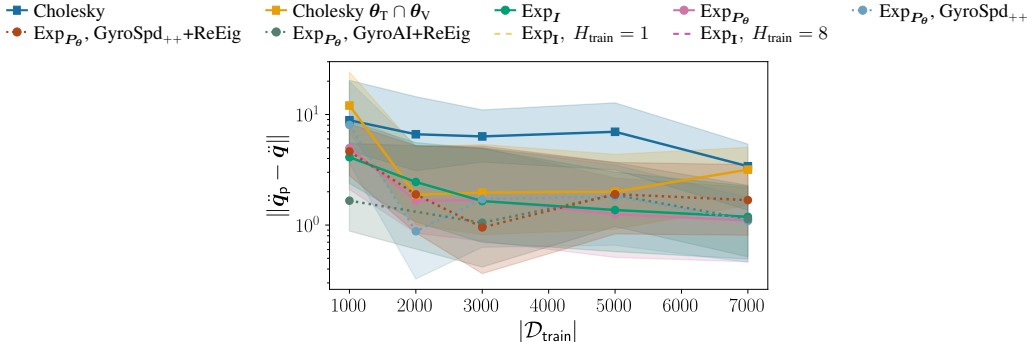

Figure 10: Median acceleration prediction error for different LNNs and training set sizes $\mathcal{D}_{\text{train}}$ over 10 testing trajectories. The models were trained on the sine-tracking 2-DoF pendulum dataset. Shaded regions represent first and third quartiles.

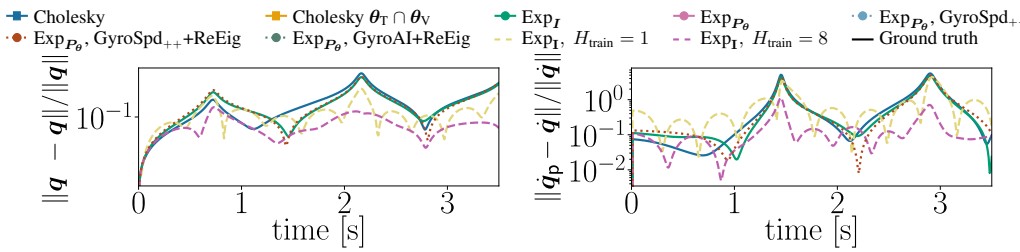

Figure 11: Relative errors of numerically-integrated position and velocity predictions w.r.t. ground truth trajectory over prediction horizon $H_{\text{test}}$ for LNNs trained on the sine-tracking 2-DoF pendulum dataset.

layers. Specifically, the abundance of matrix exponentials and logarithms within the SPD layers tends to cause numerical issues in the optimization procedure. These numerical instabilities also prevent the convergence of several models, as indicated by nan values in Tables 4 and 5. Notice that this mostly occurs for the SPD networks containing ReEig layers. Potential solutions include implementing analytic layer derivatives to replace unstable autodiff computations, or endowing the SPD manifold with a different Riemannian metric.

Due to the limited performance improvements and increased computational complexity (see App. G.1.5) of the SPD layers), we consider SPD networks $M(q; \theta_{\text{T}}) = (g_{\text{Exp}} \circ g_{\mathbb{R}})(q)$ composed of Euclidean and exponential map components when learning reduced-order Lagrangian dynamics in the experiments of Section 5.2.

Table 4: Mean and standard deviation of the acceleration prediction errors $\|\ddot{q}_{\text{p}} - \ddot{q}\|$ for geometric LNNs with different SPD network architectures trained on various training sizes on the unactuated 2-DoF pendulum dataset.

| | | | $|\mathcal{D}_{\text{train}}|$ | | | | |
|---|---|---|---|---|---|---|---|
| | | | 1000 | 2000 | 3000 | 5000 | 7000 |
| GyroSpd$_{++}$ | Exp$_I$ | $L_{\text{T},\mathbb{R}} = 1, L_{\text{T},\mathcal{S}_{++}^n} = 3$ | $5.95 \pm 5.69$ | $5.22 \pm 3.78$ | $1.83 \pm 2.61$ | $1.86 \pm 2.98$ | $\underline{0.92 \pm 1.31}$ |
| | | $L_{\text{T},\mathbb{R}} = L_{\text{T},\mathcal{S}_{++}^n} = 2$ | $15.7 \pm 19.5$ | $5.15 \pm 3.66$ | $4.43 \pm 2.96$ | $2.71 \pm 3.22$ | $1.52 \pm 1.69$ |
| | Exp$_{P_\theta}$ | $L_{\text{T},\mathbb{R}} = 1, L_{\text{T},\mathcal{S}_{++}^n} = 3$ | $\underline{1.78 \pm 2.30}$ | $4.48 \pm 2.70$ | $4.45 \pm 2.96$ | $1.27 \pm 1.47$ | $1.75 \pm 2.32$ |
| | | $L_{\text{T},\mathbb{R}} = L_{\text{T},\mathcal{S}_{++}^n} = 2$ | $4.37 \pm 2.65$ | $5.34 \pm 3.72$ | $2.08 \pm 3.02$ | $\mathbf{0.89 \pm 1.11}$ | $1.44 \pm 2.02$ |
| GyroSpd$_{++}$+ReEig | Exp$_I$ | $L_{\text{T},\mathbb{R}} = 1, L_{\text{T},\mathcal{S}_{++}^n} = 3$ | $4.66 \pm 2.99$ | $1.75 \pm 2.33$ | $\underline{1.77 \pm 2.40}$ | $1.66 \pm 2.58$ | $1.10 \pm 1.42$ |
| | | $L_{\text{T},\mathbb{R}} = L_{\text{T},\mathcal{S}_{++}^n} = 2$ | $4.66 \pm 2.81$ | $1.72 \pm 2.19$ | $1.80 \pm 2.52$ | $1.78 \pm 2.21$ | $\mathbf{0.84e \pm 1.25}$ |
| | Exp$_{P_\theta}$ | $L_{\text{T},\mathbb{R}} = 1, L_{\text{T},\mathcal{S}_{++}^n} = 3$ | $4.60 \pm 2.87$ | $4.58 \pm 2.93$ | $4.38 \pm 2.85$ | $1.43 \pm 1.89$ | $1.18 \pm 1.93$ |
| | | $L_{\text{T},\mathbb{R}} = L_{\text{T},\mathcal{S}_{++}^n} = 2$ | $1.99 \pm 2.23$ | $1.78 \pm 2.38$ | $1.79 \pm 2.52$ | $\underline{1.20 \pm 1.77}$ | $1.14 \pm 1.31$ |
| GyroAI+ReEig | Exp$_I$ | $L_{\text{T},\mathbb{R}} = 1, L_{\text{T},\mathcal{S}_{++}^n} = 3$ | $2.31 \pm 2.33$ | $7.28 \pm 8.43$ | $\mathbf{1.52 \pm 2.08}$ | $1.32 \pm 1.91$ | $1.13 \pm 1.48$ |
| | | $L_{\text{T},\mathbb{R}} = L_{\text{T},\mathcal{S}_{++}^n} = 2$ | $\mathbf{1.77 \pm 2.34}$ | nan | $1.99 \pm 2.70$ | nan | $1.30 \pm 2.17$ |
| | Exp$_{P_\theta}$ | $L_{\text{T},\mathbb{R}} = 1, L_{\text{T},\mathcal{S}_{++}^n} = 3$ | $2.76 \pm 2.49$ | $\underline{1.71 \pm 2.29}$ | $21.2 \pm 19.5$ | $1.26 \pm 1.66$ | $1.07 \pm 1.48$ |
| | | $L_{\text{T},\mathbb{R}} = L_{\text{T},\mathcal{S}_{++}^n} = 2$ | nan | $\mathbf{1.57 \pm 2.01}$ | nan | $1.82 \pm 2.48$ | $4.22 \pm 2.60$ |

Table 5: Mean and standard deviation of the acceleration prediction errors $\|\ddot{q}_p - \ddot{q}\|$ for different geometric LNNs with different SPD network architectures trained on various training sizes on the sine-tracking 2-DoF pendulum dataset.

| | | | $|\mathcal{D}_{\text{train}}|$ | | | | |
|---|---|---|---|---|---|---|---|
| | | | 1000 | 2000 | 3000 | 5000 | 7000 |
| GyroSpd$_{++}$ | Exp$_I$ | $L_{\text{T},\mathbb{R}}=1, L_{\text{T},\mathcal{S}^n_{++}}=3$ | $\mathbf{3.73 \pm 4.46}$ | $4.22 \pm 4.42$ | $3.95 \pm 4.43$ | $3.37 \pm 3.54$ | $2.05 \pm 2.41$ |
| | | $L_{\text{T},\mathbb{R}}=L_{\text{T},\mathcal{S}^n_{++}}=2$ | $10.9 \pm 8.62$ | $3.95 \pm 4.50$ | $3.89 \pm 4.49$ | $2.80 \pm 2.66$ | $2.18 \pm 2.29$ |
| | Exp$_{P_\theta}$ | $L_{\text{T},\mathbb{R}}=1, L_{\text{T},\mathcal{S}^n_{++}}=3$ | $12.0 \pm 13.6$ | $\underline{3.58 \pm 4.58}$ | $3.70 \pm 4.59$ | $3.73 \pm 3.95$ | nan |
| | | $L_{\text{T},\mathbb{R}}=L_{\text{T},\mathcal{S}^n_{++}}=2$ | $17.0 \pm 21.7$ | $\mathbf{3.44 \pm 4.84}$ | $3.66 \pm 4.40$ | $2.75 \pm 2.80$ | $\underline{1.49 \pm 1.47}$ |
| GyroSpd$_{++}$+ReEig | Exp$_I$ | $L_{\text{T},\mathbb{R}}=1, L_{\text{T},\mathcal{S}^n_{++}}=3$ | $4.59 \pm 4.58$ | $4.17 \pm 4.43$ | $4.15 \pm 4.48$ | $71.0 \pm 79.6$ | $5.80 \pm 5.23$ |
| | | $L_{\text{T},\mathbb{R}}=L_{\text{T},\mathcal{S}^n_{++}}=2$ | $13.5 \pm 15.3$ | $4.05 \pm 4.26$ | $3.95 \pm 4.38$ | $5.78 \pm 4.91$ | $\mathbf{1.27 \pm 1.25}$ |
| | Exp$_{P_\theta}$ | $L_{\text{T},\mathbb{R}}=1, L_{\text{T},\mathcal{S}^n_{++}}=3$ | $4.29 \pm 4.53$ | $4.55 \pm 4.47$ | $4.13 \pm 4.43$ | $\mathbf{1.88 \pm 1.74}$ | $3.34 \pm 3.65$ |
| | | $L_{\text{T},\mathbb{R}}=L_{\text{T},\mathcal{S}^n_{++}}=2$ | $6.13 \pm 4.61$ | $3.90 \pm 4.42$ | $3.45 \pm 4.81$ | $2.95 \pm 3.17$ | $2.64 \pm 2.82$ |
| GyroAI+ReEig | Exp$_I$ | $L_{\text{T},\mathbb{R}}=1, L_{\text{T},\mathcal{S}^n_{++}}=3$ | $6.39 \pm 5.73$ | $4.00 \pm 4.45$ | $\mathbf{3.33 \pm 4.30}$ | nan | $6.74 \pm 5.59$ |
| | | $L_{\text{T},\mathbb{R}}=L_{\text{T},\mathcal{S}^n_{++}}=2$ | $9.14 \pm 9.73$ | $3.79 \pm 4.41$ | $3.67 \pm 4.60$ | $2.36 \pm 2.58$ | $49.1 \pm 58.2$ |
| | Exp$_{P_\theta}$ | $L_{\text{T},\mathbb{R}}=1, L_{\text{T},\mathcal{S}^n_{++}}=3$ | $4.02 \pm 4.66$ | nan | $4.11 \pm 4.34$ | $\underline{2.35 \pm 2.08}$ | $1.86 \pm 2.13$ |
| | | $L_{\text{T},\mathbb{R}}=L_{\text{T},\mathcal{S}^n_{++}}=2$ | $3.99 \pm 4.66$ | nan | $\underline{3.40 \pm 4.55}$ | $2.78 \pm 2.68$ | $1.68 \pm 1.75$ |

Table 6: Mean and standard deviation of the $H_{\text{test}} = 25$-steps velocity prediction errors for a geometric LNN trained via multi-step integration with different horizons $H_{\text{train}}$ and datasets.

| Dataset | $H_{\text{train}} = 1$ | $H_{\text{train}} = 4$ | $H_{\text{train}} = 8$ | $H_{\text{train}} = 12$ |
|---|---|---|---|---|
| Unactuated | $4.61 \times 10^{-1} \pm 6.83 \times 10^{-1}$ | $6.64 \times 10^{-2} \pm 1.73 \times 10^{-1}$ | $4.80 \times 10^{-2} \pm 1.17 \times 10^{-1}$ | $4.80 \times 10^{-2} \pm 1.13 \times 10^{-1}$ |
| Sine-tracking | $2.32 \times 10^{-2} \pm 4.07 \times 10^{-2}$ | $8.87 \times 10^{-1} \pm 2.05$ | $1.25 \times 10^{-2} \pm 1.89 \times 10^{-2}$ | $1.36 \times 10^{-2} \pm 2.22 \times 10^{-2}$ |

### G.1.4 INFLUENCE OF MULTI-STEP INTEGRATION DURING TRAINING

We assess the influence of the number $H_{\text{train}}$ of steps considered when training our geometric LNNs. The featured networks are trained as described in F.1.2. For this experiment, the architectures are as described for multi-step integration in section F.1.2. Table 6 reports the velocity prediction errors averaged over 10 testing trajectories for models trained on different prediction horizons $H_{\text{train}}$ for each of the 2-DoF pendulum datasets. We generally notice a significant decrease in the average error for geometric LNNs trained on longer horizons $H_{\text{train}}$. Ultimately, we select $H_{\text{train}}$ by trading off the increase in performance and the training time. The later is investigated next.

### G.1.5 RUNTIMES

Table 7 shows the average training times of several LNNs trained on datasets of $|\mathcal{D}_{\text{train}}| = 1000$ datapoints for 1000 epochs. The potential energy network and Euclidean part of the kinetic energy network of all models consist of 2 hidden Euclidean layers of 64 neurons. For SPD layers, we only consider GyroSpd$_{++}$+ReEig layers as they are the computationally most demanding SPD layer combination. The featured model consists of $L_{\text{T},\mathcal{S}^n_{++}} = 2$ of these layers.

Importantly, the well-performing geometric LNN with identity basepoint and $L_{\text{T},\mathcal{S}^n_{++}} = 0$ trains almost as fast as DeLaN with its Cholesky layers. Instead, integrating GyroSpd$_{++}$+ReEig layers within the SPD network doubles the training time. Note that similar increases were observed for other combinations of SPD layers. As expected, training the LNNs with multi-step integration increases the training time. However, such training leads to increased data-efficiency and improved long-term predictions.

During evaluation, running the well-performing LNN with 2 Euclidean layers and the Exp$_I$ layer in combination with numerical Euler integration over 25 steps, takes 0.01191 secs, which could enable certain online applications.

Table 7: Training times of various LNNs in seconds.

| DeLaN | Geometric LNN | | | |
|---|---|---|---|---|
| Cholesky | Exp$_I$ | Exp$_{P_\theta}$ | Exp$_{P_\theta}$, GyroSpd$_{++}$+ReEig | Exp$_I$, $H_{\text{train}} = 8$ |
| 88 | 92 | 121 | 189 | 105 |

Table 8: Mean and standard deviations of the reconstructed and latent prediction errors for different prediction horizons with RO-LNNs trained on acceleration and via multi-step integration.

| | | $\sum_{j=1}^{H} \|\tilde{q}_{\mathrm{p}}(t_j) - q(t_j)\|^2$ | $\ell_{\mathrm{LNN},n}$ | $\sum \ell_{\mathrm{LNN},d}$ |
|---|---|---|---|---|
| $H_{\mathrm{test}} = 8$ | acc | $2.65 \times 10^{-1} \pm 3.13 \times 10^{-1}$ | $1.95 \times 10^{0} \pm 2.14 \times 10^{0}$ | $1.70 \times 10^{-3} \pm 4.94 \times 10^{-2}$ |
| | ODE | $\mathbf{1.86 \times 10^{-4} \pm 2.57 \times 10^{-4}}$ | $\mathbf{4.06 \times 10^{-2} \pm 8.70 \times 10^{-2}}$ | $\mathbf{9.22^{-4} \pm 3.76 \times 10^{-3}}$ |
| $H_{\mathrm{test}} = 25$ | acc | $2.64 \times 10^{-1} \pm 3.12 \times 10^{-1}$ | $2.25 \times 10^{0} \pm 2.50 \times 10^{0}$ | $1.45 \times 10^{-2} \pm 4.28 \times 10^{-2}$ |
| | ODE | $\mathbf{2.05 \times 10^{-4} \pm 2.74 \times 10^{-4}}$ | $\mathbf{1.66 \times 10^{-1} \pm 4.85 \times 10^{-1}}$ | $\mathbf{7.78 \times 10^{-3} \pm 3.25 \times 10^{-2}}$ |
| $H_{\mathrm{test}} = 50$ | acc | $2.64 \times 10^{-1} \pm 3.10 \times 10^{-1}$ | $3.26 \times 10^{0} \pm 4.62 \times 10^{0}$ | $5.31 \times 10^{-2} \pm 1.753 \times 10^{-1}$ |
| | ODE | $\mathbf{4.03 \times 10^{-4} \pm 8.62 \times 10^{-4}}$ | $\mathbf{5.41 \times 10^{-1} \pm 1.70 \times 10^{0}}$ | $\mathbf{2.76 \times 10^{-2} \pm 1.16 \times 10^{-1}}$ |

## G.2 ADDITIONAL RESULTS ON THE 16-DoF COUPLED PENDULUM EXPERIMENT OF SECTION 5.2.1

This section presents additional results on learning the high-dimensional dynamics of a 16-DoF pendulum with geometric RO-LNN.

### G.2.1 ACCELERATION VS MULTI-STEP TRAINING

Here, we further evaluate the impact of the losses used to train geometric RO-LNNs. Table 8 shows the average acceleration errors of trajectory predictions of RO-LNNs trained on accelerations or via multi-step integration with horizon $H_{\mathrm{train}} = 8$. We observe that the RO-LNN trained via multi-step integration consistently outperforms the RO-LNN trained on accelerations for all prediction horizons $H_{\mathrm{test}}$. This validates the effectiveness of considering successive states during training to learn the dynamics of physical systems. Notice that training on acceleration data requires computing the second derivatives in the AE layers, resulting in large-scale matrix multiplications, which not only leads to longer training times, but depending on dimensionality of the system, requires vast amounts of memory. In this sense, training via multi-step integration also avoids the computation of these Hessian matrices.

### G.2.2 CONSTRAINED VS STANDARD AUTOENCODERS

We further investigate the effectiveness of our geometric constrained AE whose parameters are obtained via Riemannian optimization compared to two baselines, namely *(1)* constrained AE optimized via the overparametrization of (Otto et al., 2023), and *(2)* a regular (unconstrained) AE. All AEs are trained to reduce a set of position data $\mathcal{D}_{\mathrm{train}} = \{q_i\}_{i=1}^{N}$ from 15 trajectories of the 16-DoF pendulum to a $d = 4$-dimensional latent space. The constrained AEs are implemented as in Section 5.2.1 and described in App. F.2.2. The regular AE consists of 4 Euclidean layers with ReLu activations of sizes $n_l = [64, 64, 64, 16]$ for $l = 1, ..., 4$ encoder and decoder layers $\rho_{\mathcal{Q}}^{(l)} : \mathbb{R}^{n_l} \to \mathbb{R}^{n_{l-1}}$ and $\varphi_{\mathcal{Q}}^{(l)} : \mathbb{R}^{n_{l-1}} \to \mathbb{R}^{n_l}$. Our constrained AE is trained using Riemannian Adam with learning rate $5 \times 10^{-2}$. The overparametrized and regular AEs are trained using Adam with learning rate $5 \times 10^{-2}$ for the overparametrized, and $1e-2$ for the regular AE. All models are trained for 2500 epochs.

Fig. 12 shows the reconstruction errors of the AEs for different sizes. We observe that the constrained AEs consistently outperform the regular one. Moreover, the Riemannian optimization leads to lower reconstruction errors than the overparametrization, especially in low-data regimes.

### G.2.3 COMPUTATIONAL SPEEDUP OF ROM W.R.T. FOM

Despite that we consider scenarios where the FOM is unknown and that trainings of a full-order LNN were unsuccessful (see Section 5.2.1), we aim at providing an idea of the computational effort of our model compared to the evaluation of the FOMs. To do so, we symbolically derive the Lagrangian equations of motion with known physical quantities of the considered 16-DoF pendulum from Section F.2.1 to obtain a FOM. We compare the wall-clock time of evaluation of this FOM against the ODE-trained RO-LNN. For both models, we consider a roll-out of 3000 timesteps from the same initial conditions $q(t = 0\,\mathrm{s})$ and $\dot{q}(t = 0\,\mathrm{s})$ via Euler forward integration with a timestep $\Delta t = 10^{-3}\,\mathrm{s}$. Averaged over 10 runs on the same local CPU, we achieve evaluation times of $113.59\,\mathrm{s}$ for the FOM, and $1.57\,\mathrm{s}$ for the ROM. We conclude that the ROMs obtained with our proposed method not only enable structure-preserving learning of high-dimensional system dynamics, but also significantly improve computational efficiency compared to evaluating FOMs.

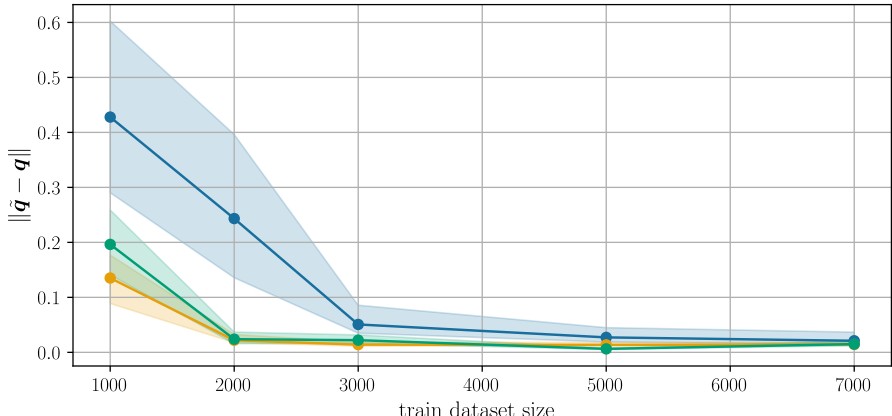

Figure 12: Median and quartiles of position reconstruction error for the regular AE (—), the biorthogonal constrained AE (—), and the overparametrized constrained AE (—) for RO-LNN trained on different sizes of training sets.

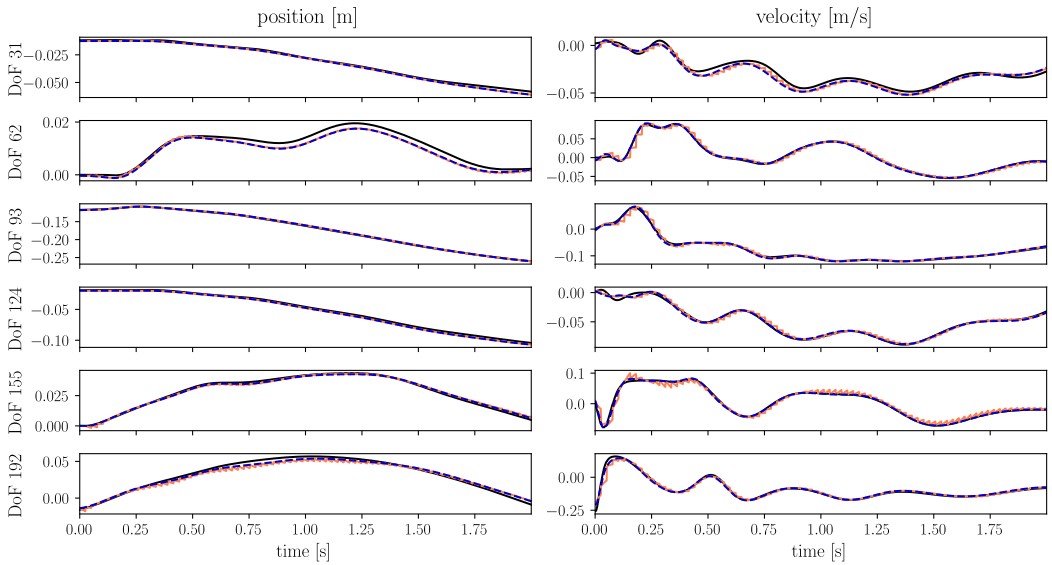

Figure 13: Rope position and velocity predictions (—) from the RO-LNNs trained via multi-step integration and ground truth (—). The corresponding AE reconstructions (····) are depicted for completeness. The model is updated with a new initial condition $(\boldsymbol{q}_0, \dot{\boldsymbol{q}}_0)$ every $0.025\,\mathrm{s}$ ($H_{\text{test}} = 25$).

## G.3  ADDITIONAL RESULTS ON THE 192-DOF ROPE EXPERIMENT OF SECTION 5.2.2

Fig. 13 shows the AE reconstruction and the RO-LNN predictions ($H_{\text{test}} = 25$) for selected DoFs of a test trajectory. We observe that the the RO-LNN accurately models the high-dimensional dynamics of the rope.

## G.4  ADDITIONAL RESULTS ON THE 600-DOF CLOTH EXPERIMENT OF SECTION 5.2.3

Table 9 shows the reconstruction and prediction errors of our model averaged over 10 testing trajectories. Moreover, Figs. 14 and 15 show the AE reconstruction and the predictions of the RO-LNN of selected DoFs of a test trajectory for horizons $H_{\text{test}} = 25$ and $H_{\text{test}} = 3000$, respectively. Notice that the ground truth only contains actuation data for the first $0.25\,\mathrm{s}$ (i.e., 2500 timesteps). Therefore, we

Table 9: Cloth reconstruction and $H_{\text{test}} = 25$-steps prediction errors over 10 testing trajectories for the RO-LNN with $d = 10$ trained via multi-step integration with the loss (14).

| $\|\bar{\boldsymbol{q}} - \boldsymbol{q}\|^2$ | $\|\dot{\bar{\boldsymbol{q}}} - \dot{\boldsymbol{q}}\|^2$ | $\ell_{\text{LNN},n}$ | $\ell_{\text{LNN},d}$ |
|---|---|---|---|
| $2.47 \times 10^{-3} \pm 1.15 \times 10^{-3}$ | $1.50 \times 10^{0} \pm 1.81 \times 10^{0}$ | $1.51 \times 10^{0} \pm 1.81 \times 10^{0}$ | $5.07 \times 10^{-4} \pm 1.09 \times 10^{-3}$ |

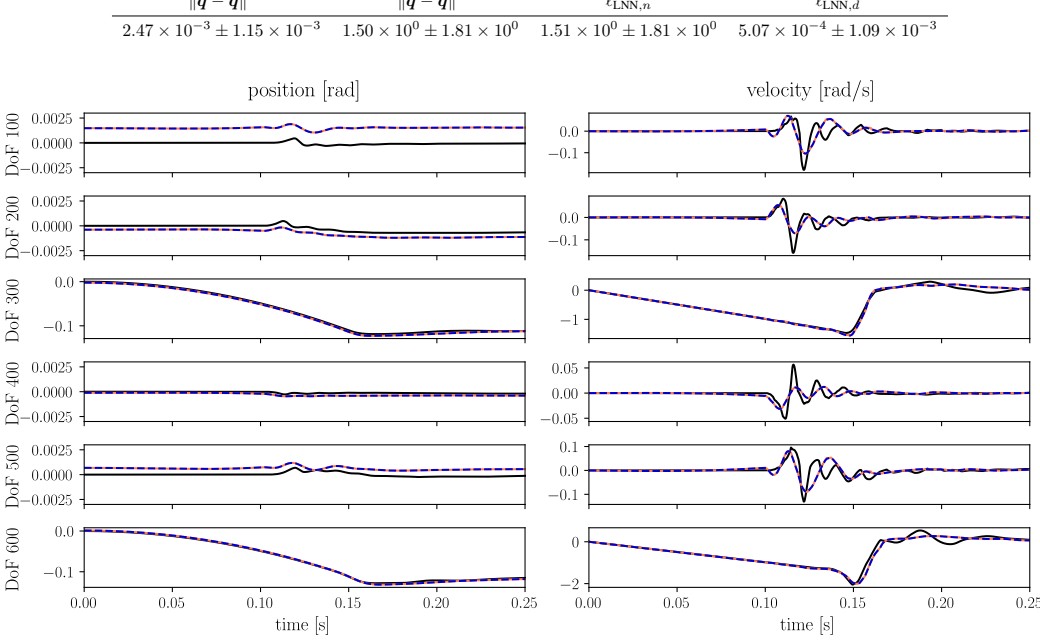

Figure 14: Cloth position and velocity predictions (—) from the RO-LNNs trained via multi-step integration and ground truth (—). The corresponding AE reconstructions (⋯) are depicted for completeness. The model is updated with a new initial condition $(\boldsymbol{q}_0, \dot{\boldsymbol{q}}_0)$ every $0.025\,\text{s}$ ($H_{\text{test}} = 25$).

reasonably assume that the cloth remains still for the additional time horizon $t_{\text{aug}} = [0.25, 0.30]\,\text{s}$ and set the ground truth states and torques as equal to their last recorded values. Fig. 16 displays the predicted latent energy to be compared with the groud-truth energy projected in the latent space. Overall, our results demonstrate the ability of our RO-LNN to infer long-term predictions of complex high-dimensional deformable systems.

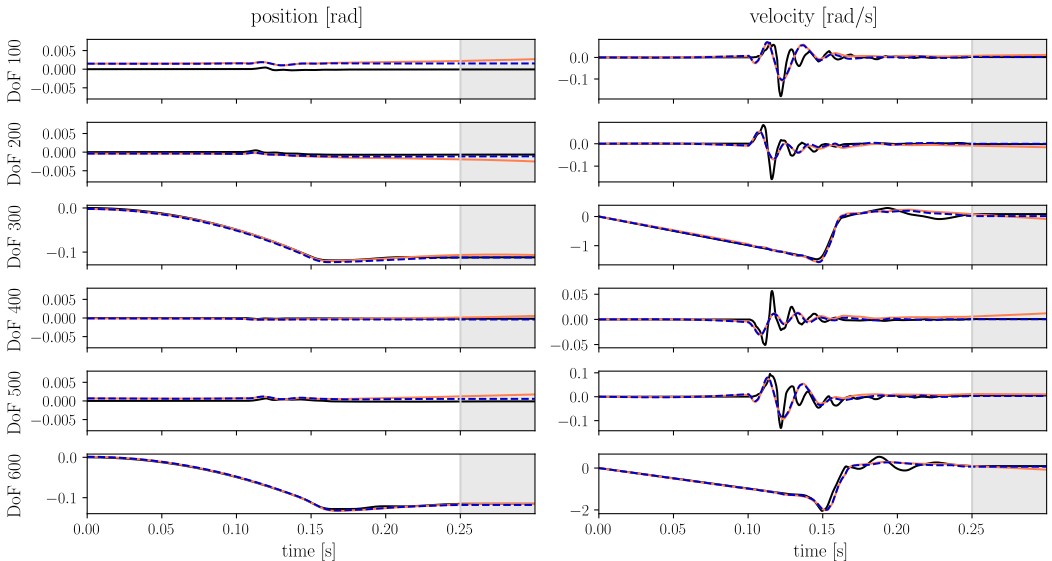

Figure 15: Cloth position and velocity full-horizon predictions (—) from the RO-LNNs trained via multi-step integration and ground truth (—). The corresponding AE reconstructions (····) are depicted for completeness. The dynamics are predicted from a given initial condition $(\boldsymbol{q}_0, \dot{\boldsymbol{q}}_0)$ for $0.3\,\mathrm{s}$ ($H_{\text{test}} = 3000$). The grey-shaded areas indicate the interval during which ground-truth data are extrapolated from the last observation.

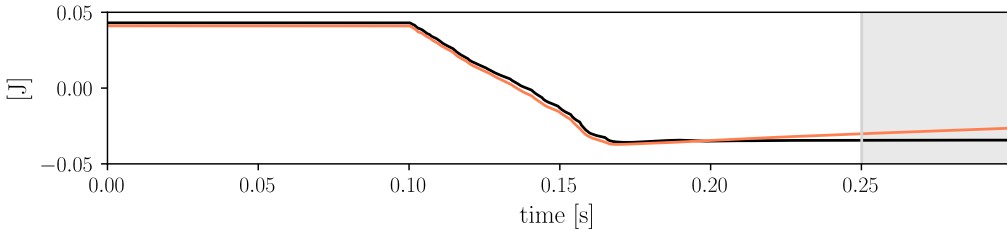

Figure 16: Predicted (—) and ground truth (—) latent energy $\check{\mathcal{E}}$ (up to a constant) over time. The grey-shaded areas indicate the interval during which ground-truth data are extrapolated from the last observation.

