# OpenReview forum: "A Riemannian Framework for Learning Reduced-order Lagrangian Dynamics"
_ICLR.cc/2025/Conference — ICLR 2025 Poster_

### Official Review · Reviewer_wYEc · 2024-10-30

**Soundness:** 3
**Presentation:** 2
**Contribution:** 2
**Rating:** 6
**Confidence:** 2

**Summary:**

The paper proposes a reduced-order learning model with the preservation of the state geometry for high-dimensional Lagrangian systems.

**Strengths:**

The author proposes a dimensionality reduction framework with physical consistency and geometry preservation. The writing is clear and the experiment demonstrates the effectiveness of the model.

**Weaknesses:**

1.	In the Introduction, for readers unfamiliar with the high-dimensional Lagrangian system, the author should provide some references and emphasize their importance. The author is also suggested to provide some quantitative descriptions, e.g., the range of the dimensionality.

2.	Is preserving the geometry important between FOM and ROM? Can the author summarize what the benefits are of preserving the structure in physical systems and control?

3.	What’s the difference between Eq. (3) and Eq. (4)? For example, in Eq. (3), you state $X|\_{\gamma(t)}\in \mathcal{T}\_{\gamma(t)}\mathcal{T}\mathcal{Q} $. In Eq. (4), however, the set is $\mathcal{T}_{\gamma(t)}\mathcal{Q}$. Why do smooth trajectory lie in $\mathcal{Q}$ rather than $\mathcal{T}\mathcal{Q}$, where is $\dot{q}$?

4.	What is $V(q;\theta_V)$ in Line 162? Since there are massive notations, the author is advised to complete the proofreading.

5.	It’s unclear of the transition between Section 2.3 and 2.4. For example, to fix what problems in Section 2.3, should we introduce LNN in Section 2.4?

6.	The advantage of the proposed SPD network should be better illustrated. The authors claim that their model makes use of Riemannian geometry. However, it seems that other methods can also maintain this constraint. For example, Cholesky decomposition can also make sure the output belongs to SPD space.

7.	Notations in Eq. (8) are unclear to the reader. What are $l$, $h$, $x$, and PT? Can the author align them with the previous notations?

8.	It seems weird to state that computing the second-order derivative in Eq. (13) is expensive, but computing the integration of multi-step in Eq. (14) is cheap.

9.	The legends in Fig. 2 are hard to align with the curves. Please refine this figure.

10.	Why is the error in Table 1 very high? I am not sure if this is a validated result.

**Questions:**

Q1. Please refine the notation system in the paper. You may remove some unnecessary notations and descriptions. See my Weakness points 3, 4, and 7.

Q2. The background and design motivation should be emphasized for readers unfamiliar with Lagrangian systems. See my Weakness points 1, 2.

Q3. The writing logic should be improved. See my Weakness points 5, 6.

Q4. The computational cost should be illustrated and reported. See my Weakness point 8.

Q5. The result presentation can be improved. See my Weakness points 9 and 10.

---

> ### Author Response · Authors · 2024-11-18
> **Response to Reviewer wYEc (1)**
>
> Thank you very much for your detailed review! We are delighted that you found that our “experiments demonstrates the effectiveness of the model” and that our “writing is clear”. Please see our answers to your comments below.
>
> **Questions:**
>
> **Q1.**
> > **Please refine the notation system in the paper. You may remove some unnecessary notations and descriptions. See my Weakness points 3, 4, and 7.**
> >
> >3. What’s the difference between Eq. (3) and Eq. (4)? For example, in Eq. (3), you state $X_{\gamma(t)} \in \mathcal T_{\gamma(t)}\mathcal T \mathcal Q$. In Eq. (4), however, the set is $\mathcal T_{\gamma(t)}\mathcal Q$. Why do smooth trajectory lie in $\mathcal Q$ rather than $\mathcal T \mathcal Q$, where is $\dot q$?
> >
> >4. What is $V(q;\theta_V)$ in Line 162? Since there are massive notations, the author is advised to complete the proofreading.
> >
> >7. Notations in Eq. (8) are unclear to the reader. What are $l, h, x$, and PT? Can the author align them with the previous notations?
>
> Thank you for pointing out to unclear notations. We answer the individual points below.
>
> 3. We understand the confusion of using the same letters for both equations and updated them to denote the general configuration manifold as $\mathcal{M}$ in Section 2.2, and refer to its instance **in the Lagrangian case** as $\mathcal T\mathcal Q$, i.e., $\mathcal{M}=\mathcal{TQ}$. We clarified this point in Section 2.2. Also, we removed a previous equation to save space, so the indexing shifted by one number up.
>
>    Eq. 4 (now 3) refers to the standard problem in MOR, where we aim at reducing a first-order dynamical system that lives on a configuration manifold $\mathcal Q$ (now  $\mathcal M$), so that it generates trajectories $\mathbf\gamma(t) \in \mathcal Q$ (now $\mathbf\gamma(t) \in \mathcal M$). Its state evolution is described by a vector field $\mathbf{X}$ that lies on the tangent space, i.e., $\dot{\mathbf{\gamma}}(t) = \mathbf{X} \in \mathcal T_{\mathbf{\gamma}(t)} \mathcal Q$ (or now: $ \in \mathcal T_{\mathbf{\gamma}(t)} \mathcal M$)).
>
>     Eq. 3 (now 2) describes the Lagrangian system, which is characterized by second-order dynamics, rewritten in a first-order form. In this case, a trajectory is part of the tangent bundle (being a smooth manifold representing the state-space) $\mathbf{\gamma}(t) = \left[ \mathbf{q}^{\intercal}, \dot{\mathbf{q}}^{\intercal}\right]^{\intercal}\in \mathcal T\mathcal Q$ . Its time derivative (i.e. the vector field) $\dot{\mathbf{\gamma}}(t) = \left[ \dot{\mathbf{q}}^{\intercal}, \ddot{\mathbf{q}}^{\intercal}\right]^{\intercal}\in \mathcal T_{\mathbf{\gamma}(t)}\mathcal T\mathcal Q$ is part of the tangent space of the tangent bundle at the point $\mathbf{\gamma (t)}$.
>
>     Eq. 4 (now 3) is equivalent to Eq. 3 (now 2) by setting the configuration space of the system in equation 4, $\mathcal Q$ (now equation 3 and $\mathcal M$,), to be the tangent bundle from equation 3 (now 2), i.e. $\mathcal T\mathcal Q$.
>
>
> 4.  $V(q;\theta_V)$ is the potential energy of a Lagrangian function $\mathcal L$ (see line 103) parametrized as a network $V(q;\theta_V)$.
>
> 7. In this section, we denote the loss function generally by $\ell$ and the optimization variable by $\mathbf{x}$. $h$ is a function of the Riemannian gradient, which depends on the selected optimization method. Finally $\mathrm{PT}$ denotes a parallel transport operation, which is described in Appendix A.1. We thank the reviewer for pointing this out as missing and completed Section 3.2 accordingly.
>
> **Q2.**
> > **The background and design motivation should be emphasized for readers unfamiliar with Lagrangian systems. See my Weakness points 1, 2.**
> >
> >1. In the Introduction, for readers unfamiliar with the high-dimensional Lagrangian system, the author should provide some references and emphasize their importance. The author is also suggested to provide some quantitative descriptions, e.g., the range of the dimensionality.
> >
> >2. Is preserving the geometry important between FOM and ROM? Can the author summarize what the benefits are of preserving the structure in physical systems and control?
>
> 1. Thank you for pointing this out! We emphasized the importance of high-dimensional Lagrangian systems for fluid flows, continuum mechanics, and robotics in Section 1. Moreover, we now explicitly stated that gray-box approaches, such as LNNs and HNNs are mostly limited to systems with 5 or less dimensions.
> 2.  Preserving the geometry between FOM and ROM is crucial for the ROM to preserve the physical properties of the FOM. For instance, structure-preserving ROMs preserve the energy-conservation and stability properties of the corresponding FOM. This may be leveraged, e.g., for energy-based control of FOMs via the ROMs.

---

> > ### Author Response · Authors · 2024-11-18
> > **Response to Reviewer wYEc (2)**
> >
> > **Q3.**
> > > **The writing logic should be improved. See my Weakness points 5, 6.**
> > >
> > >5. It’s unclear of the transition between Section 2.3 and 2.4. For example, to fix what problems in Section 2.3, should we introduce LNN in Section 2.4?
> > >
> > >6. The advantage of the proposed SPD network should be better illustrated. The authors claim that their model makes use of Riemannian geometry. However, it seems that other methods can also maintain this constraint. For example, Cholesky decomposition can also make sure the output belongs to SPD space.
> > >
> > 5. Thanks for pointing this out! We added a transition sentence.
> > 6. Cholesky-decomposition-based networks $\mathbf{L}(\mathbf{q};\theta)$  were used in prior works [1] to parametrize the mass-inertia matrix, so that $\mathbf{M}=\mathbf{L}\mathbf{L}^T$.  As pointed out by the reviewer, this ensures that the mass-inertia matrix is symmetric positive-definite (SPD). However, it defines the distances between two SPD matrices based on the Euclidean distance between their Cholesky decompositions and it is well known that this distance suffers from the problematic swelling effect [2,3], where the volume of the SPD matrices grows significantly while interpolating between two matrices of identical volumes. Therefore, mass-inertia matrices inferred via a Cholesky network will suffer from this swelling effect and lead to erroneous predictions of the dynamics. The solution is to equip the SPD manifold with the affine-invariant or Log-Euclidean metrics, which avoid the swelling effect, and directly parametrize the mass-inertia matrix with a SPD network $\mathbf{M}(\mathbf{q};\theta)$, as presented in Section 3.1. As shown in Section 5.1, this leads to improved performances compared to parametrizing the mass-inertia matrix via a Cholesky decomposition. We clarified these points in Appendix B.
> >
> > **Q4.**
> > >**The computational cost should be illustrated and reported. See my Weakness point 8.**
> > >
> > >8. It seems weird to state that computing the second-order derivative in Eq. (13) is expensive, but computing the integration of multi-step in Eq. (14) is cheap.
> > Note that equation (14) is now equation (13), and equation (13) is now (12):
> > We would like to clarify that the multi-step integration in Eq. (13) is not a “per se cheap” procedure. However, as mentioned in section 4.3, the Hessians that are required for the 2nd derivatives of the AE mappings in Eq. (12) involve the high-dimensional full-order space (which is, e.g., 600-dimensional for the cloth experiment). Such Hessians are computationally expensive to compute and very large to store, leading to a general trend in avoiding to compute such Hessians in machine learning algorithms. It is also worth noting that the multi-step integration of Eq. (13) takes place only in the lower-dimensional space, which reduces its cost. Moreover, the ablations presented in Section 5.1 on the double pendulum tend to show that the multi-step loss of Eq. (13) outperforms the acceleration loss of Eq. (12). We attribute this to the additional inductive bias given by predicting multiple steps during training. We clarify this in Section 4.3.
> >
> > **Q5.**
> > >**The result presentation can be improved. See my Weakness points 9 and 10.**
> > >
> > >9. The legends in Fig. 2 are hard to align with the curves. Please refine this figure.
> > >
> > >10. Why is the error in Table 1 very high? I am not sure if this is a validated result.
> >
> > 9. We uploaded a new version of Figure 2 depicting the relative prediction errors in position and velocity (and also adapted the format accordingly in Fig.11 for the sine-tracking example). We hope that this makes this figure more easily readable.
> > 10. Thank you for pointing this out. Table 1 provided unnormalized results on the testing data and reported the mean of the squared distances between predictions and ground truth values over the whole testing dataset of N samples, i.e. $\frac{1}{n}\sum_{i=1}^N||\tilde{\ddot{\mathbf{q}}}^{\text{p,i}} - \ddot{\mathbf{q}}^i||^2$. For large absolute acceleration values, high-dimensional (16 dimensions) vectors $\ddot{\mathbf{q}}^i$, and an inaccurate model, we indeed observed high absolute values, which were hard to interpret.
> >
> >     We updated Table 1 to report relative errors in acceleration $||\tilde{\ddot{\mathbf{q}}}^{\text{p,i}} - \ddot{\mathbf{q}}^{i}|| / ||\ddot{\mathbf{q}}^i||$, velocity $||\tilde{\dot{\mathbf{q}}}^{\text{p,i}} - \dot{\mathbf{q}}^i|| / ||\dot{\mathbf{q}}^{i}||$, and position $||\tilde{{\mathbf{q}}}^{\text{p,i}} - {\mathbf{q}}^{i}|| / ||{\mathbf{q}}^{i}||$, which leads to lower and more interpretable values. We also extended Table 1 to incorporate the additional baseline.

---

> > > ### Author Response · Authors · 2024-11-18
> > > **Response to Reviewer wYEc (3/3)**
> > >
> > > **References**
> > >
> > > [1] Michael Lutter and Jan Peters. Combining physics and deep learning to learn continuous-timedynamics models. Intl. Journal of Robotics Research, 42(3):83–107, 2023.
> > >
> > > [2] Aasa Feragen and Andrea Fuster. Modeling, Analysis, and Visualization of Anisotropy. In Geometries and Interpolations for Symmetric Positive Definite Matrices 85–113 (2017).
> > >
> > > [3] Z. Lin. Riemannian Geometry of Symmetric Positive Definite Matrices via Cholesky Decomposition. SIAM Journal on Matrix Analysis and Applications 40, 1353–1370 (2019).

---

> ### Comment · Reviewer_wYEc · 2024-11-22
>
> 1. For my question, "Is preserving the geometry important between FOM and ROM? Can the author summarize what the benefits are of preserving the structure in physical systems and control?"
>
> I am still confused about your answer. Can you provide some examples or references?
>
> 2. From Section 2.3 to 2.4, you said "In contrast to MOR, LNN consider low-dimensional systems with unknown dynamics that we aim to learn." From my understanding, MOR also involves dimensionality reduction. E.g., you include AE in your Equation (5). So, it's still unclear about the difference between MOR and LNN.

---

> > ### Author Response · Authors · 2024-11-23
> > **second response to reviewer wYEc (1)**
> >
> > Thank you for your response! We hope to clarify the remaining open questions below.
> >
> > > 1. For my question, "Is preserving the geometry important between FOM and ROM? Can the author summarize what the benefits are of preserving the structure in physical systems and control?"
> > >
> > >
> > > I am still confused about your answer. Can you provide some examples or references?
> > >
> >
> > Thank you for following up on this point, which we gladly elaborate on next.
> >
> > We would like to highlight that by enforcing certain geometric properties of the network, we aim at preserving the Lagrangian structure of the model. In our case, we focus on two main properties:
> >
> > - a symmetric positive-definite mass-inertia matrix $\mathbf{M}(\mathbf{q})\in \mathcal{S}_{++}^{n}$ that enforces the quadratic kinetic-energy structure of the Lagrangian dynamics [1];
> > - biorthogonal AE weights $\mathbf{\Psi}^\intercal\mathbf{\Phi} = \mathbf{I}$ that ensure that the embedded latent space preserves Lagrangian structure [2,3].
> >
> > These two geometric properties, in addition to modeling of the network’s equations of motion via the Euler-Lagrange equation (Equation (1), and in the loss equation (13)), ensure the Lagrangian structure in our reduced model.
> >
> > **Taking a Lagrangian viewpoint on mechanical systems allows us to account for important properties of such systems, notably:**
> >
> > - Their energy is conserved in the absence of dissipation and forcing;
> > - Quantities associated with the symmetries of the system are also conserved;
> > - Their dynamics satisfy the variational principle expressed via the equations of motions given in Equation 1.
> >
> > Preserving the Lagrangian structure of the FOM in the ROM essentially leads to the preservation of these properties [4]. Specifically, the Lagrangian ROM $\check{\mathcal{L}}$ is obtained from the Lagrangian FOM $\mathcal{L}$ through the pullback of the lifted embedding as $\check{\mathcal{L}}=\mathcal{L} \circ \psi$ (see Equation 9). Then, the Lagrangian ROM preserve the properties above as:
> >
> > - The reduced energy $\check{\mathcal{E}} = \mathcal{E} \circ \psi$ obtained via the pullback of the lifted embedding is conserved when the FOM energy $\mathcal{E}$ is conserved, see [2];
> > - Quantities associated with the symmetries of the FOM are also conserved in the ROM;
> > - The dynamic of the ROM follow the same variational principle as the dynamics of the FOM, i.e., they follow the reduced equation of motions given by Equation 10, which relates to the equation of motion of the FOM (Equation 1) via the embedding as described in line 269-270.
> >
> > Several prior works on structure-preserving MOR [5, 6] investigated the benefits of preserving the structure of the FOM within the ROM. These works found that preserving the structure increased the accuracy and stability of the predicted trajectory regardless of the number of training data. This is particularly important for numerical simulations of high-dimensional systems, and for enabling successful model-based control applications. For instance, Lepri et al. [7] designed an energy-based posture-regulation controller in the latent space of a structure-preserving Hamiltonian ROM. Convergence to the desired posture can be ensured thanks to the preservation of the Hamiltonian structure of the FOM in the latent space. Similar controllers, e.g., energy-based controllers inspired from [8,9], can be designed in the latent space of Lagrangian ROMs.
> >
> > > 2. From Section 2.3 to 2.4, you said "In contrast to MOR, LNN consider low-dimensional systems with unknown dynamics that we aim to learn." From my understanding, MOR also involves dimensionality reduction. E.g., you include AE in your Equation (5). So, it's still unclear about the difference between MOR and LNN.
> > >
> >
> > As you correctly pointed out, model-order reduction (MOR) [2, 5, 6] aims at reducing the dimensionality of known high-dimensional dynamic models. In other words, given a high-dimensional known dynamic model, traditional MOR methods obtain a low-dimensional surrogate model, e.g., by projecting the high-dimensional trajectories of the system into a subspace learned by an AE.
> >
> > In contrast, LNNs such as [1] do not reduce the dimensionality of the system, but learns its unknown dynamic parameters. The systems consider in previous works are typically low dimensional (2-5 dimensions).
> >
> > In other words, **MOR methods** consider **high-dimensional systems with known dynamic parameters**. **I**n contrast, **LNNs consider low-dimensional systems with unknown dynamic parameters.**

---

> > > ### Author Response · Authors · 2024-11-23
> > > **second response to reviewer wYEc (2)**
> > >
> > > Our contribution lies at the intersection of these two different concepts: We consider ***high-dimensional* systems with *unknown* dynamic parameters**. Our motivation is that LNNs fail to learn dynamic parameters of high-dimensional systems. Therefore, we take inspiration from MOR methods and propose to reduce the dimensionality of the high-dimensional system and to learn the low-dimensional dynamic parameters of the low-dimensional surrogate model. Note that our approach differs from MOR methods, as we assume that the high-dimensional parameters are unknown.
> > >
> > > The AE described by Equation 5 is a typical AE used by some MOR methods [2, 3]. We describe this equation as a background on MOR methods. Section 2.4 then described the background on LNN, and does not relate to the AE described in Section 2.3. To make this more clear, we merged Section 2.2 and 2.3 in a single background section on “Model order reduction”.
> > >
> > > [1] Michael Lutter and Jan Peters. Combining physics and deep learning to learn continuous-timedynamics models. Intl. Journal of Robotics Research, 42(3):83–107, 2023.
> > >
> > > [2] Patrick Buchfink, Silke Glas, Bernard Haasdonk, and Benjamin Unger. Model reduction on manifolds: A differential geometric framework. Physica D: Nonlinear Phenomena 468 (2024).
> > >
> > > [3] Samuel E. Otto, Gregory R. Macchio, and Clarence W. Rowley. Learning nonlinear projections
> > > for reduced-order modeling of dynamical systems using constrained autoencoders. Chaos: An
> > > Interdisciplinary Journal of Nonlinear Science, 33(11), 2023.
> > >
> > > [4] S. Lall, P. Krysl, and J. E. Marsden. Structure-preserving model reduction for mechanical systems. Phys. D: Nonlinear Phenom. 184, 304–318, 2003.
> > >
> > > [5] Kevin Carlberg, Ray Tuminaro, and Paul Boggs. Preserving Lagrangian structure in nonlinear model
> > > reduction with application to structural dynamics. SIAM Journal on Scientific Computing, 37(2):
> > > B153–B184, 2015.
> > >
> > > [6] Harsh Sharma, Hongliang Mu, Patrick Buchfink, Rudy Geelen, Silke Glas, and Boris Kramer. Symplectic model reduction of Hamiltonian systems using data-driven quadratic manifolds. Computer Methods in Applied Mechanics and Engineering, 417:116402, 2023.
> > >
> > > [7] M. Lepri, D. Bacciu, and C. Della Santina. Neural Autoencoder-Based Structure-Preserving Model Order Reduction and Control Design for High-Dimensional Physical Systems. arXiv, 2023.
> > >
> > > [8] Lutter, M. and  Listmann, K. and  Peters, J. Deep Lagrangian Networks for end-to-end learning of energy-based control for under-actuated systems, International Conference on Intelligent Robots & Systems (IROS), 2019.
> > >
> > > [9] Yaofeng Desmond Zhong, Biswadip Dey, Amit Chakraborty. Symplectic ODE-Net: Learning Hamiltonian Dynamics with Control. International COnference on Learning Representations (ICLR), 2020.

---

### Official Review · Reviewer_rYLi · 2024-11-02

**Soundness:** 3
**Presentation:** 3
**Contribution:** 1
**Rating:** 5
**Confidence:** 4

**Summary:**

The paper proposes a geometric network architecture that learns reduced-order Lagrangian dynamics for high-dimensional systems, using a Riemannian approach to jointly learn a structure-preserving latent space and the associated low-dimensional dynamics. This method efficiently predicts long-term dynamics for rigid and deformable systems while improving interpretability based on physics.

**Strengths:**

The strengths are listed as followed
- Utilizes geometry and physics as inductive biases, leading to a physically consistent model to improve the interpretability.
- Effectively learns reduced-order dynamics for high-dimensional rigid and deformable systems with accurate long-term predictions.
- Joint optimization of latent space and reduced dynamic parameters enhances performance, particularly in low-data regimes.
- Incorporates intrinsic geometry of the problem, aiding in the preservation of energy and interpretability of the model.
- Outperforms standard autoencoders by employing a constrained auto-encoder trained with Riemannian optimization.

**Weaknesses:**

### Contribution and Originality

- I question the novelty of this paper. The proposed approach closely resembles existing methods described in [1, 2]. Additionally, the authors have not cited reference [1], which appears to be highly relevant to their work.

- The contribution of this paper is overstated, and the true contribution appears to be quite incremental. Based on my understanding, the theoretical foundation relies heavily on the published work in [2], with many of the equations directly derived from [2]. There seem to be no novel contributions to the methods or framework, as the use of an autoencoder for reduced-order modeling was already proposed in [2]. This paper primarily appears to be an implementation of the original idea from [2].

- For the learning framework, this paper replaces the Galerkin method (subspace method) with an autoencoder structure for learning Lagrangian dynamics. However, the autoencoder representation is fundamentally similar to the Galerkin (subspace) method in its underlying principles. The primary distinction from reference [1] lies in the representation of the mass-inertia matrix as a symmetric positive definite (SPD) manifold within the configuration space, whereas the loss functions and implementation details remain largely consistent with those in [1]. The authors should clearly elucidate the key differences from prior work and provide a justification for why reference [1] was not cited.


### Comparison

- There is no comparison with other existing algorithms [1, 3]. The authors claim scability of its algorithm, but the current work only tests up to 600 DoFs using deep learning. In contrast, the existing work in [1] conducted experiments with 259,744 DoFs, even using Matlab. This raises questions about the claimed scability and performance improvements.

### Computational Complexity

- The computational complexity is significantly higher compared to [1], mainly due to the learning of the SPD manifold. It does not seem necessary to optimize such a complex manifold for high DoF cases, as the constant symmetric positive-definite surrogate used in [1] performs adequately in a complex scenario.


[1]. Sharma, Harsh, and Boris Kramer. "Preserving Lagrangian structure in data-driven reduced-order modeling of large-scale dynamical systems." Physica D: Nonlinear Phenomena 462 (2024): 134128.

[2] Buchfink, Patrick, et al. "Model reduction on manifolds: A differential geometric framework." Physica D: Nonlinear Phenomena 468 (2024): 134299.

[3] Harsh Sharma, Hongliang Mu, Patrick Buchfink, Rudy Geelen, Silke Glas, and Boris Kramer. Sym- plectic model reduction of Hamiltonian systems using data-driven quadratic manifolds. Computer Methods in Applied Mechanics and Engineering, 417:116402, 2023. doi: 10.1016/j.cma.2023. 116402.

**Questions:**

see the weakness part.

---

> ### Author Response · Authors · 2024-11-18
> **Response to Reviewer rYLi (1)**
>
> Thank you very much for your valuable feedback. Please find our response to your comment and questions below.
>
>
> ### Contribution and originality:
>
> - > I question the novelty of this paper. The proposed approach closely resembles existing methods described in [1, 2]. Additionally, the authors have not cited reference [1], which appears to be highly relevant to their work.
> - > The contribution of this paper is overstated, and the true contribution appears to be quite incremental. Based on my understanding, the theoretical foundation relies heavily on the published work in [2], with many of the equations directly derived from [2]. There seem to be no novel contributions to the methods or framework, as the use of an autoencoder for reduced-order modeling was already proposed in [2]. This paper primarily appears to be an implementation of the original idea from [2].
> - > For the learning framework, this paper replaces the Galerkin method (subspace method) with an autoencoder structure for learning Lagrangian dynamics. However, the autoencoder representation is fundamentally similar to the Galerkin (subspace) method in its underlying principles. The primary distinction from reference [1] lies in the representation of the mass-inertia matrix as a symmetric positive definite (SPD) manifold within the configuration space, whereas the loss functions and implementation details remain largely consistent with those in [1]. The authors should clearly elucidate the key differences from prior work and provide a justification for why reference [1] was not cited.
>
> Please refer to **our response to all reviewers** for the discussion and clarification of our contributions:
>
> - Point 1 clarifies our contributions, see in particular 1a, 1b, and 1c for the similarities and differences with MOR methods, with [2], and with [1], respectively.
> - Point 2 analyses the similarities and differences of our approach in comparison to [1].
>
> In particular, we would like to emphasize again that:
>
> 1. Our work differs from the traditional MOR setting and from [2] in that **our model does not have access to the full-order model dynamic parameters**, while MOR methods including [2] are intrusive and assume to have access to the full-order dynamics, i.e,  the governing equations and the dynamic parameters.
> 2. Our work differs from L-OpInf [1] in that **it can deal with systems that are not linearly reducible and it does not assume a fixed constant reduced-order mass-inertia matrix**. Therefore, it is **more general** and allows for **more expressive** models as shown by our additional comparison in Section 5.2.
>
> We would also like to thank you for pointing us to [1] and to highlight that we now corrected this unfortunate omission by adding both a theoretical (Section 1) and experimental (Section 5.2) comparisons with L-OpInf in the revised version of the paper.
>
> ### Comparison:
>
> > There is no comparison with other existing algorithms [1, 3]. The authors claim scability of its algorithm, but the current work only tests up to 600 DoFs using deep learning. In contrast, the existing work in [1] conducted experiments with 259, 744 DoFs, even using Matlab. This raises questions about the claimed scability and performance improvements.
>
> Please refer to **our response to all reviewers** (points 2 and 3) for the description and analysis of the additional experimental comparisons with L-OpInf.
>
> We would like to highlight that **L-OpInf [1] already fails in the 16-dimensional example** of Section 5.2 due to the fact that the high-dimensional state space of the coupled pendulum **is not linearly reducible**. Moreover, we were not able to generate stable results with L-OpInf for the rope and cloth datasets: The CVX solver returns solution status as ‘Infeasible’, and fails to provide a reasonable solution. Despite its reduced computational complexity and its impressive performance on linearly-reducible systems reported in [1], L-OpInf is not able to learn the dynamics of high-dimensional systems which are not linearly reducible.

---

> > ### Author Response · Authors · 2024-11-18
> > **Response to Reviewer rYLi (2)**
> >
> > ### Computational complexity:
> >
> > > The computational complexity is significantly higher compared to [1], mainly due to the learning of the SPD manifold. It does not seem necessary to optimize such a complex manifold for high DoF cases, as the constant symmetric positive-definite surrogate used in [1] performs adequately in a complex scenario.
> >
> > Please refer to **our response to all reviewers** (points 2 and 3) for the description and analysis of the additional experimental comparisons with L-OpInf. Despite its reduced computational complexity and its impressive performance on linearly-reducible systems reported in [1], L-OpInf is not able to learn the dynamics of high-dimensional systems which are not linearly reducible. All in all, we believe that **the application of computationally more intensive deep learning methods such as RO-LNN is justified for systems with complex dynamics that are not linearly reducible, as such approach obviously lead to significantly-increased performance**.
> >
> > ### References
> >
> > [1] Harsh Sharma and Boris Kramer. Preserving Lagrangian structure in data-driven reduced-order modeling of large-scale dynamical systems. Physica D: Nonlinear Phenomena, 462:134128, 2024.
> >
> > [2] Patrick Buchfink, Silke Glas, Bernard Haasdonk, and Benjamin Unger. Model reduction on manifolds: A differential geometric framework. Physica D: Nonlinear Phenomena 468 (2024).
> >
> > [3] Harsh Sharma, Hongliang Mu, Patrick Buchfink, Rudy Geelen, Silke Glas, and Boris Kramer. Symplectic model reduction of Hamiltonian systems using data-driven quadratic manifolds. ComputerMethods in Applied Mechanics and Engineering, 417:116402, 2023.

---

> ### Comment · Area_Chair_Qyhf · 2024-11-27
> **Rebuttal Response**
>
> Please let the authors know if their rebuttal has addressed any of your concerns. Thank you.

---

> ### Comment · Reviewer_rYLi · 2024-12-01
>
> Response to the novelty
>
> Thank you for your detailed response. I have raised my score to 5. However, I still have some concerns, which are outlined below.
>
> > It can deal with systems that are not linearly reducible and it does not assume a fixed constant reduced-order mass-inertia matrix.
>
> 1. The difference between [1] and your work seems primarily related to the mass-inertia matrix. From my perspective, the main change is replacing linear projection with an auto-encoder structure, which, theoretically, is not a significant departure. While the proposed method is presented as more general, this particular idea is not entirely novel. The real contribution appears to be in changing the constant mass-inertia matrix to a structure defined on an SDP manifold. I have compared it in detail with [2], and the formulation differences are minimal.  I'm not sure that this contribution is sufficient for publication at a top-tier conference like ICLR.
>
> Moreover, the optimization on an SDP manifold relates to Hilbert's 5th problem, making it particularly challenging to solve and optimize for high-dimensional matrices [3]. This raises concerns about the scalability of the proposed approach.
>
> 2. The authors mention that the mass-inertia matrix is linearly irreducible. Could this issue be resolved by using a functional representation via the Galerkin method? If so, the claimed contribution of the proposed approach may be diminished.
>
>
> [1]. Sharma, Harsh, and Boris Kramer. "Preserving Lagrangian structure in data-driven reduced-order modeling of large-scale dynamical systems." Physica D: Nonlinear Phenomena 462 (2024): 134128.
>
> [2] Buchfink, Patrick, et al. "Model reduction on manifolds: A differential geometric framework." Physica D: Nonlinear Phenomena 468 (2024): 134299.
>
> [3]  https://terrytao.wordpress.com/wp-content/uploads/2014/11/gsm-153.pdf

---

> > ### Author Response · Authors · 2024-12-02
> > **second response to reviewer rYLi (1)**
> >
> > Thank you very much for your feedback and for raising your score! We hope to clarify the remaining concerns in the following.
> >
> > > 1. The difference between [1] and your work seems primarily related to the mass-inertia matrix. From my perspective, the main change is replacing linear projection with an auto-encoder structure, which, theoretically, is not a significant departure. While the proposed method is presented as more general, this particular idea is not entirely novel. The real contribution appears to be in changing the constant mass-inertia matrix to a structure defined on an SDP manifold. I have compared it in detail with [2], and the formulation differences are minimal. I'm not sure that this contribution is sufficient for publication at a top-tier conference like ICLR.
> > >
> >
> > As highlighted in our introduction, our work closely relates to [1] as both approaches are non-intrusive and identify low-dimensional dynamic parameters in a structure-preserving linear subspace obtained from high-dimensional state observations. However, we would like to emphasize that **our contribution compared to [1] goes beyond considering the SPD structure of the mass-inertia**. Our work distinguishes from [1] as:
> >
> > 1. As mentioned by the Reviewer, in contrast to L-OpInf [1] which consider linear subspaces, we pursue a more general approach and learn a non-linear embedding of the FOM obtained via a constrained auto-encoder.
> > 2. We alleviate the constraints imposed by L-OpInf [1] on the mass-inertia matrix. While L-OpInf sets $\check{\mathbf{M}}=\mathbf{I}$, **our approach considers a reduced-order mass-inertia matrix as a general function of the reduced-order position**, i.e., $\check{\mathbf{M}}(\check{\mathbf{q}})$.
> > 3. We alleviate the constraints imposed by L-OpInf [1] on the form of the potential energy. While L-OpInf sets the potential energy as quadratic $V(\mathbf{q}) = \frac{1}{2}\mathbf{q}^\intercal\mathbf{K}\mathbf{q}$, our approach considers the reduced-order potential energy a general function of the reduced-order position, i.e., $\check{V}(\check{\mathbf{q}})$.
> > 4. We use a deep Lagrangian network to learn the ROM parameters. As mentioned by the Reviewer, we propose to **directly parametrize the mass-inertia matrix with a SPD network,** which accounts for geometry of the mass-inertia matrix.
> > 5. L-OpInf [1] identifies the linear subspace and the reduced-order dynamic parameters sequentially. Instead, our approach jointly identifies a non-linear reduced representation and the associated dynamic parameters.
> >
> > As highlighted in Section 2.2., we build on the work of Buchfink et al [2], who present a geometric MOR framework. However, the framework in [2] is intrusive as it assume a high-dimensional model (FOM) with **fully-known dynamic parameters**. In our work, **we depart from [2] in that we consider that the high-dimensional dynamics are unknown**. However, **we leverage the theoretical framework of [2] to define a non-linear reduced representation** of the high-dimensional Lagrangian configuration space. As opposed to [2], we learn both a non-linear reduced representation of the high-dimensional Lagrangian configuration space along with a reduced mass-inertia matrix and potential energy, i.e. $\check{\mathbf{M}}(\check{\mathbf{q}})$ and $\check{V}$ from snapshots of system trajectories.
> >
> > All in all, our method enables learning dynamic parameters of nonlinear high-dimensional systems from trajectory observations.  Although they also preserve Lagrangian structure, this is not achieved by state-of-the-art methods. Specifically:
> >
> > - MOR methods such as [2] are **intrusive**, i.e., they **assume to have full access to the full-order system dynamics;**
> > - LNNs such as [3] fail to learn high-dimensional dynamic parameters and neglect the geometry of the mass-inertia matrix;
> > - Linear non-intrusive MOR methods such as [1] lack expressivity and are limited to linearly-reducible systems.
> >
> > Our approach builds upon prior works from all three subfields, but achieves a combined goal. Moreover, we would like to emphasize that our proposed approach also lead to **significant experimental improvements** compared to the state of the art:
> >
> > 1. Our proposed geometric LNN (Section 3) **outperforms the state-of-the-art LNN** [3] to learn low-dimensional dynamic parameters, as shown in Section 5.1.
> > 2. Our proposed reduced-order LNN (Section 4) **significantly outperforms both the state-of-the-art LNN [3] and L-OpInf [1]** when learning high-dimensional dynamic parameters. As shown in Section 5.2, **our approach achieves prediction errors that are three order of magnitude lower than LNN and L-OpInf** (see Table 1).

---

> > > ### Author Response · Authors · 2024-12-02
> > > **second response to reviewer rYLi (2)**
> > >
> > > 3. Although this may not be a significant departure from a theoretical perspective, our nonlinear treatment of the problem is of high importance when working with nonlinear systems. Specifically, it allows our approach to **significantly outperforms L-OpInf, whose performance is highly limited** when learning the dynamics of systems that require **non-linear reductions** to feature good approximation properties, as shown by our experiment in Section 5.2.1.
> > >
> > > Overall, our experiments show the practical relevance of the theoretical contributions of this paper.
> > >
> > >
> > >
> > > > Moreover, the optimization on an SDP manifold relates to Hilbert's 5th problem, making it particularly challenging to solve and optimize for high-dimensional matrices [3]. This raises concerns about the scalability of the proposed approach.
> > > >
> > >
> > > Thanks for raising this interesting point! However, Hilbert’s 5th problem is concerned with the topological description of Lie groups. In our work, **we treat the SPD manifold as a Riemannian manifold, and leverage tools from Riemannian geometry** in our SPD network and to optimize SPD parameters.
> > >
> > > We would also like to emphasize that the **SPD matrices learned and optimized in our work are of limited dimensionality**. For low-dimensional systems, we utilize our geometric LNN presented in Section 3. For high-dimensional systems, we leverage the reduced-order model LNN, which optimizes reduced-order dynamical systems, whose dimensionality is significantly lower than that of the original system. For instance, our experiments in Section 5.2 consider reduced-order configuration spaces of dimension ranging from 4 to 14. This range of dimension is well handled both by SPD networks presented in our paper and in previous works [4,5].
> > >
> > > > 2. The authors mention that the mass-inertia matrix is linearly irreducible. Could this issue be resolved by using a functional representation via the Galerkin method? If so, the claimed contribution of the proposed approach may be diminished.
> > > >
> > >
> > > The considered ROM (cf. lines 269-270) corresponds to a manifold Galerkin projection [2,6] of the FOM **onto a nonlinear subspace** via the projection basis $d\mathbf{\varphi}_{\mathcal{Q}}| _{\check{\mathbf{q}}}$ (see for Lagrangian systems, section 5 in [2]). However, it is important to emphasize again that, compared to Buchfink et al. [2], we do not have access to the FOM parameters, and thus cannot obtain the ROM parameters via this projection.
> > >
> > > ### Additional comparison to linear Galerkin method
> > >
> > > To emphasize the significance of considering nonlinear submanifolds, such as the ones learnt by an AE, we implemented an additional baseline that applies a linear Galerkin projection in a non-intrusive manner and leverages our geometric LNNs to learn the low-dimensional dynamic parameters. We carry out experiments for the coupled pendulum presented in Section 5.2.1. Therefore, this additional experiment would correspond to an additional column in Table 1, which we will update in the final version of our manuscript.
> > >
> > > Analogously to [1], we consider a training set of trajectory snapshots $\\{\\mathbf{q}^i, \\dot{\\mathbf{q}}^i, \\mathbf{\\tau}^i \\}^N_{i=1}$ to obtain a linear basis $\mathbf{V}$ via proper orthogonal decomposition (POD) of the configuration manifold for a 4-dimensional latent space. We then train a modified version of the RO-LNN with ODE-loss (equation (14)) over multiple steps $H_{\text{train}}=8$, where the AE-related components in the loss equation are dropped and instead the POD basis $\mathbf{V}$ is used to reduce high-dimensional observations. We then test the obtained ROM analogously as the other models of Table 1 over 10 trajectories. We obtain a mean and standard deviation for the final relative error in reconstructed position predictions $||\tilde{\mathbf{q}}^{}_{p,i} - \mathbf{q}^{}_i || / ||\mathbf{q}^{}_i||= 2.45\times 10^{-1} \pm 9.64\times 10^{-2}$. Although this error is lower than that achieved by L-OpInf ($1.31 \times 10^{1} \pm 8.54 \times 10^{0}$) , it remains **2 orders of magnitude higher than the error achieved by our model RO-LNN (ODE), for which** $|| \tilde{\mathbf{q}}^{} _{p,i} - \mathbf{q}^{}_i|| / ||\mathbf{q}^{}_i|| = 9.39 \times 10^{-3} \pm 3.16 \times 10^{-3}$	.
> > >
> > > In other words, both the geometric LNN and the jointly-trained nonlinear projection contribute to the increased performance of our model.

---

> > > > ### Author Response · Authors · 2024-12-02
> > > > **second response to reviewer rYLi (3/3)**
> > > >
> > > > [1] Harsh Sharma and Boris Kramer. Preserving Lagrangian structure in data-driven reduced-order modeling of large-scale dynamical systems. Physica D: Nonlinear Phenomena, 462:134128, 2024.
> > > >
> > > > [2] Patrick Buchfink, Silke Glas, Bernard Haasdonk, and Benjamin Unger. Model reduction on manifolds: A differential geometric framework. Physica D: Nonlinear Phenomena 468 (2024).
> > > >
> > > > [3] Michael Lutter and Jan Peters. Combining physics and deep learning to learn continuous-timedynamics models. Intl. Journal of Robotics Research, 42(3):83–107, 2023.
> > > >
> > > > [4] Xuan Son Nguyen, Shuo Yang, and Aymeric Histace. Matrix manifold neural networks++.
> > > > In The Twelfth International Conference on Learning Representations (ICLR), 2024
> > > >
> > > > [5] Federico Lopez, Beatrice Pozzetti, Steve Trettel, Michael Strube, and Anna Wienhard.
> > > > Vector-valued distance and gyrocalculus on the space of symmetric positive definite matri-
> > > > ces. In Neural Information Processing Systems (NeurIPS), 2021.
> > > >
> > > > [6] Kookjin Lee and Kevin T. Carlberg. Model reduction of dynamical systems on nonlinear manifolds
> > > > using deep convolutional autoencoders. Journal of Computational Physics, 404:108973, 2020.

---

### Official Review · Reviewer_7VkD · 2024-11-03

**Soundness:** 2
**Presentation:** 2
**Contribution:** 2
**Rating:** 6
**Confidence:** 3

**Summary:**

The paper presents an idea of a neural network to learn the dynamics of a system using Reduced Order Model (ROM). ROM is achieved by mapping the state-space dimensions into a lower dimension and learn the dynamics in the reduced lower dimensional space. The authors proposes a neural network that learns the mass matrix and the potential function in the lower dimensional, and compute the acceleration based on Lagrangian principle. The model is trained in 2 ways: (1) learning the acceleration, and (2) learning the trajectory (and velocity) in a roll-out dynamics. The results show significantly better results than Lagrangian Neural Networks.

**Strengths:**

1. The paper presents experimental results on increasing complexity of the case
2. It shows better results compared to Lagrangian Neural Network

**Weaknesses:**

Major:
1. It is unclear where the novelty of the idea lies and what advantage it brings with respect to the previous works. The use of auto-encoder to reduce the dimensionality of the system is not a new idea. In fact, several papers in the same field like Hamiltonian Neural Network, Lagrangian Neural Network also used auto-encoder to learn dynamics of systems from images. If I understand it correctly, the novelty of the idea lies in a neural network learning both the mass matrix and potential for the Lagrange term. However, it is unclear what benefits does this new formulation of Lagrange term has in comparison with, for example, DeLan and LNN.
2. Reducing the dynamics to lower order dimension does not mean lower computational demand. It is possible, for example, that the lower dimensional dynamics has stronger stiffness than the original dimension. The paper does not present any results comparing the wall-clock time of solving ODE in the original dimension vs solving learned ODE in the lower dimension using the same hardware (e.g., CPU and GPU) to be convincing that their method really achieve computational reduction.
3. Lack of comparisons with other architectures. It seems like the authors only comparing the experiment with LNN. There are a lot of other works in this area that the authors can compare their work with. For example, Hamiltonian Neural Networks (https://arxiv.org/abs/1906.01563), dissipative HNN (https://arxiv.org/abs/2201.10085), dissipative SymOden (https://arxiv.org/abs/2002.08860), Constant of motion network (https://arxiv.org/abs/2208.10387), etc.

Minor comments:
1. Please avoid using the unscientific notation like $1e^{-5}$. Instead, please use notations like $1\times 10^{-5}$.

**Questions:**

1. The idea of using Lagrangian principle is usually made to conserve the energy. However, some of the case (e.g., the cloth) does not seem to conserve the energy. How can the authors justify the use of Lagrangian principle in such cases?
2. The error in Table 1 seems to be very high (it reaches $1\times 10^8$). Why can this happen? Normally, a well-setup neural network train should be normalized so the error starts around 1. The error reaches $10^8$ seems to indicate that the error is not normalized. Does this mean the training is not normalized?
3. (Related to weakness #2) Computational complexity in solving the ODE of a system does not only depend on how many dimension it is, but also how stiff the ODE is. Is there any effort in reducing the stiffness of the equation to ensure lower computational demand?
4. How is the roll-out of the NN done? What kind of ODE solver is used? I found in the appendix that the simulation is done using RK4. Is the same method used to roll-out the NN for the ODE loss?
5. Why the number of rolled out time steps $H=8$ quite small in computing the ODE loss? What is preventing it from using higher number of time steps?
6. How was the time step $\Delta t=10^{-3}$ chosen? How would choosing larger time step affect the roll-out prediction error?

---

> ### Author Response · Authors · 2024-11-18
> **Response to Reviewer 7VkD (1)**
>
> Many thanks for your insightful feedback! Please see below for our answers.
>
> ## Weaknesses:
>
> ### Major:
>
> 1. >It is unclear where the novelty of the idea lies and what advantage it brings with respect to the previous works. The use of auto-encoder to reduce the dimensionality of the system is not a new idea. In fact, several papers in the same field like Hamiltonian Neural Network, Lagrangian Neural Network also used auto-encoder to learn dynamics of systems from images. If I understand it correctly, the novelty of the idea lies in a neural network learning both the mass matrix and potential for the Lagrange term. However, it is unclear what benefits does this new formulation of Lagrange term has in comparison with, for example, DeLan and LNN.
>
>     Please refer to **our response to all reviewers** for a discussion: Point (1) clarifies our contributions, see in particular 1d.
>
>     We would like to additionally emphasize that **our approach features two main differences compared to LNN and DeLaN**:
>
>     - As in our approach, DeLaN [1] uses two networks to parametrize the mass-inertia matrix and the potential energy. **However, DeLaN relies on a Cholesky decomposition network** $\mathbf{L}(\mathbf{q};\theta)$ to parametrize the mass-inertia matrix, so that $\mathbf{M}=\mathbf{L}\mathbf{L}^T$.  Instead, we directly parametrize the mass-inertia matrix with a SPD network $\mathbf{M}(\mathbf{q};\theta)$, which accounts for the intrinsic geometry of mass-inertia matrices. This leads to improved performances compared to DeLaN, as shown in Section 5.1.
>     - **DeLaN and LNN are limited to predict the dynamics of low-dimensional physical spaces** (i.e., 2-5 dimensions) and do not scale to high dimensions. We propose an approach that allows us to learn the dynamics of high-dimensional physical systems by learning a surrogate structure-preserving low-dimensional model of the high-dimensional dynamics. Our approach differs from previous works that used LNNs and HNNs in combination with AEs to learn dynamics from images in that it does not consider high-dimensional observations (images) of low-dimensional physical systems but **systems with high-dimensional state spaces,** a.k.a high-dimensional dynamics.
>
> 2. > Reducing the dynamics to lower order dimension does not mean lower computational demand. It is possible, for example, that the lower dimensional dynamics has stronger stiffness than the original dimension. The paper does not present any results comparing the wall-clock time of solving ODE in the original dimension vs solving learned ODE in the lower dimension using the same hardware (e.g., CPU and GPU) to be convincing that their method really achieve computational reduction.
>
>     We thank the reviewer for pointing this out! As discussed in to **our response to all reviewers** (point 1), our method tackles scenarios where **the high-dimensional dynamic parameters are unknown**. Therefore, the ODE cannot be solved in the original dimension and we cannot provide solution times for both the FOM and the ROM.
>
> 3. >Lack of comparisons with other architectures. It seems like the authors only comparing the experiment with LNN. There are a lot of other works in this area that the authors can compare their work with. For example, Hamiltonian Neural Networks (https://arxiv.org/abs/1906.01563), dissipative HNN (https://arxiv.org/abs/2201.10085), dissipative SymOden (https://arxiv.org/abs/2002.08860), Constant of motion network (https://arxiv.org/abs/2208.10387), etc.
>
>     Thank you for these suggestions!
>     Please refer to **our response to all reviewers,** points 2 and 3 for a discussion. In particular, we provide **an additional comparison with the approach presented in [2]**, which is the closest to our work, in Section 5.2. Moreover, we would like to highlight that HNNs [3] and extensions [4,5] are, as LNNs, limited to learning low-dimensional dynamics. Preliminary experiments with HNNs with the high-dimensional systems of Section 5.2 led to similar results as LNNs.
>
> ### Minor comments:
>
> 1. > Please avoid using the unscientific notation like $1e^{-5}$. Instead, please use notations like $1\times 10^{-5}$.
>
>     We updated the notations according to the comment of the reviewer.

---

> > ### Comment · Reviewer_7VkD · 2024-11-22
> > **Addressed most of my concern, but a major concern remains**
> >
> > I would like to thank the authors for their replies to my concerns and questions. Although most of my concerns has been addressed, I still have some doubts on the benefits or the aim of the method.
> >
> > On the paper, it is mentioned the sentence below.
> > > Predicting trajectories of high-dimensional systems is also notoriously difficult due to the computational cost of solving high-dimensional and highly nonlinear differential equations. In this context, model order reduction (MOR) techniques find a computationally efficient yet descriptive low-dimensional surrogate system — a reduced-order model (ROM) — of a given high-dimensional dynamical system or full-order model (FOM) with known dynamics (Schilders et al., 2008).
> >
> > This gives an impression that MOR technique is to get a more computationally efficient system than FOM. However, the authors did not show the computational advantage of their MOR technique vs FOM.
> >
> > On the other hand, the authors argued that their method "*tackles scenarios where the high-dimensional dynamic parameters are unknown*." However, the test cases shown on the paper have known dynamics and according to appendix F, the dataset is generated using simulations (showing that the dynamics are known). Although the neural network might not have the full observation of the simulations' states and the dynamics are hidden from it, the chosen test cases does not convincingly show that it can learn high-dimensional simulations with unknown dynamic parameters.
> >
> > My suggestions to satisfy my concerns are either:
> > 1. Show that, at least in one case, that the ROM can be simulated faster than FOM (an example case would be a molecular dynamics from DFT or any case where it's expensive to simulate); or
> > 2. Find a case of a high-dimensional system with really unknown dynamic parameters and show that it can be learned (not just unknown from the neural network perspective).

---

> > > ### Author Response · Authors · 2024-11-23
> > > **response to remaining concern (1)**
> > >
> > > Many thanks for your response and suggestions! We hope to clarify the remaining major concern in the following.
> > >
> > > > On the paper, it is mentioned the sentence below.
> > > >
> > > >
> > > > > *Predicting trajectories of high-dimensional systems is also notoriously difficult due to the computational cost of solving high-dimensional and highly nonlinear differential equations. In this context, model order reduction (MOR) techniques find a computationally efficient yet descriptive low-dimensional surrogate system — a reduced-order model (ROM) — of a given high-dimensional dynamical system or full-order model (FOM) with known dynamics (Schilders et al., 2008).*
> > > > >
> > > >
> > > > This gives an impression that MOR technique is to get a more computationally efficient system than FOM. However, the authors did not show the computational advantage of their MOR technique vs FOM.
> > > >
> > >
> > > This sentence was aimed at providing a brief motivation and overview of MOR methods, as we leverage them in our paper. Here, we stated why they are usually used in the context of known high-dimensional dynamics.
> > >
> > > Many prior works on MOR methods have clearly demonstrated the computational speedup achieved by projected (structure-preserving) ROMs w.r.t. to FOMs [1-3]. As traditional MOR methods assume the FOM to be known (i.e., that its dynamics parameters are known), these works can use both the FOM and ROM to predict trajectories, and provide a comparison of their computational complexity.
> > >
> > > However, we would like to emphasize that **we are interested in scenarios where the FOM is unknown, and should be inferred from data,** e.g., by using a Lagrangian Neural Network. In such a scenario, there is no known model to reduce in the first place and traditional MOR methods are not applicable. In other words, **our primary objective is to predict trajectories of the FOM, not in reducing the computational cost of predicting them.** We motivate our work with the fact that physics-preserving deep networks fail to learn the nonlinear dynamic parameters of higher-dimensional systems (as experimentally shown in Section 5.2).
> > >
> > > To even enable the learning of such models, we propose to reduce the dimensionality of the problem, and learn the dynamic parameters of a Lagrangian ROM instead of a FOM. To do so, we leverage the MOR framework to obtain a structure-preserving nonlinear subspace via AEs, in which the dynamic parameters of a ROM are learned together with the parameters of the nonlinear embedding map of that subspace. Our ODE-model solely requires a dataset of $N$ observations $\\{\mathbf{q}_i, \dot{\mathbf{q}}_i, \mathbf{\tau}_i\\} _{i=1}^{N}$ to learn the reduced equations of motion.
> > >
> > > > On the other hand, the authors argued that their method "*tackles scenarios where the high-dimensional dynamic parameters are unknown*." However, the test cases shown on the paper have known dynamics and according to appendix F, the dataset is generated using simulations (showing that the dynamics are known). Although the neural network might not have the full observation of the simulations' states and the dynamics are hidden from it, the chosen test cases does not convincingly show that it can learn high-dimensional simulations with unknown dynamic parameters.
> > > >
> > >
> > > As you point out correctly, we use simulations (more precisely, the general-purpose physics engine Mujoco [4]) to generate these observations $\\{\mathbf{q}_i, \dot{\mathbf{q}}_i, \mathbf{\tau}_i\\} _{i=1}^{N}$. However, we would like to emphasize that **we are not able to access the parametrized dynamic model, i.e., the mass-inertia matrix and potential energy, as general functions of any configurations from the simulator.**
> > >
> > > > My suggestions to satisfy my concerns are either:
> > > > 1. Show that, at least in one case, that the ROM can be simulated faster than FOM (an example case would be a molecular > dynamics from DFT or any case where it's expensive to simulate); or
> > > > 2. Find a case of a high-dimensional system with really unknown dynamic parameters and show that it can be learned (not just > unknown from the neural network perspective).
> > > >
> > >
> > > Regarding 1., we manually derived the equations of motions of the 16-DoF pendulum showcased in our experiments of Section 5.2.1, which we used to simulate the FOM. We then compared the trajectory inference time of this FOM with the trajectory inference time of our model with learned dynamics. We obtain the following inference time for 3000 prediction steps:
> > >
> > > FOM: 113.59 s
> > >
> > > Our learned model: **1.57 s**
> > >
> > > This showcases the computational efficiency of our model. We added this comparison in an additional Appendix G.2.3. However, we would like to emphasize that our contribution considers the case where the FOM dynamics parameters are unknown and the FOM cannot be simulated.

---

> > > > ### Author Response · Authors · 2024-11-23
> > > > **response to remaining concern (2)**
> > > >
> > > > Regarding 2., we would like to highlight that we evaluate our approach similarly as previous works aiming at learning dynamic parameters, e.g., [5, 6]. Namely, the dynamic parameters are learned from a dataset of trajectory observations $\\{\mathbf{q}_i, \dot{\mathbf{q}}_i, \mathbf{\tau}_i\\} _{i=1}^N$ obtained either by simulating the system or by recording real-world trajectories. Note that, in the former case, the simulation engine only serves the purpose of generating realistic trajectory data, and is not a baseline to evaluate performance against. Moreover, real-world experiments on real-world systems, e.g., cloths, are planned as future work.
> > > >
> > > > [1] Kevin Carlberg, Ray Tuminaro, and Paul Boggs. Preserving Lagrangian structure in nonlinear model
> > > > reduction with application to structural dynamics. SIAM Journal on Scientific Computing, 37(2):
> > > > B153–B184, 2015.
> > > >
> > > > [2] Joshua Barnett and Charbel Farhat. Quadratic approximation manifold for mitigating the Kol-
> > > > mogorov barrier in nonlinear projection-based model order reduction. Journal of Computational
> > > > Physics, 464, 2022.
> > > >
> > > > [3] Kookjin Lee and Kevin T. Carlberg. Model reduction of dynamical systems on nonlinear manifolds
> > > > using deep convolutional autoencoders. Journal of Computational Physics, 404:108973, 2020.
> > > >
> > > > [4] Todorov, Emanuel and Erez, Tom and Tassa, Yuval: MuJoCo: A physics engine for model-based control. IEEE/RSJ International Conference on Intelligent Robots and Systems, 2012.
> > > >
> > > > [5] Michael Lutter and Jan Peters. Combining physics and deep learning to learn continuous-timedynamics models. Intl. Journal of Robotics Research, 42(3):83–107, 2023.
> > > >
> > > > [6] Samuel Greydanus, Misko Dzamba, and Jason Yosinski. Hamiltonian neural networks. In Neural Information Processing Systems (NeurIPS), volume 32, 2019.

---

> > > > ### Comment · Reviewer_7VkD · 2024-11-25
> > > > **Raised my score from 5 to 6 ...**
> > > >
> > > > ... because the authors have addressed all my concerns.

---

> ### Author Response · Authors · 2024-11-18
> **Response to Reviewer 7VkD (2)**
>
> ## Questions:
>
> 1. >The idea of using Lagrangian principle is usually made to conserve the energy. However, some of the case (e.g., the cloth) does not seem to conserve the energy. How can the authors justify the use of Lagrangian principle in such cases?
>
>     As you pointed out, unforced Lagrangian systems (i.e., with generalized forces $\mathbf \tau = \mathbf 0$) preserve the energy. This corresponds, e.g., to the double pendulum scenario in Section 5.1. In the cloth-experiment, we model an object with mass in contact with the sphere, and obtain the contact forces from the simulator as input to the model in the form of generalized forces $\mathbf{\tau}\neq 0$. Therefore, in this actuated scenario, the model is not expected to conserve the energy throughout the whole trajectory. However, we expect the energy to be conserved during the first part of the trajectory which corresponds to the free fall. As shown in Fig. 16, the energy level indeed seems constant during the first ~0.1 second. Notice that the generalized forces are also handled by the structure-preserving reduction, see Section 4.1.
>
> 2. >The error in Table 1 seems to be very high (it reaches $1\times 10^8$). Why can this happen? Normally, a well-setup neural network train should be normalized so the error starts around 1. The error reaches $10^8$ seems to indicate that the error is not normalized. Does this mean the training is not normalized?
>
>     Thank you for pointing this out. During training, data was appropriately scaled, as mentioned by the reviewer. Table 1 provided unnormalized results on the testing data and reported the mean of the squared distances between predictions and ground truth values over the whole testing dataset of N samples, i.e. $\frac{1}{n}\sum_{i=1}^N||\tilde{\ddot{\mathbf q}}^{\text{p,i}} - \ddot{\mathbf q}^i||^2$. For large absolute acceleration values, high-dimensional (16 dimensions) vectors $\ddot{\mathbf q}^i$, and an inaccurate model, we indeed observed high absolute values, which were hard to interpret.
>
>     We agree that this format was not easily readable and hard to interpret. Therefore, we updated Table 1 to report relative errors in acceleration $||\tilde{\ddot{\mathbf q}}^{\text{p,i}} - \ddot{\mathbf q}^i|| / ||\ddot{\mathbf q}^i ||$, velocity $||\tilde{\dot{\mathbf q}}^{\text{p,i}} - \dot{\mathbf q}^i|| / ||\dot{\mathbf q}^i||$, and position $||\tilde{{\mathbf q}}^{\text{p,i}} - {\mathbf q}^i|| / ||{\mathbf q}^i||$. We also extended Table 1 to incorporate the additional baseline. Notice that the relative error for the full-order LNN is still significantly >1, i.e. we infer no significant training progress.
>
>
> 3. >(Related to weakness #2) Computational complexity in solving the ODE of a system does not only depend on how many dimension it is, but also how stiff the ODE is. Is there any effort in reducing the stiffness of the equation to ensure lower computational demand?
>
>     Thank you for this interesting suggestion! We would like to emphasize that our primary goal is not to accelerate the ODE computation, but to **learn unknown dynamic parameters**. Although we could consider incorporating additional bias to accelerate the computation of the low-dimensional ODE, we see this as being out of the scope of the contributions of this paper and will consider it for future research.
>
>
> 4. >How is the roll-out of the NN done? What kind of ODE solver is used? I found in the appendix that the simulation is done using RK4. Is the same method used to roll-out the NN for the ODE loss?
>
>     The roll-out (both in training and testing) uses a simple Euler forward integration of variable number of steps. This is a common integration method used for training of other structure-preserving gray-box methods, as described by [7]. We use the same integration method in our multi-step integration loss in Section 4.3. We added the information concerning the roll-out in Appendix F.
>
> 5. >Why the number of rolled out time steps $H=8$ quite small in computing the ODE loss? What is preventing it from using higher number of time steps?
>
>     Theoretically, any number $H$ could be chosen. However, the training complexity increases with the number of steps, due to the increased complexity of backpropagating through a multi-step loss with a longer horizon. We empirically found that $H=8$ is a good trade-off and leads to good results in our experiments.
>
>
> 6. >How was the time step $\Delta t = 10^{-3}$ chosen? How would choosing larger time step affect the roll-out prediction error?*
>
>     We used commonly-used values for simulating physical systems via differentiable ODE solvers, as mentioned in [6] and increased the sampling rate when working with very high-dimensional deformable structures, e.g., $\Delta t = 10^{-4}$ for the cloth experiment in Section 5.2.3, as is generally recommended, e.g. by the simulation software [7].

---

> ### Author Response · Authors · 2024-11-18
> **Response to Reviewer 7VkD (3)**
>
> ## References
>
> [1] Michael Lutter and Jan Peters. Combining physics and deep learning to learn continuous-timedynamics models. Intl. Journal of Robotics Research, 42(3):83–107, 2023.
>
> [2] Harsh Sharma and Boris Kramer. Preserving Lagrangian structure in data-driven reduced-order modeling of large-scale dynamical systems. Physica D: Nonlinear Phenomena, 462:134128, 2024.
>
> [3] Samuel Greydanus, Misko Dzamba, and Jason Yosinski. Hamiltonian neural networks. In Neural Information Processing Systems (NeurIPS), volume 32, 2019.
>
> [4] Zhong, Y. D., Dey, B. & Chakraborty, A. Dissipative SymODEN: Encoding Hamiltonian Dynamics with Dissipation and Control into Deep Learning. In *ICLR* Workshop on Integration of Deep Neural Models and Differential Equations (DeepDiffEq), 2020.
>
> [5] Andrew Sosanya and Sam Greydanus. Dissipative Hamiltonian Neural Networks: Learning Dissipative and Conservative Dynamics Separately. arXiv, 2022.
>
> [6] Thai Duong, Nikolay Atanasov: Hamiltonian-based Neural ODE Networks on the SE(3) Manifold For Dynamics Learning and Control. Robotics: Science and Systems (RSS), 2020.
>
> [7] Todorov, Emanuel and Erez, Tom and Tassa, Yuval: MuJoCo: A physics engine for model-based control. IEEE/RSJ International Conference on Intelligent Robots and Systems, 2012.

---

### Official Review · Reviewer_4zWX · 2024-11-08

**Soundness:** 3
**Presentation:** 2
**Contribution:** 2
**Rating:** 6
**Confidence:** 4

**Summary:**

This paper investigates the problem of efficiently learning high-dimensional system dynamics from trajectory data. It simultaneously learns a reduced-order model using constrained AE and a dynamics model in the latent space that satisfies Euler-Lagrange equations. Preserving this Lagrangian structure imposes an inductive bias in the latent space to increase the generalization capability of the learned model. The authors proposed a new SPD network architecture to parameterize the mass-inertia matrix in LNN and leveraged Riemannian optimization methods to train the model considering the parameters' geometry.

**Strengths:**

The overall structure of RO-LNN is visualized well in Fig. 1, and the authors put great effort into explaining their architectures and learning algorithms that contain lots of equations in the limited space.

**Weaknesses:**

It seems like the overall idea of learning ROM using constrained AE that preserves the Lagrangian structure is already presented in [Buchfink et al.] cited in the paper. However, this is not clearly pointed out in the paper, and the authors' claims in the introduction and their explanation of the methodology in Section 3&4 somehow mislead the readers to think that the idea is also the main contribution of this paper. This reviewer thinks this paper's contribution on top of the prior work is proposing a new SPD network architecture for parameterizing the mass-inertia matrix and using Riemannian optimization in training to consider the parameters' geometry.

This identification of the contribution should be explained clearly, and the performance of RO-LNN should be compared to the prior works in the experiments.

There are no experimental results that show the benefit of preserving the Lagrangian structure in the latent space. Adding a baseline that learns the model without LNN would help to understand this benefit.

For the proposed architecture of the SPD network in section 3.1, the role of the SPD layers is not explained well, and the experiments show better results without those layers. Justification of having that component is required.

While using Riemannian optimization methods in training RO-LNN is one of the main contributions of this paper, the process is not explained well in Section 3.2.

For eq (13), there is no description of how to compute $\ddot{\check{q}}\_i$. Also, it seems $\tilde{\ddot{q}}\_i = \tilde{\ddot{q}}\_{p,i}$ so that the third term of $\ell_\text{AE}$  should be disregarded, or it should be explained what $\tilde{\ddot{q}}\_i$ is.

In the experiments, the trajectories in fig.2,3 are too cluttered. It might be better to just plot the log-scale difference between the true and predicted trajectories. Also, the plots mislead the readers into thinking that the learned models are used to predict the long-term trajectories while they were fed with ground-truth initial points every 0.025s.

For the cloth experiments, it seems a much shorter timestep size is used for full-horizon prediction (25 steps for 0.025s vs 2500 steps for 0.25s). Then, fig. 6 is not a fair comparison, and this should be pointed out clearly.

**Questions:**

Most of this reviewer's questions is dealt with in the weakness section above.

Additionally, how could we connect the satisfaction of the Euler-Lagrange equations of the latent dynamics to that of the original dynamics?

---

> ### Author Response · Authors · 2024-11-18
> **Response to Reviewer 4zWX (1)**
>
> We are grateful for your detailed and helpful review. We appreciate that you recognized our “great effort into explaining their architectures and learning algorithms”. We address your feedback below.
>
> ### Weaknesses
>
> 1. > It seems like the overall idea of learning ROM using constrained AE that preserves the Lagrangian structure is already presented in [Buchfink et al.] cited in the paper. However, this is not clearly pointed out in the paper, and the authors' claims in the introduction and their explanation of the methodology in Section 3&4 somehow mislead the readers to think that the idea is also the main contribution of this paper. This reviewer thinks this paper's contribution on top of the prior work is proposing a new SPD network architecture for parameterizing the mass-inertia matrix and using Riemannian optimization in training to consider the parameters' geometry.
>
>     >This identification of the contribution should be explained clearly, and the performance of RO-LNN should be compared to the prior works in the experiments.
>
>     Please refer to **our response to all reviewers** for a discussion:
>
>     - Point (1) clarifies our contributions, see in particular 1a-1b and 1e.
>     - Point (3) discusses the comparison to prior work.
>
> 2. > There are no experimental results that show the benefit of preserving the Lagrangian structure in the latent space. Adding a baseline that learns the model without LNN would help to understand this benefit.
>
>     Thank you for your comment! The main benefits of Lagrangian structure-preserving is that **they enable the preservation of the physical properties of the FOM in the ROM**. For instance, structure-preserving ROMs preserve the energy-conservation and stability properties of the corresponding FOM. These properties further enable the application of downstream methods such as energy-based controllers in the ROM. These benefits have been thoroughly discussed theoretically and demonstrated experimentally in the MOR literature [1, 8, 9]. Notably, **they lead to more accurate and numerically-stable results than MOR methods that do not preserve structure**. The linear model from [1], which relates closely to our approach, also reported stability increases compared to non-structure-preserving methods. We added L-OpInf [1] as a baseline in Section 5.2 and show that it outperformed by our model.
>
>     Moreover, it is worth highlighting that considering a Lagrangian (or Hamiltonian) structure is highly beneficial when learning low-dimensional dynamics compared to considering black-box methods [2, 10].
>
>     All in all, our model builds on existing works that have already demonstrated the benefits of physically-consistent models, i.e., (1) of considering a Lagrangian structure when learning dynamics, and (2) of preserving the Lagrangian structure in MOR.

---

> > ### Author Response · Authors · 2024-11-18
> > **Response to Reviewer 4zWX (2)**
> >
> > 3. >For the proposed architecture of the SPD network in section 3.1, the role of the SPD layers is not explained well, and the experiments show better results without those layers. Justification of having that component is required.
> >
> >     Prior works, e.g., [2], rely on a Cholesky decomposition network $\mathbf{L}(\mathbf{q};\theta)$ to parametrize the mass-inertia matrix, so that $\mathbf{M}=\mathbf{L}\mathbf{L}^T$.  While this ensures that the mass-inertia matrix is symmetric positive-definite (SPD), it defines the distances between two SPD matrices based on the Euclidean distance between their Cholesky decompositions. However, it is well known that this distance suffers from the problematic swelling effect [3,4], where the volume of the SPD matrices grows significantly while interpolating between two matrices of identical volumes. Therefore, **mass-inertia matrices inferred via a Cholesky network will suffer from this swelling effect and lead to erroneous predictions of the dynamics**. The solution is to equip the SPD manifold with the affine-invariant or Log-Euclidean metrics, which avoid the swelling effect, and **directly parametrize the mass-inertia matrix with a SPD network** $\mathbf{M}(\mathbf{q};\theta)$.  We clarified these points in Appendix B.
> >
> >     Most state-of-the-art SPD networks aim at transforming SPD matrices to SPD matrices and use SPD layers $g_{\mathcal S_{++}^{n}}:\mathcal S_{++}^{n} \to \mathcal S_{++}^{n}$ [5,6,7]. Our scenario differ from these works in that we aim at learning the mass-inertia matrix as a map $\mathcal Q \to \mathcal S_{++}^{n}$. This justifies the addition of the MLP and exponential map layer to the our network. In this case, the SPD layers add some expressivity to the network by allowing additional transformation in the space of SPD matrices. **Following the state-of-the-art of SPD networks [5,6,7], we believe that implementing the different SPD layers and assessing their performance to learn dynamics was necessary.** Due to the limited performance improvements and increased computational complexity of the SPD layers demonstrated in Section 5.1, we later consider SPD networks composed of an MLP and an exponential-map layer.
> >
> > 4. >While using Riemannian optimization methods in training RO-LNN is one of the main contributions of this paper, the process is not explained well in Section 3.2.
> >
> >     Thank you for pointing this out! We completed Section 3.2 to clarify the notations and the missing descriptions. We also refer to existing literature as we use Riemannian optimization algorithms that are well studied to optimize the parameters of our network. We are glad to further update our description or to add an additional appendix with more detailed explanations if the reviewer thinks that this is necessary.
> >
> > 5. >For eq (13), there is no description of how to compute  $\ddot{\check{\mathbf{q}}}^i$. Also, it seems $\tilde{\ddot{\mathbf{q}}}^i = \tilde{\ddot{\mathbf{q}}}^{p,i}$ so that the third term of $\ell_{\text{AE}}$ should be disregarded, or it should be explained what  is.
> >
> >      $\ddot{\check{\mathbf{q}}}^i$ denotes the point-reduced acceleration (a.k.a. the second derivative of the encoder output), and $\tilde{\ddot{\mathbf{q}}}^i$ refers to the reconstruction of $\ddot{\check{\mathbf{q}}}$ by the decoder. Instead, $\tilde{\ddot{\mathbf{q}}}^{p,i}$ corresponds to the decoded acceleration prediction of the latent network $\ddot{\check{\mathbf{q}}}^{p,i}$.  Therefore, we have  $\tilde{\ddot{\mathbf{q}}}^i \neq \tilde{\ddot{\mathbf{q}}}^{p,i}$, as the former is the AE reconstruction (used to train the AE), and the latter is the decoded prediction of the latent LNN. We clarified this point after Eq. (12) (previously 13) and corrected a typo which led to this confusion.
> >
> > 6. > In the experiments, the trajectories in fig.2,3 are too cluttered. It might be better to just plot the log-scale difference between the true and predicted trajectories. Also, the plots mislead the readers into thinking that the learned models are used to predict the long-term trajectories while they were fed with ground-truth initial points every 0.025s.
> >
> >     Thanks for this feedback! We uploaded a refined version of Figures 2, showcasing the relative prediction errors in position and velocity on a log-scale over time. We also changed Fig. 11, showing results on the sine-tracking task in the appendix, to the same format.
> >
> >     We tested our models on full-horizon in Section 5.1 ($H_{\text{test}}=2000$). For the 16-DoF coupled pendulum and rope experiments of Sections 5.2.1 and 5.2.2, we considered a testing horizon $H_{\text{test}}=25$. Then, in the cloth experiment of Section 5.2.3, we considered both a testing horizon $H_{\text{test}}=25$ and a full horizon ($H_{\text{test}}=2500$). We specified these numbers in the text and in the corresponding figures, adding them where they were previously missing (Figure 2, and Figure 5).

---

> ### Author Response · Authors · 2024-11-18
> **Response to Reviewer 4zWX (3)**
>
> 7. >For the cloth experiments, it seems a much shorter timestep size is used for full-horizon prediction (25 steps for 0.025s vs 2500 steps for 0.25s). Then, fig. 6 is not a fair comparison, and this should be pointed out clearly.
>
>     This comment arose from a typo, which we now corrected. Many thanks for the alertness!
>
>     All experiments with our cloth model were carried out with $\Delta t = 1\times 10^{-4}$ s. The full-horizon experiment predicts cloth configurations for 2500 steps (i.e., 0.25 seconds) from an initial condition, without any updates. In the 25-steps cloth experiment, the model received ground-truth updates every 25 timesteps. This indeed corresponds to predictions over only 0.0025 seconds away from an initial condition. We corrected this in the paper.
>
>
> ### Questions:
>
> 1. >Additionally, how could we connect the satisfaction of the Euler-Lagrange equations of the latent dynamics to that of the original dynamics?
>
>     Following Buchfink et al. [8], and our section 4.1., the pullback of the Lagrangian functional $\mathcal L$ via the lifted embedding $\varphi$ guarantees (under the specific projection properties of the embedding) the preservation of the Lagrangian structure on the embedded submanifold (in this case: the tangent bundle $\mathcal T \check{\mathcal Q}$). Hereby obtaining a reduced Lagrangian $\check{\mathcal L}$, the reduced Euler-Lagrange equations can still be obtained in the standard manner from the principle of least action (see Eq. 10). The reduced solution to the equations of motion is then related to the full-order solution via the (lifted) embedding, which has been obtained via the constrained AE.
>
>
> ### References
>
> [1] Harsh Sharma, Hongliang Mu, Patrick Buchfink, Rudy Geelen, Silke Glas, and Boris Kramer. Symplectic model reduction of Hamiltonian systems using data-driven quadratic manifolds. ComputerMethods in Applied Mechanics and Engineering, 417:116402, 2023.
>
> [2] Michael Lutter and Jan Peters. Combining physics and deep learning to learn continuous-timedynamics models. Intl. Journal of Robotics Research, 42(3):83–107, 2023.
>
> [3] Aasa Feragen and Andrea Fuster. Modeling, Analysis, and Visualization of Anisotropy. In Geometries and Interpolations for Symmetric Positive Definite Matrices 85–113 (2017).
>
> [4] Z. Lin. Riemannian Geometry of Symmetric Positive Definite Matrices via Cholesky Decomposition. SIAM Journal on Matrix Analysis and Applications 40, 1353–1370 (2019).
>
> [5] Federico Lopez, Beatrice Pozzetti, Steve Trettel, Michael Strube, and Anna Wienhard.Vector-valued distance and gyrocalculus on the space of symmetric positive definite matri-ces. In Neural Information Processing Systems (NeurIPS), volume 34, pp. 18350–18366,2021.
>
> [6] Xuan Son Nguyen. The gyro-structure of some matrix manifolds. In Neural Informa-tion Processing Systems (NeurIPS), volume 35, pp. 26618–26630. Curran Associates, Inc.,2022.
>
> [7] Xuan Son Nguyen, Shuo Yang, and Aymeric Histace. Matrix manifold neural networks++. In The Twelfth International Conference on Learning Representations, 2024.
>
> [8] Patrick Buchfink, Silke Glas, Bernard Haasdonk, and Benjamin Unger. Model reduction on manifolds: A differential geometric framework. Physica D: Nonlinear Phenomena 468 (2024).
>
> [9] Kevin Carlberg, Ray Tuminaro, and Paul Boggs. Preserving Lagrangian structure in nonlinear model reduction with application to structural dynamics. SIAM Journal on Scientific Computing, 37(2):B153–B184, 2015
>
> [10] Samuel Greydanus, Misko Dzamba, and Jason Yosinski. Hamiltonian neural networks. In Neural Information Processing Systems (NeurIPS), volume 32, 2019.

---

> > ### Comment · Reviewer_4zWX · 2024-11-26
> >
> > Thank you for the detailed response. The authors clarified all raised concerns, so I increased my score to 6.

---

### Author Response · Authors · 2024-11-18
**Response to all reviewers**

We would like to thank all reviewers greatly for taking the time to review our work, and for their insightful feedback and constructive questions! Here, we address the concerns that were common across different reviewers. We then elaborate on individual questions and comments in the answers to each reviewer. Changes in the manuscript (beyond typos) are highlighted in blue in the revised version of the paper. We hope to reach the reviewers’ attention and we are looking forward to engaging in an interesting discussion!

### **Novelty of the proposed approach**

We would like to emphasize that our contribution is an approach to **learn high-dimensional dynamics**, which is achieved by building on (1) prior works that learn low-dimensional dynamics and on (2) structure-preserving model-order reduction (MOR).

1. **Relationship to MOR [1,2,3]:** Traditional model-order reduction methods are **intrusive**, i.e., they **assume to have full access to the full-order system dynamics.** The dimensionality of the system is then reduced by inferring a lower-dimensional dynamic model (e.g. via projection into subspaces that are obtained from trajectory observations), which is computationally efficient to evaluate. In other words, **MOR methods do not learn dynamics, but reduce the dimension of known dynamics**. In mathematical terms, MOR approaches have access to the mass-inertia matrix $\mathbf{M}(\mathbf{q})$ and potential energy $V(\mathbf{q})$, which are used to obtain reduced parameters, i.e. $\check{\mathbf{M}}$ and $\check{V}$.
              As opposed to the classical MOR framework, **we do not have access to the system's dynamic parameters and aim at learning them.** We believe that this sets our contributions substantially apart from the traditional MOR scenario considered, e.g., in [1,2,3]. Instead, we use MOR-inspired approach to reduce the dimensionality of the problem and obtain a low-dimensional surrogate configuration space. We learn the dynamics in this low-dimensional space without having access to the full-dimensional dynamic parameters.

2. **Relationship to Buchfink et al [1]:** In our work, we employ the theoretical framework in [1] to obtain a low-dimensional structure-preserving nonlinear manifold in which reduced-order dynamic parameters are learned. As discussed above, our work differs from [1] in that **the full-order dynamic parameters are unknown**, while [1] is intrusive and assumes to have access to the full-order Lagrangian $\mathcal L$. In mathematical terms, our approach learns the reduced-order parameters $\check{\mathbf{M}}$ and $\check{V}$ **without having access** to the mass-inertia matrix $\mathbf{M}(\mathbf{q})$ and potential energy $V(\mathbf{q})$. We refer to some formulations that have originally been presented by Buchfink et al [1] in our methodology section, since we transfer them to the scenario of unknown full-order Lagrangians $\mathcal L$ in our section 4.1.

3. **Relationship to Sharma & Kramer [4]:** We would like to thank Reviewer rYLi for pointing out this paper, which is close to our work. The approach presented in [4] identifies reduced-order dynamic parameters in a linear subspace obtained from high-dimensional state observations. It is similar to our approach in that it does not assume access to the full-order dynamic model and identifies low-dimensional dynamics in a latent space. However, **our work is more general and more expressive in that it can deal with systems that are not linearly reducible and does not assume a fixed reduced-order mass-inertia matrix**. We thoroughly discuss the differences between [4] and our approach, as well as additional comparisons in points 2 and 3 below.

4. **Relationship to LNNs and HNNS that use AE [5,6]:** As pointed out by Reviewer 7VkD, several works used LNNs and HNNs in combination with AEs to learn dynamics from images. Despite that the observations (i.e., the images) were high-dimensional, the considered physical systems were low-dimensional (i.e., less than 10 DoFs). **The role of the AE was therefore to extract the states of the low-dimensional system from images**, and the dynamics were learned based on this low-dimensional states. Our approach differs in that it does not consider high-dimensional observations (images) of low-dimensional physical systems but **systems with high-dimensional state spaces,** a.k.a high-dimensional dynamics.

5. **Summary of our contributions:** All in all, our first and main contribution is an approach for **learning unknown high-dimensional dynamics**. Our approach jointly learns a low-dimensional non-linear representation of the configuration space and the associated low-dimensional dynamics, which are then used to predict the behavior of the original system. Our second and third contributions are the introduction of a SPD architecture to parametrize the mass-inertia matrix in the LNN and the use of Riemannian optimization to infer the LNN and AE parameters.

---

> ### Author Response · Authors · 2024-11-18
> **Response to all reviewers (2)**
>
> ### **Discussion and comparison with Sharma & Kramer [4]**
> In contrast to the aforementioned MOR methods, Sharma & Kramer proposed a non-intrusive approach called Lagrangian operator inference (L-OpInf), which identifies reduced-order dynamic parameters in a linear subspace of the high-dimensional state space. L-OpInf differs from our approach as follows:
>
> 1. L-OpInf projects the observed high-dimensional states in a linear subspace obtained via the singular value decomposition of the data matrix. However, it is well known that such linear subspaces with good approximation properties cannot be guaranteed, see [1,7]. Notice that this limitation is also acknowledged in the conclusion of [4] and that the impressively high-dimensional example considered in [4] was known to be linearly reducible. In contrast, **we pursue a more general approach and learn a non-linear embedding of the FOM obtained via a constrained auto-encoder**.
>
> 2. L-OpInf assumes a specific form of reduced-order dynamics where the reduced mass-inertia is fixed as equal to identity, i.e., $\check{\mathbf{M}}=\mathbf{I}$, thus restricting heavily the solution space. As discussed in Section 3.3.3 of [4], this is required to overcome numerical issues arising when identifying the dynamic parameters when $\check{\mathbf{M}}\neq\mathbf{I}$. We argue that **considering a constant reduced mass-inertia metric significantly limits the expressivity of the learned model**, and that it limits its applicability to a certain range of systems. In contrast, **our approach considers a reduced-order mass-inertia matrix as a general function of the reduced-order position**, i.e., $\check{\mathbf{M}}(\check{\mathbf{q}})$.
>
> 3. L-OpInf assumes a specific form of quadratic potential energy $V(\mathbf{q}) = \frac{1}{2}\mathbf{q}\mathbf{K}\mathbf{q}$. Our approach, as DeLaN and LNN, assumes a general form of potential energy specified via a general function $V(\mathbf{q})$ and is therefore more expressive.
>
> 4. L-OpInf identifies the linear subspace and the reduced-order dynamic parameters sequentially. Instead, our approach **jointly** identifies a non-linear reduced representation and the associated dynamic parameters. As shown in Section 5.2.1 and Fig. 4, jointly-trained models outperform the sequentially-trained ones.
>
> All in all, **our approach is more general, flexible, and expressive than L-OpInf** and allows us to learn the high-dimensional dynamics of systems that require **non-linear reductions** to feature good approximation properties.
>
>
> ### **Additional experimental comparison to prior work**
>
> 1. **Comparisons with L-OpInf [4]:** We implemented L-OpInf and considered it as an additional baseline in Section 5.2. The results on the 16-DoF coupled pendulum are reported in Table 1. We observe that L-OpInf shows significantly higher prediction errors than our propose RO-LNN. Specifically, it results in a position prediction error of 2 magnitude higher than the RO-LNN trained on acceleration, and 4 orders of magnitude higher than the RO-LNN trained with multi-step integration. Notice that we only provide results with L-OpInf for the 16-DoF coupled pendulum as, similarly to the full-order LNN, we were not able to generate stable results with the rope and cloth datasets. This illustrates the limitations of considering a linear subspace and a constant inertia matrix in the latent space, which hinder the expressivity of L-OpInf compared to our approach.
>
> 2. **Comparisons with other MOR approaches:** In our scenario, we assume that we do not have access to the high-dimensional dynamic parameters. Therefore, we cannot not compare our approach with other MOR methods such as [1,2,3], as they are invasive and assume that the high-dimensional FOM dynamics are known.
>
> 3. **Comparisons with HNN [5] and extensions:** We would like to highlight that HNNs [5] and extensions thereof are, as LNNs, limited to learning low-dimensional dynamics. We believe that additional comparisons with Hamiltonian systems go beyond the scope of this work and prefer to focus on Lagrangian models such as [4]. It is worth highlighting that preliminary experiments with HNNs on the high-dimensional systems of Section 5.2 led to similar results as LNNs.
>
>
> ### **Additional video**
>
> We added an additional video that illustrates the results of our experiments as supplementary material.

---

> ### Author Response · Authors · 2024-11-18
> **Response to all reviewers (3/3)**
>
> ### **Summary of main changes in the paper**
>
> - We refined the introduction to better highlight the contributions of our paper, as well as its novelty with respect to previous works.
> - We explicitly cited [4] and explained its differences with respect to our approach in the introduction.
> - We reformulated parts of Section 2.2, as well as Section 4.1 to highlight our contributions and emphasize the differences between our approach and traditional MOR methods, in particular [1].
> - We added an additional comparison with L-OpInf [4] in Section 5.2, highlighting the limitations of this approach to identify high-dimensional dynamics which are not linearly reducible.
> - Further minor additions corresponding to an individual reviewer’s feedback are addressed in the individual responses.
>
> ### **References**
>
> [1] Patrick Buchfink, Silke Glas, Bernard Haasdonk, and Benjamin Unger. Model reduction on manifolds: A differential geometric framework. Physica D: Nonlinear Phenomena 468 (2024).
>
> [2] Harsh Sharma, Hongliang Mu, Patrick Buchfink, Rudy Geelen, Silke Glas, and Boris Kramer. Symplectic model reduction of Hamiltonian systems using data-driven quadratic manifolds. ComputerMethods in Applied Mechanics and Engineering, 417:116402, 2023.
>
> [3] Joshua Barnett and Charbel Farhat. Quadratic approximation manifold for mitigating the Kolmogorov barrier in nonlinear projection-based model order reduction. Journal of ComputationalPhysics, 464, 2022.
>
> [4] Harsh Sharma and Boris Kramer. Preserving Lagrangian structure in data-driven reduced-order modeling of large-scale dynamical systems. Physica D: Nonlinear Phenomena, 462:134128, 2024.
>
> [5] Samuel Greydanus, Misko Dzamba, and Jason Yosinski. Hamiltonian neural networks. In Neural Information Processing Systems (NeurIPS), volume 32, 2019.
>
> [6] Yaofeng Desmond Zhong and Naomi Leonard. Unsupervised learning of Lagrangian dynamics from images for prediction and control. In Neural Information Processing Systems (NeurIPS), vol-ume 33, pp. 10741–10752, 2020.
>
> [7] Lee, K. & Carlberg, K. T. Model reduction of dynamical systems on nonlinear manifolds using deep convolutional autoencoders. J. Comput. Phys. 404, 108973 (2020).

---

### Author Response · Authors · 2024-11-25

Dear Reviewers,

We would like to warmly thank Reviewers 7VkD and wYEc for engaging in a discussion and ask if our responses addressed your remaining issues. Moreover, we would like to kindly ask Reviewers 4zWX and rYLi if our initial responses addressed your concerns. We would gladly address any remaining issues.

---

### Author Response · Authors · 2024-11-29

Dear reviewers,
We would like to sincerely thank Reviewers 7VkD & 4zWX for their valuable feedback, and for raising their scores!
We kindly encourage reviewer wYEc to provide further feedback and confirm if our response addressed their remaining concerns adequately.
Moreover, we respectfully invite reviewer rYLi to provide their thoughts on our initial response and assess whether our response resolves their raised concerns.

---

### Meta-Review · Area_Chair_Qyhf · 2024-12-21

**Metareview:**

**Summary** This paper provides a geometric approach to learning high dimensional dynamical systems.  The method involves an order reduction autoencoder model which maps system states to a lower dimensional reduced manifold. The order reduction network has layers mathematically constrained to ensure the reduced manifold is embedded and is optimized using Reimannian gradient descent. Dynamics are learned in the lower dimensional latent space with a  Lagrangian neural network approach in which an SPD network parameterizes the mass-inertia matrix and an MLP parameterizes the potential function. Together they define equations of motion in the reduced manifold.

**Strengths** This paper presents an effective method leveraging physical and geometric inductive biases for improving physical fidelity in dynamics modeling and obtaining high accuracy long-term predictions even in systems with many degrees of freedom and unknown dynamics.  The paper is clear.  The empirical evaluation shows the method improves upon previous methods with similar goals such as Lagrangian Neural Networks and L-OpInf (Sharma & Kramer).

**Weaknesses** The main shared concern of reviewers was that the work showed limited novelty relative to previous work, especially Buchfink et al and Sharma & Kramer.  The authors expanded extensively in the rebuttal and revision on the novelty and advantages relative to these and other works, convincing most reviewers.  This leads me to believe that this issue was partly a matter of the writing and evaluations in the initial paper which did not focus enough on clarifying and empirically justifying these differences relative to previous work.  That said, while most reviewers increased their scores to accept, the consensus belief is still that the significance of the contribution is not that larger. Additional concerns which were largely addressed during the rebuttal include the need for more baseline comparisons and a request to show the time complexity.

**Conclusion** The authors largely addressed most reviewer concerns and showed a reasonable contribution relative to previous work.

**Additional Comments On Reviewer Discussion:**

4zWX raised their score to 6 after concerns about novelty and ablations were addressed in the rebuttal and ablation., 7VkD also raised their score to 6 after all their concerns were addressed., rYLi raised their score after the rebuttal to 5.    The reviewers and authors are really to be commended for engaging so actively in the discussion period.

---

### Decision · Program_Chairs · 2025-01-22

Accept (Poster)